# Functionally refined encoding of threat memory by distinct populations of basal forebrain cholinergic projection neurons

**Prithviraj Rajebhosale[1†], Mala R Ananth[1†], Ronald Kim[1], Richard Crouse[2§], Li Jiang[1], Gretchen López-Hernández[3], Chongbo Zhong[1], Christian Arty[4#], Shaohua Wang[5], Alice Jone[6¶], Niraj S Desai[1], Yulong Li[7,8,9], Marina R Picciotto[2,10], Lorna W Role[1*‡], David A Talmage[1*‡]**

[1]National Institute of Neurological Disorders and Stroke, NIH, Bethesda, United States; [2]Yale Interdepartmental Neuroscience Program, Yale University, New Haven, United States; [3]Kansas City University of Medicine and Biosciences, Kansas City, United States; [4]Stony Brook University, Stony Brook, United States; [5]National Institute of Environmental Health Sciences, Durham, United States; [6]Program in Neuroscience, Stony Brook University, Stony Brook, United States; [7]State Key Laboratory of Membrane Biology, Peking University School of Life Sciences, Beijing, China; [8]PKU-IDG/McGovern Institute for Brain Research, Beijing, China; [9]Peking-Tsinghua Center for Life Sciences, Academy for Advanced Interdisciplinary Studies, Peking University, Beijing, China; [10]Department of Psychiatry, Yale University, New Haven, United States

**\*For correspondence:**
Lorna.Role@nih.gov (LWR);
david.talmage@nih.gov (DAT)

[†]These authors contributed equally to this work
[‡]These authors also contributed equally to this work

**Present address:** [§]Office of New Haven Affairs, Yale University, New Haven, United States; [#]LinkedIn Corporation, Sunnyvale, United States; [¶]Regulatory Affairs Division, STERIS Corporation, Mentor, United States

**Competing interest:** The authors declare that no competing interests exist.

**Abstract** Neurons of the basal forebrain nucleus basalis and posterior substantia innominata (NBM/SI$_p$) comprise the major source of cholinergic input to the basolateral amygdala (BLA). Using a genetically encoded acetylcholine (ACh) sensor in mice, we demonstrate that BLA-projecting cholinergic neurons can 'learn' the association between a naive tone and a foot shock (training) and release ACh in the BLA in response to the conditioned tone 24 hr later (recall). In the NBM/SI$_p$ cholinergic neurons express the immediate early gene, Fos following both training and memory recall. Cholinergic neurons that express Fos following memory recall display increased intrinsic excitability. Chemogenetic silencing of these learning-activated cholinergic neurons prevents expression of the defensive behavior to the tone. In contrast, we show that NBM/SI$_p$ cholinergic neurons are not activated by an innately threatening stimulus (predator odor). Instead, VP/SI$_a$ cholinergic neurons are activated and contribute to defensive behaviors in response to predator odor, an innately threatening stimulus. Taken together, we find that distinct populations of cholinergic neurons are recruited to signal distinct aversive stimuli, demonstrating functionally refined organization of specific types of memory within the cholinergic basal forebrain of mice.

## Editor's evaluation

This important study examines the existence of a specific population of memory encoding acetylcholine neurons of the basal forebrain which regulate the amygdala for fear expression. Using a combination of techniques including genetic access to c-Fos expressing neurons, in-vivo chemogenetics, and optical detection of acetylcholine (ACh), the authors present convincing evidence that posteriorly located amygdala projecting basal forebrain cholinergic neurons participate in cue-specific threat learning and memory. This paper will be of interest to those studying circuit-level mechanisms of learning and emotion regulation.

## Introduction

Acetylcholine (ACh) is critical for cognition. Basal forebrain cholinergic neurons (BFCNs), neurons that synthesize and release ACh that are sparsely distributed throughout the base of the forebrain, provide the primary source of ACh to the cortex, hippocampus, and amygdala. Disruptions to normal cholinergic transmission are thought to contribute to several neuropsychiatric disorders (*Sarter et al., 1999*; *Higley and Picciotto, 2014*) as well as to cognition (*Ananth et al., 2023*) and salience-related behaviors (*Jiang et al., 2016*; *Hersman et al., 2017*; *Crouse et al., 2020*). BFCNs are anatomically divided into several clusters: the medial septum/diagonal band complex (MS/DB), the ventral pallidum (VP), the substantia innominata (SI), and the nucleus basalis (NBM). Between and within these anatomical groupings, BFCNs comprise heterogenous subclusters (*Zaborszky and Gyengesi, 2012*). How this heterogeneity contributes to the significant control that cholinergic signaling exerts over large, behaviorally relevant circuits is unclear (*Zaborszky et al., 2015*; *Gielow and Zaborszky, 2017*).

ACh plays an important role in modulating emotionally salient memories (*Luchicchi et al., 2014*; *Ballinger et al., 2016*; *Knox, 2016*; *Ananth et al., 2023*). We and others have found that cholinergic signaling in the basolateral amygdala (BLA) is important for generating defensive behaviors in response to both learned and innate threats (*Power and McGaugh, 2002*; *Jiang et al., 2016*; *Wilson and Fadel, 2017*). Optogenetic manipulation of endogenous ACh release in the BLA during learning modulates the expression of threat response behaviors in mice upon recall of a conditioned stimulus (*Jiang et al., 2016*). Stimulating release of ACh increases activity of BLA principal neurons, in part by increasing the release probability of glutamatergic inputs to these neurons, and is sufficient to induce long-term potentiation (LTP) when paired with minimal (non-LTP generating) stimulation of glutamatergic input to the BLA (*Unal et al., 2015*; *Jiang et al., 2016*). Memory formation and retrieval are associated with fast synaptic mechanisms that are modulated by ACh, that are in turn necessary for the proper learning and expression of threat response behaviors (*Nonaka et al., 2014*). Given the broad distribution of cholinergic input across the BLA, and the well-established role of ACh in modulating BLA plasticity, the basal forebrain cholinergic system is well positioned to serve an important role in the encoding of threat memories and generation of threat response behaviors (*Ananth et al., 2023*).

The BLA receives dense cholinergic input from neurons located in various regions within the basal forebrain (such as the VP, SI, and NBM). In this study, we asked how these distinct populations of BLA-projecting BFCNs contribute to threat responses. Using a genetically encoded ACh sensor, activity-dependent genetic tagging, chemogenetic manipulations, and electrophysiological recordings, we identify a population of BFCNs in the NBM/$SI_p$ ($SI_p$ defined as the portion of the sublenticular SI posterior to bregma −0.4 mm) that are required for learned threat responsiveness. We find that NBM/$SI_p$ cholinergic neurons are necessary for freezing behavior following cue-conditioned threat learning while freezing behavior elicited by an innately threatening stimulus activates cholinergic neurons in the VP/$SI_a$ (VP/$SI_a$; $SI_a$ defined as the portion of the SI ventral to the anterior commissure located anterior to bregma −0.4 mm).

## Results

Animals recognize varied sensory stimuli and categorize them as either threatening or non-threatening. Recognition of threatening stimuli can be innate or acquired, for example, by association of an aversive experience with an innocuous, co-occurring sensory input. In this study, we sought to understand if the basal forebrain cholinergic system participates in the encoding of associative threat or in response to innate threat.

### ACh is released in the basal lateral amygdala in response to threat

The BLA plays a central role in associative threat learning and in the generation of threat responses. We have previously demonstrated that silencing cholinergic input to the BLA during cue-conditioned threat learning (pairing a naive tone with a foot shock) blunts learned freezing in response to the conditioned stimulus (tone) (*Jiang et al., 2016*). Given this, the first question we asked was whether ACh was released in the BLA during associative threat learning (*Figure 1*, *Figure 1—figure supplements 1–4*). To monitor acute changes in extracellular ACh levels during the cue conditioned threat-learning task, we expressed a genetically encoded ACh sensor, GRAB$_{ACh3.0}$ (*Jing et al., 2018*; *Jing et al., 2020*) in BLA neurons and visualized fluorescence using fiber photometry (*Figure 1A*). Our

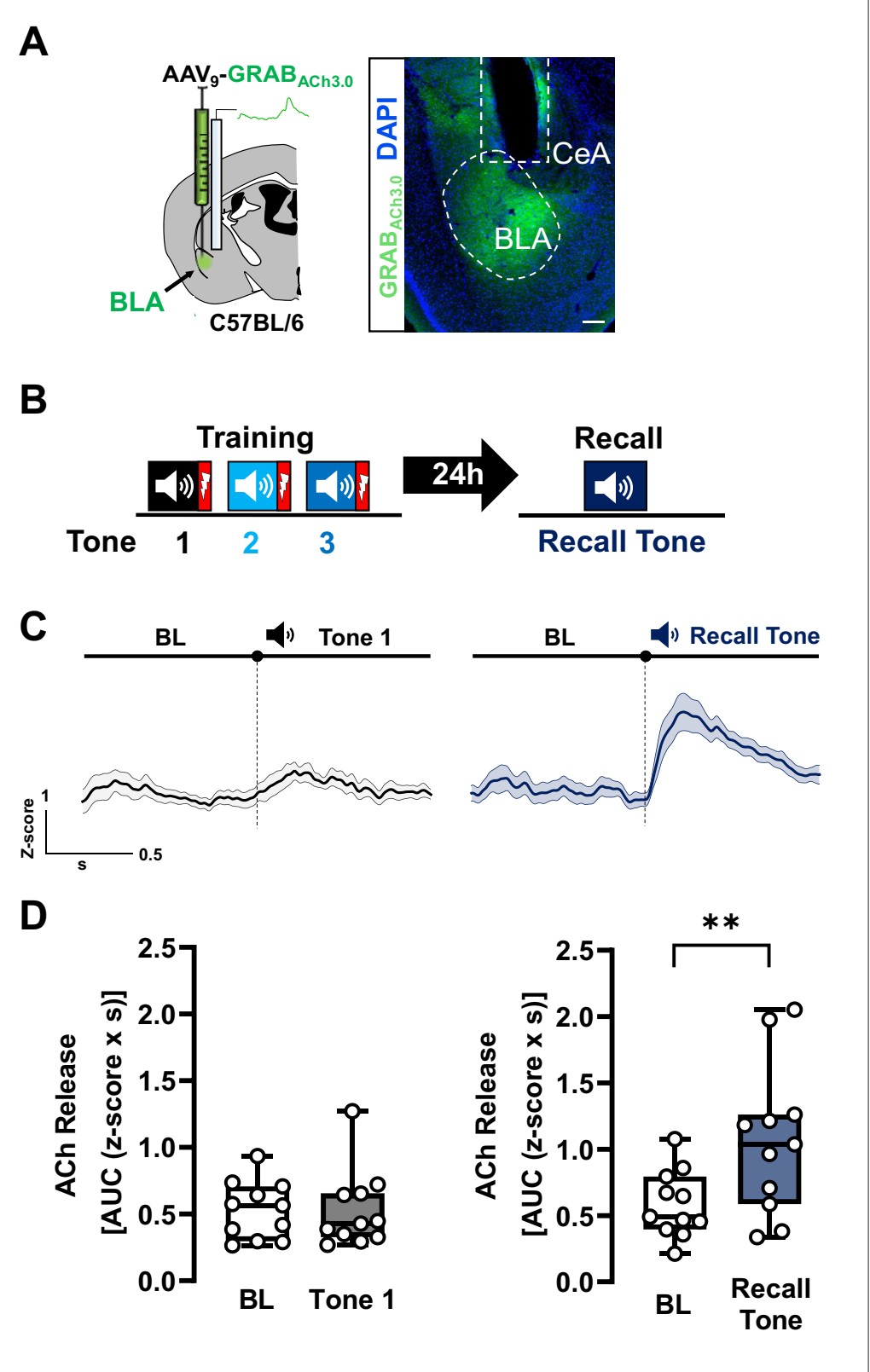

**Figure 1.** Acetylcholine (ACh) is released in basolateral amygdala (BLA) during threat recall (see also *Figure 1— figure supplements 1–4*). (**A**) Left: Schematic of strategy for targeting the genetically encoded ACh sensor (GRAB_{ACh3.0}) to BLA. Right: Image of ACh sensor expression (green). White dotted oval delineates ACh sensor-labeled BLA. White dotted box denotes prior location of optical fiber. Scale bar = 100 μm. Please refer to

*Figure 1 continued on next page*

*Figure 1 continued*

**Figure 1—figure supplement 4** for fiber placement mapping in all mice. (**B**) Schematic of the associative threat-learning protocol employed consisting of three tone + shock pairings during the training period and tone alone during the recall session. (**C**) Average traces of ACh release in response to tone; shading represents standard error of the mean (SEM): naive tone (Tone 1 during training) in black, recall tone in deep blue (tone onset indicated by bar above; $n$ = 11). BL = baseline. (**D**) Quantification of ACh release (area under the curve (AUC)) during baseline period (1 s prior to tone onset) and in response to the first (naive) tone and in response to the recall tone (1 s following tone onset). Naive tone did not induce significant increase in ACh release in the BLA (Wilcoxon matched-pairs signed rank test, BL vs. Tone 1, p = 0.8311, $W$ = −6). Recall tone induced a significant increase in ACh release in the BLA (paired *t*-test, BL vs. Recall tone, ** p = 0.0039 (two-tailed), $t$ = 3.732, df = 10). See also **Figure 1—figure supplements 1–4**.

The online version of this article includes the following source data and figure supplement(s) for figure 1:

**Source data 1.** Processed ACh release fiber photometry data for **Figure 1**.

**Figure supplement 1.** Exposure to tones during tone–shock pairings does not significantly alter acetylcholine (ACh) release in the basolateral amygdala (BLA) during training.

**Figure supplement 2.** Repeat tone exposures without foot shock during either a training session or a recall session fail to induce significant changes in acetylcholine (ACh) release in the basolateral amygdala (BLA).

**Figure supplement 2—source data 1.** Processed ACh release fiber photometry data for **Figure 1—figure supplement 2**.

**Figure supplement 3.** Shock alone does not sensitize cholinergic tone responses in the basolateral amygdala (BLA).

**Figure supplement 3—source data 1.** Processed ACh release data for **Figure 1—figure supplement 3**.

**Figure supplement 4.** Remapping of optic fiber placements.

associative threat-learning protocol involved placing mice in a novel chamber and exposing them to an 80 dB tone for 30 s. During the final 2 s of the tone the mice received a foot shock (0.7 mA). The tone–shock pairing was repeated twice (for a total of three pairings). Twenty-four hours later, mice were placed in a different chamber (with different tactile, visual, and olfactory cues to the training chamber) and exposed to tone alone.

Foot shock, either alone or paired with tone, increased ACh release in the BLA whereas the naive tone that is the first tone before shock presentation (Tone 1), did not (**Figure 1—figure supplement 1C**, **Figure 1C, D**, left; baseline (BL) vs. Tone 1, p = 0.8311). In contrast to Tone 1, the recall tone, presented 24 hr after the three tone–shock pairings, resulted in significant increase in ACh release in the BLA (**Figure 1C, D**, right; p = 0.0039). The change in tone-associated ACh release required pairing with foot shock: naive tone (**Figure 1—figure supplement 2C** left, p = 0.8437), three consecutive tones alone (without shock), or a subsequent repeat tone presentation after 24 hr (not previously paired with shock) (**Figure 1—figure supplement 2C**, right; p = 0.3152), did not induce significant changes in ACh release in the BLA (**Figure 1—figure supplement 2**).

To verify that the increases in ACh release were indeed specific to the tone–shock association and not due to generalization from prior shock exposure, we also subjected mice to three shocks (day 1) followed by a tone presentation 24 hr later (day 2) (**Figure 1—figure supplement 3A**). While mice demonstrated freezing behavior during the session on day 2, there was no significant increase in freezing behavior to the 24 hr tone presentation (**Figure 1—figure supplement 3C**, p = 0.2418). There was no increase in ACh in response to the tone when it was not explicitly paired with a shock, confirming that the changes in ACh release were indeed associative (**Figure 1—figure supplement 3D**; baseline (pre-tone, day 2)–24 hr tone (tone presentation, day 2): p = 0.7272). Therefore, after repeated tone–shock pairings, BLA-projecting cholinergic neurons acquire enhanced tone responsiveness.

## NBM/SI$_p$ cholinergic neurons are activated by threat learning and reactivated during threat memory recall

Following associative threat-learning, cholinergic neurons exhibited increased ACh release in the BLA in response to a previously innocuous auditory stimulus; this increase occurred exclusively following pairing of the tone with a shock. Using a two-color labeling system, we asked whether NBM/SI$_p$

cholinergic neurons were activated during the training session and reactivated during the recall session. To do this, we injected the offspring of a cross of Chat-IRES-Cre x Fos-tTA:Fos-shGFP with a viral vector, AAV$_9$-TRE-DIO-mCherry-P2A-tTA$^{H100Y}$, resulting in <u>a</u>ctivity (tTA) <u>d</u>ependent, <u>C</u>re <u>d</u>ependent (aka ADCD) mCherry expression (see methods and *Figure 2—figure supplement 1*). These mice carry three transgenes: one encoding Cre recombinase in cholinergic neurons, a second doxycycline (Dox) repressible, tetracycline transactivator (tTA) expressed following activation of the *fos* promoter, and a third destabilized green fluorescent protein (short half-life GFP) also under transcriptional regulation of the *fos* promoter. tTA and shGFP are transiently expressed in activated neurons. In the absence of Dox (delivered via chow diet), activation of Cre-expressing cholinergic neurons leads to tTA expression and expression of the virally transduced mCherry along with a mutant tTA, which is insensitive to Dox. Thus, after closure of the labeling window by re-administration of Dox, cholinergic neurons activated during the Dox off period maintain mCherry expression permanently driven by the mutant tTA. When ADCD labeling is coupled with the transient expression of Fos–shGFP, we can label and visualize participation of cholinergic neurons in two separate behavioral sessions (mCherry+ = session 1 activated cells and GFP+ = session 2 activated cells) (*Figure 2—figure supplement 1B*).

Two to three weeks following injection with the ADCD virus, mice were either (1) kept in home cage throughout, (2) exposed to tone without foot shock (tone alone), or (3) put through the standard threat-learning paradigm (tone + shock). Twenty-four hours prior to the training session (session 1) mice were switched from Dox-containing to Dox-free chow to allow function of tTA. Immediately following tone–shock pairings, mice were placed back on Dox-containing chow (*Figure 2A*). This switch from Dox on → Dox off → Dox on was also performed for mice that remained in their home cages and for those that were exposed to tones without shock. Recall was performed 72 hr later (tone alone in a new context), and mice were sacrificed ~2.5 hr following recall (the peak of the Fos–shGFP expression). We quantified the number of mCherry+/GFP+ (double positive) neurons following session 2 (e.g. white arrow, *Figure 2B*). Significantly more double positive cholinergic neurons were seen following the complete associative threat-learning paradigm (tone + shock followed by tone recall) compared to mice that underwent session 1 without shocks (*Figure 2C*, p = 0.0249). To further ensure that the reactivation of these cholinergic neurons was not due to a generalized increase in responsiveness of these neurons following shock exposure, we quantified reactivated neurons in mice exposed to shock alone during session 1 followed by tone alone during session 2 (shock alone (session 1) → tone alone (session 2)) along with shock alone (session 1) → home cage (session 2), and home cage controls (*Figure 2D*). All three conditions showed few reactivated neurons and no differences between groups (p = 0.9471). Thus, associative threat-learning results in activation of NBM/SI$_p$ cholinergic neurons which are reactivated during subsequent cue-induced memory recall.

## Reactivation of cholinergic neurons activated by training is required for learned behavioral responses

BLA-projecting cholinergic neurons acquire tone responsiveness following associative threat learning (*Figure 1*) and a population of NBM/SI$_p$ cholinergic neurons are activated during tone–shock pairing and reactivated during the recall session (*Figure 2*). If these cholinergic neurons are indeed part of a threat memory engram, then their reactivation would be required for generation of learned threat responses. To block reactivation of cholinergic neurons in response to tone, we expressed the inhibitory, designer receptor hM4Di, in an activity dependent, Cre-dependent manner in NBM/SI$_p$ cholinergic neurons (ADCD-hM4Di; *Figure 3A*) and subjected these mice to the threat-learning paradigm (*Figure 3A*). Mice were taken off Dox-chow 24 hr prior to the training session, immediately placed back on Dox-chow after training, and then tested for tone recall after 72 hr. ADCD-hM4Di and sham operated control mice were injected with clozapine (CLZ; 0.1 mg/kg; injected intraperitoneally (i.p.)) 10 min prior to the recall session to selectively silence the population of NBM/SI$_p$ cholinergic neurons that were previously activated during training (*Figure 3A*). Freezing behavior was quantified during both the training and recall sessions. Freezing was compared between the 'Pre-Tone' period and 'Recall Tone Response' (defined as freezing occurring from the onset of the recall tone through the end of the recall session) (*Figure 3—figure supplement 1B*). Both groups of mice showed the same freezing behavior during the training session (*Figure 3C*, p = 0.6482, *Figure 3—figure supplement 1A*). In the recall session, sham mice displayed typical freezing behavior in response to tone (*Figure 3D*, gray boxes; Pre-Tone vs. Recall Tone Response, p = 0.0001). In contrast, ADCD-hM4Di

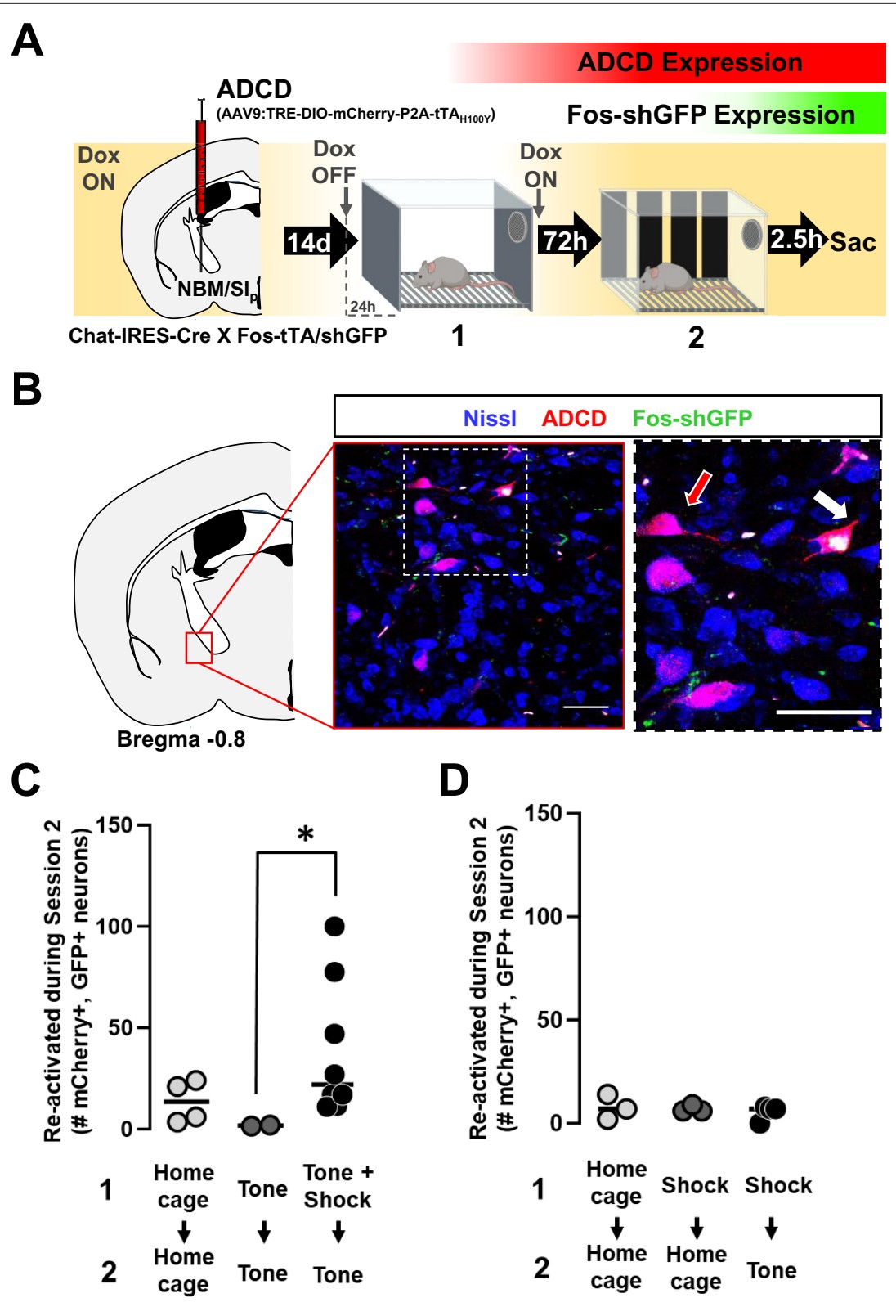

**Figure 2.** Nucleus basalis and posterior substantia innominata (NBM/SIp) cholinergic neurons are activated by threat learning and reactivated during threat memory recall (see *Figure 2—figure supplement 1*). (**A**) Strategy for labeling activated NBM/SIp cholinergic neurons during both training and recall. Chat-IRES-Cre × Fos-tTA/shGFP mice ($n$ = 14) were injected in the NBM/SIp with ADCD-mCherry virus (AAV9: TRE-DIO-mCherry-P2A-tTA_H100Y). During session 1 (off Dox) mice either remained in their home cage, were exposed to three tones (Tone alone), or were exposed to three

*Figure 2 continued on next page*

*Figure 2 continued*

tone–shock pairings (training session). During session 2, mice remained in home cage or were exposed to a single tone (recall session). Cholinergic neurons activated during training express ADCD-mCherry stably after training (red during training), and neurons activated during recall transiently express GFP (green during recall). (**B**) Image of the NBM/SI$_p$ showing cholinergic neurons activated during training (red arrow) or by both training and recall (reactivated – white arrow; image taken at A/P ~ −0.8 from Bregma; scale bar = 50 μm). (**C**) Quantification of the number of cholinergic neurons activated during session 1 (ADCD-mCherry+) that were reactivated during session 2 (both mCherry and GFP positive (activated both during session 1 (training) and during session 2 (recall))). Home cage (*n* = 7 sections from 4 mice), tone only (*n* = 4 sections from 2 mice) and tone + shock (*n* = 17 sections from 8 mice) conditions. Significantly more cholinergic neurons were reactivated by tone following tone–shock pairings (Kruskal–Wallis p = 0.0249). Tone–shock compared to tone only (*p = 0.0464, Dunn's corrected). (**D**) Quantification of number of reactivated cholinergic neurons (activated both during session 1 and during session 2 vs. the total number of cholinergic neurons activated during session 1). Home cage → home cage (*n* = 9 sections from 3 mice), shock → home cage (*n* = 17 sections from 4 mice), and shock → tone (*n* = 11 sections from 3 mice). Kruskal–Wallis p = 0.9471, KW = 0.1219.

The online version of this article includes the following figure supplement(s) for figure 2:

**Figure supplement 1.** Construction of viral vectors for tagging activated cholinergic neurons.

mice did not show increased freezing in response to the tone (*Figure 3D*, red boxes; BL vs. tone response, p = 0.8451). Overall ADCD-hM4Di mice showed lower freezing behavior compared to sham controls (*Figure 3D*; sham – gray, hM4Di – red: p = 0.0052), indicating that reactivation of training-activated NBM/SI$_p$ cholinergic neurons during the recall session was required for the expression of learned threat response behavior.

## BLA-projecting NBM/SI$_p$ cholinergic neurons are reactivated during threat memory recall

To investigate whether NBM/SI$_p$ cholinergic neurons that are reactivated during recall are BLA projecting, we injected Chat-IRES-Cre × Fos-tTA:Fos-shGFP mice with ADCD-mCherry in the NBM/SI$_p$, and simultaneously delivered the retrograde tracer Fast Blue (FB) into the BLA (*Figure 4A*). The mice were taken off doxycycline containing chow during the training period, returned to dox-chow for 72 hr and then exposed to the tone alone. We then quantified BLA-projecting cholinergic neurons that were reactivated by tone (ChAT immunoreactive, FB labeled and ADCD-mCherry+/Fos-shGFP+; *Figure 4C*). We found that ~20% of NBM/SI$_p$ cholinergic neurons in both the home cage and the threat-learning + recall paradigm group (at Bregma −0.8 mm) were labeled with FB, with no significant differences in the percentage of cholinergic neurons with retrograde label between groups (*Figure 4D*; p = 0.5192). Next, we quantified the percentage of BLA-projecting NBM/SI$_p$ cholinergic neurons that were active during session 1 and reactivated during session 2. We found that, on average, ~21% of BLA-projecting cholinergic neurons were reactivated during recall (*Figure 4E*). This reactivation of BLA-projecting BFCNs was significantly higher in mice that underwent training + recall compared to mice that remained in their home cage but still underwent the Dox on → Dox off → Dox on protocol (*Figure 4E*; p = 0.0183). Based on these data we conclude that BLA-projecting BFCNs are activated by associative threat learning and reactivated by threat recall.

## Silencing BLA-projecting BFCNs during training or recall prevents activation of BLA neurons and conditioned freezing behavior

To determine whether chemogenetic silencing of BLA-projecting cholinergic neurons during training or during recall interfered with the activation of BLA neurons, we injected the BLA of Chat-IRES-Cre mice with CAV$_2$-DIO-hM4Di.mCherry and AAV$_9$-camk2a-GCaMP (cav.hM4Di$^{BLA}$ mice) or AAV$_9$-camk2a-GCaMP alone (sham mice) (*Figure 5A*, *Figure 5—figure supplement 1A*; GFP fluorescence from GCaMP was used to mark the injection sites). We found mCherry was expressed in cholinergic neurons predominantly in the NBM/SI$_p$, followed by the VP/SI$_a$, with a small contribution from the horizontal limb of the diagonal band of Broca (hDB) (*Figure 5A*, right). These data support previous findings (*Zaborszky and Gyengesi, 2012*) that NBM/SI$_p$ cholinergic neurons provide a major input to the BLA.

We injected cav.hM4Di$^{BLA}$ or sham control mice with CLZ 10 min prior to initiating cue-conditioned threat learning (*Figure 5B*) or 10 min prior to the memory recall session (*Figure 5C*). In both experiments, mice were sacrificed 45–60 min following recall and assessed for Fos immunoreactivity (IR) in the BLA. We found that DREADD-mediated silencing of BLA-projecting cholinergic neurons during

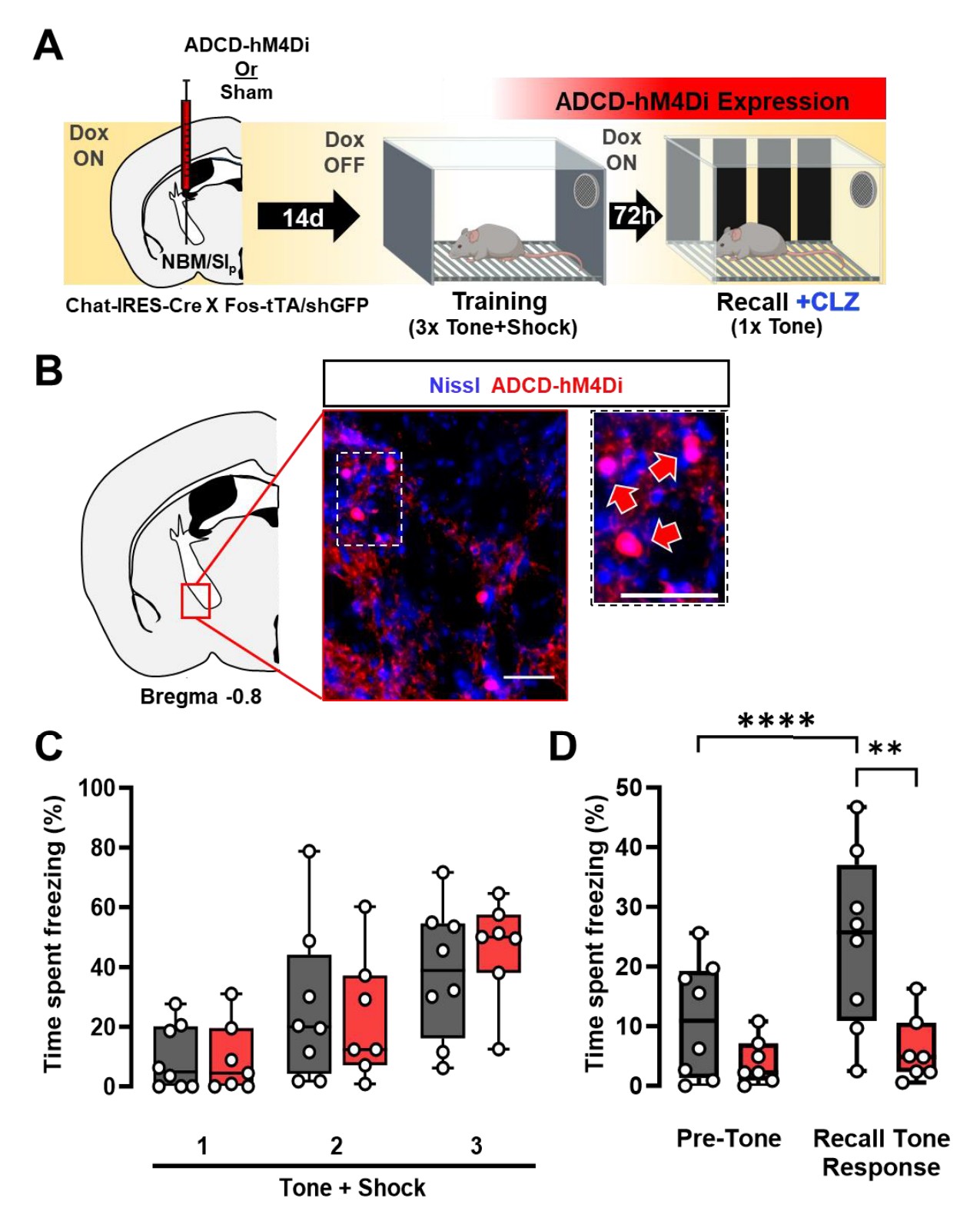

**Figure 3.** Re-activation of a subset of nucleus basalis and posterior substantia innominata (NBM/SI$_p$) cholinergic neurons is required for threat memory retrieval (see *Figure 3—figure supplement 1*). (**A**) ADCD-hM4Di (AAV9: TRE-DIO-hM4Di.mCherry) was injected into the NBM/SI$_p$ of Chat-IRES-Cre × Fos-tTA/shGFP mice. Two weeks later mice underwent training on regular chow (Dox-chow removed 24 hr prior to training session) to allow hM4Di. mCherry to be selectively expressed in training-activated cholinergic neurons. Three days later, recall was tested in Dox on conditions. Clozapine

*Figure 3 continued on next page*

Figure 3 continued

(CLZ) was injected 10 min before the recall session to activate the inhibitory DREADD, hM4Di specifically in previously activated cholinergic neurons. (B) Representative image taken at Bregma −0.8 mm of mCherry (ADCD-hM4Di.mCherry) expressing cells. Inset shows higher magnification images of ADCD expression (red arrows). Scale bar = 50 μm. (C) Freezing behavior during training in sham (gray, n = 8 mice) and ADCD-hM4Di injected (red, n = 7 mice) for each 30 s bin during tone presentation (Tone + Shock 1, 2, 3). There were no significant differences between the groups during the training session (repeated measures (RM) two-way analysis of variance (ANOVA) Time × Group p = 0.6482; Group p = 0.7311). (D) Freezing behavior during recall following selective hM4Di-mediated inhibition of training-activated cholinergic neurons in the NBM/SI$_p$. Sham (gray, n = 8 mice) and hM4Di (red, n = 7 mice) groups. There were significant differences between pre-tone vs. tone-related freezing for sham (****Pre-Tone vs. Recall Tone Response, p = 0.0001, Bonferroni corrected), response to tone between sham and hM4Di (**p = 0.0026, Bonferroni corrected) and a significant main effect of Time × Group interaction (RM two-way ANOVA (GLM) Time × Group, p = 0.0052). (See *Figure 3—figure supplement 1* for details on time periods comprising Pre-Tone and Recall Tone Response periods.)

The online version of this article includes the following figure supplement(s) for figure 3:

**Figure supplement 1.** Time-resolved freezing plot from recall following silencing of training-activated NBM/SI basal forebrain cholinergic neurons (BFCNs).

training alone blunted recall-induced freezing behavior and activation of BLA neurons (*Figure 5B*: freezing behavior, sham vs. cav.hM4Di$^{BLA}$ (Recall Tone Response), p < 0.0001, *Figure 5b', b''*: Fos density, sham vs. cav.hM4Di$^{BLA}$ p = 0.0286). Similarly, DREADD-mediated silencing of BLA-projecting cholinergic neurons during recall alone also reduced recall-induced freezing and activation of BLA neurons (*Figure 5C*: freezing behavior, sham vs. cav.hM4Di$^{BLA}$ (Recall Tone Response) p = 0.0279, *Figure 5c', c''*: Fos density, sham vs. cav.hM4Di$^{BLA}$ p = 0.0317). Mice in both sham groups showed equivalent freezing behavior (*Figure 5B, C*, gray boxes; comparing sham groups, p = 0.8155) and density of Fos-IR cells (*Figure 5b', b''*, black circles; comparing sham groups, p = 0.5273) indicating that 0.1 mg/kg CLZ alone (in the absence of DREADD expression) did not alter Fos expression or expression of the learned threat response behavior. Thus, activity of BLA-projecting cholinergic neurons is required during both training and recall for recall induction of Fos expression in BLA neurons and freezing behavior. Preventing cholinergic neuron activity during either training or recall significantly reduced the density of Fos+ BLA neurons and tone-induced freezing.

Differences in recall-induced Fos expression between sham and cav.hM4Di$^{BLA}$ mice were maximal in rostral portions of the BLA (between bregma −0.8 and −1.4 mm) (*Figure 5—figure supplement 1B*). This region of the rostral BLA has been shown to contain genetically distinguishable neurons that are activated by aversive stimuli and preferentially project to the capsular portion of the central amygdala (CeC), a region known to drive freezing behavior (*Kim et al., 2016*; *Kim et al., 2017*). We examined the CeC of mice in which BLA-projecting BFCNs were silenced during recall and found significantly reduced Fos+ cell density in cav.hM4Di$^{BLA}$ mice compared to control mice (*Figure 5—figure supplement 1C* sham vs. cav.hM4Di$^{BLA}$ p = 0.0091). Thus, silencing cholinergic input to the BLA altered activation of BLA circuits involved in the execution of defensive behaviors.

Mapping BLA-projecting BFCNs infected by CAV$_2$-DIO-hM4Di revealed that the majority of the cholinergic input to the BLA originates in the NBM/SI$_p$ (*Figure 5A*). As such, we delivered AAV$_9$-DIO-hM4Di.mCherry or AAV$_9$-DIO-eCFP (sham mice) into the NBM/SI$_p$ of Chat-IRES-Cre mice (*Figure 5—figure supplement 2*). Both hM4Di and eCFP animals were injected with CLZ 10 min prior to the recall session. Animals in which NBM/SI$_p$ cholinergic neurons were silenced during the recall session did not show increased freezing in response to tone (*Figure 5—figure supplement 2A*, sham, gray boxes: Pre-Tone to Recall Tone Response, p = 0.0004; cav.hM4Di$^{NBM}$, red boxes: Pre-Tone to Recall Tone Response, p > 0.9999). Thus, silencing NBM/SI$_p$ BFCNs was sufficient to block expression of the learned threat response behavior.

## Recall-induced activation of NBM/SI$_p$ cholinergic neurons correlates with the degree of threat response behavior

During recall, we observed variability in individual freezing responses to the conditioned tone. Based on their responsiveness, we stratified the mice into two groups – high and low responders. 'High Responders' were defined as mice who showed a >10 percentage points increase in time spent freezing in response to the tone compared to the pre-tone period (see methods for further details).

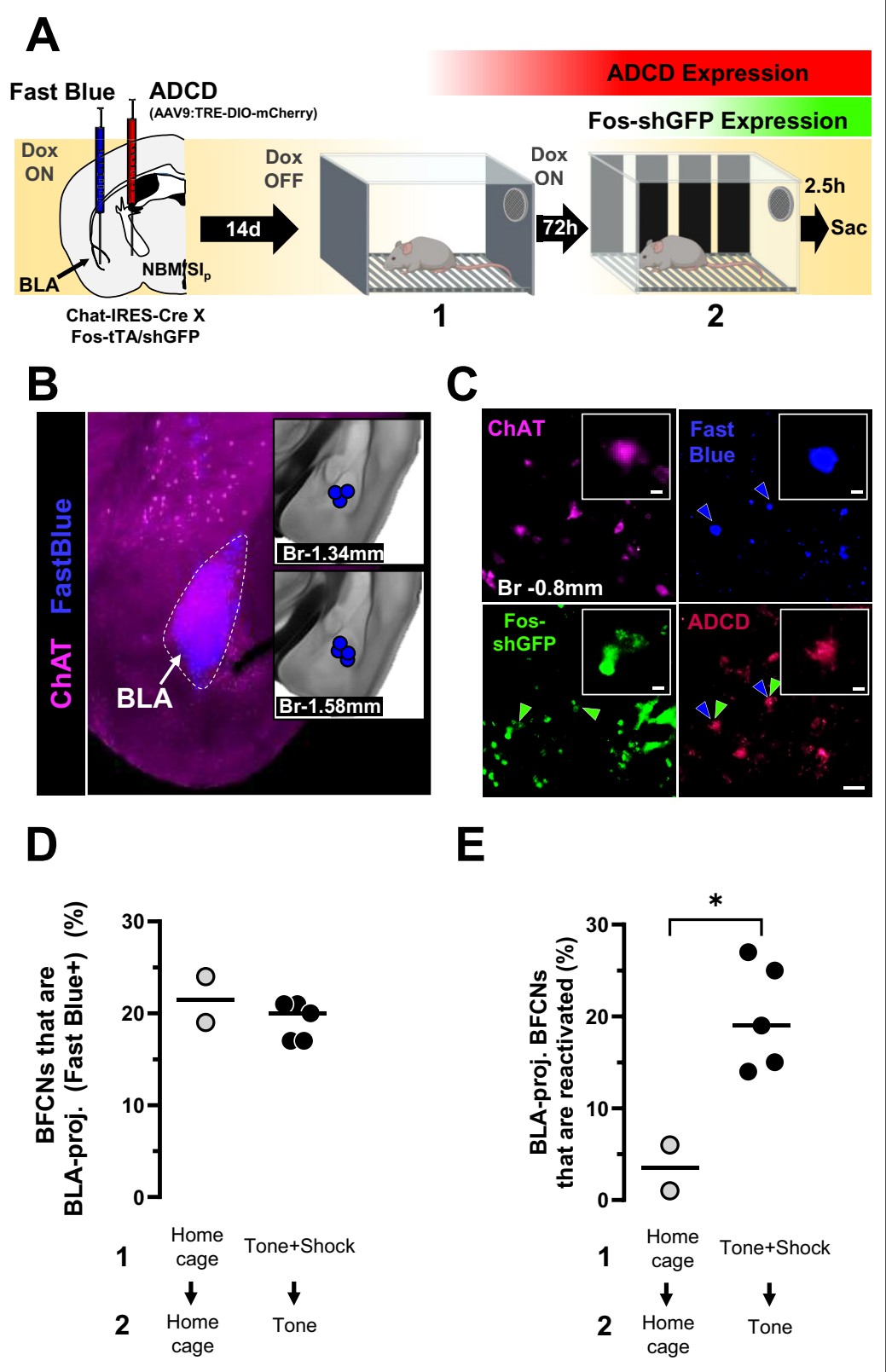

**Figure 4.** Basolateral amygdala (BLA)-projecting nucleus basalis and posterior substantia innominata (NBM/SI$_p$) cholinergic neurons are reactivated by the conditioned tone stimulus. (**A**) Left: Strategy for labeling activated NBM/SI$_p$ cholinergic during both training <u>and</u> recall along with mapping of BLA-projecting neurons. Chat-IRES-Cre × Fos-tTA/shGFP mice (n = 7) were injected in the NBM/SI$_p$ with ADCD-mCherry virus and in the BLA with

*Figure 4 continued on next page*

*Figure 4 continued*

Fast Blue dye. During session 1 (off Dox) mice either remained in their home cage or were exposed to three tone–shock pairings. During session 2 (recall session), mice remained in home cage or were exposed to a single tone. Cholinergic neurons activated during training express GFP transiently and express ADCD-mCherry stably after training (red during training), and neurons activated during recall transiently express GFP (green during recall). Neurons projecting to the BLA were labeled by Fast Blue (blue). Cholinergic neurons were identified by ChAT staining (magenta). (**B**) Image of a Fast Blue injection site in the BLA; Inset: Mapping of injection sites for all Fast Blue experiments. (**C**) Representative image showing (clockwise), ChAT+ neurons in the NBM/SI$_p$ at bregma −0.8 mm (magenta), BLA-projecting neurons (blue, blue arrowheads), training-activated cells (ADCD) (red), and recall-activated neurons (green, green arrow heads). BLA-projecting basal forebrain cholinergic neurons (BFCNs) activated by training and recall are denoted by double arrowheads (blue and green). Scale bar = 50 µm. Inset scale bar = 10 µm. (**D**) Quantification of percentage of ChAT+ neurons that were labeled by Fast Blue in mice from the home cage group (gray) and mice from the training + recall group (black) from bregma −0.8 mm. No significant differences were found between groups (Welch's *t*-test, p = 0.5192). (**E**) Quantification of percentage of BLA-projecting BFCNs (ChAT+/Fast Blue+) at bregma −0.8 mm that were reactivated during session 2 (ADCD + GFP) in mice from the home cage group (*n* = 2) (gray) and mice from the training + recall group (*n* = 5) (black). Mice that underwent training and recall had significantly higher numbers of engram-enrolled BLA-projecting BFCNs (Welch's *t*-test, *p = 0.0183).

Mice with <10 percentage points increase in time spent freezing in response to the tone compared to the pre-tone period were defined as 'Low Responders.' When stratified as high or low responders according to this criterion, only High Responders showed a statistically significant increase in freezing during the recall tone compared to the pre-tone period (*Figure 6A*; Pre-tone vs. tone: High Responders, p = 0.0016; Low Responders, p > 0.9999). High Responders showed more freezing compared to Low Responders specifically during the recall tone presentation (High vs. Low responders: recall tone blue shading, p = 0. 0454). 'High Responders' spent more time freezing in response to the tone compared to the pre-tone period (*Figure 6B*).

We next examined whether there was a relationship between the extent of freezing and the engagement of the cholinergic neurons. Since the majority of training-activated cholinergic neurons were reactivated during recall (in high responding mice −~82%, *Figure 6—figure supplement 1A*), we labeled cholinergic neurons activated during the recall session with ADCD-mCherry (on dox during training, off dox during recall; *Figure 6C*). Next, we quantified the fold change in the number of mCherry+ neurons in each group relative to corresponding home cage control mice (*Figure 6D*). While there was no difference in mCherry expression in Low Responders compared to the home cage group (fold change ~1, p > 0.9999), High Responders displayed a threefold increase (p = 0.0121) in mCherry+ cells (High Responders vs. Low Responders, p = 0.0121, *Figure 6D*).

Mapping of recall-activated NBM/SI$_p$ cholinergic neurons revealed that activated BFCNs in 'High Responder' mice were in anatomically distinct regions from those in 'Low Responder' mice (*Figure 6E*). In a different cohort of 'wild-type' mice, we assessed Fos and ChAT expression following recall and found that in the Low Responders, few ChAT and Fos co-labeled neurons were found. These co-labeled cells were located in caudal regions of the NBM/SI$_p$ (~Bregma −1.3; *Figure 6—figure supplement 2A* – bottom row). In High Responders an additional population of activated cholinergic neurons in more rostral portions of the NBM/SI$_p$ was found (~Bregma −0.8; *Figure 6—figure supplement 2A* – top row). Thus, a discrete population of activated cholinergic neurons in the rostral NBM/SI$_p$ is present in mice that respond to the learned threat. When comparing retrograde mapping of BLA-projecting cholinergic neurons using CAV$_2$-DIO-hM4Di.mCherry (*Figure 5*) to the distribution of ADCD-mCherry-labeled activated neurons (*Figure 6*), we find a similar distribution along the rostro-caudal axis of the NBM/SI$_p$ (*Figure 6—figure supplement 2B, C*).

Finally, we examined the proportion of high and low responding mice in our experiments where we silenced BLA-projecting cholinergic neurons either during training or during recall (*Figure 5B, C*). We found that under sham conditions (no cholinergic silencing), 80–90% of the mice were 'High Responders'. Silencing BLA-projecting cholinergic neurons during training shifted the proportion such that 100% of the mice were 'Low Responders' (*Figure 6—figure supplement 1B* sham vs. cav.hM4Di$^{BLA}$ inhibition during training). Silencing BLA-projecting cholinergic neurons during recall

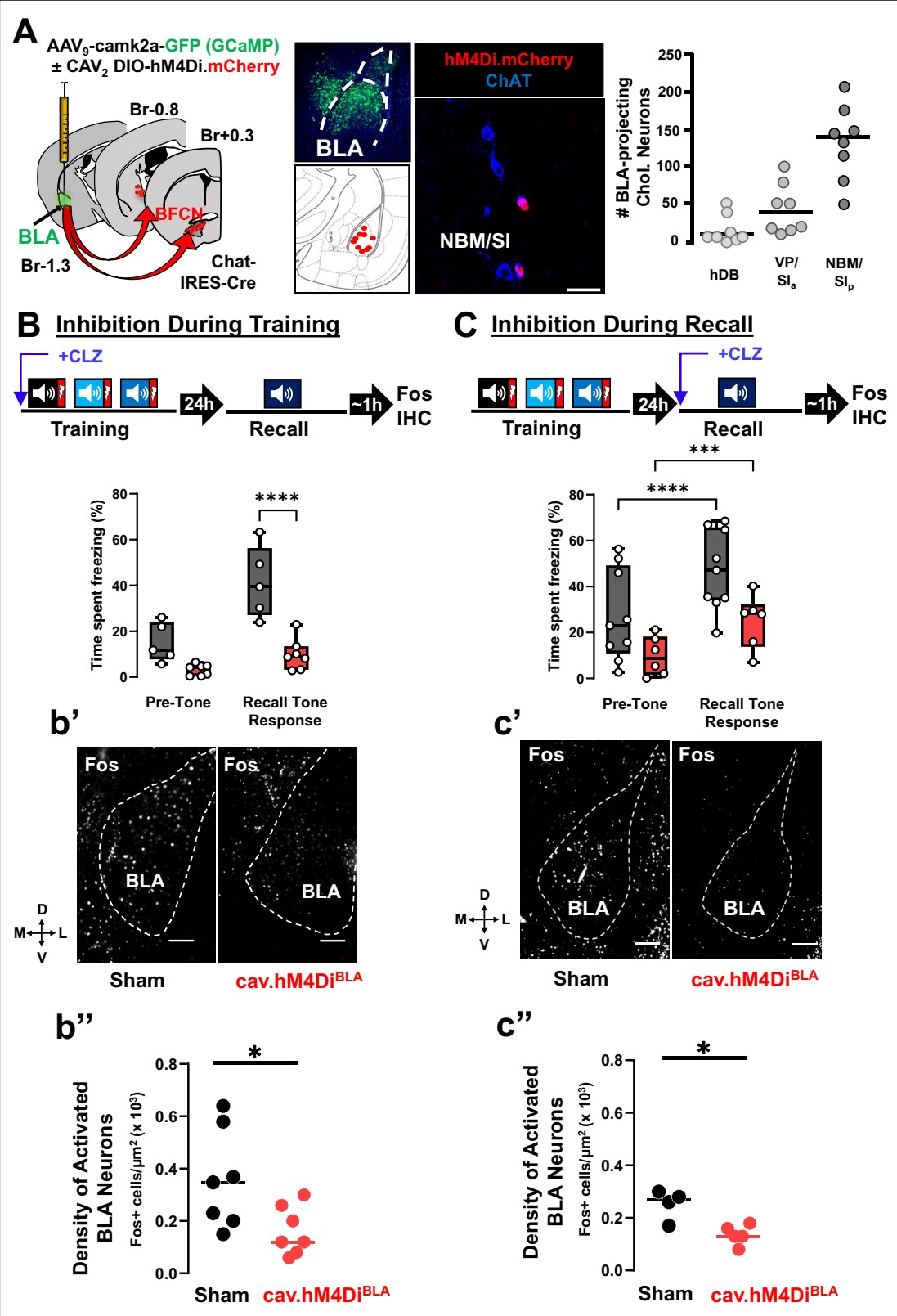

**Figure 5.** Basolateral amygdala (BLA)-projecting cholinergic neuronal activity is required both during training and during recall for learned threat processing (see *Figure 5—figure supplements 1 and 2*). (**A**) Left: Strategy for retrograde targeting of hM4Di DREADD to BLA-projecting cholinergic neurons. Middle: Re-localization of BLA injection sites (using AAV9-camk2a-GCaMP6f to mark the injection site), and identification of retrogradely labeled cholinergic neurons within the nucleus basalis and posterior substantia innominata (NBM/SIp; scale bar = 50 μm). Right: Quantification of

*Figure 5 continued*

hM4Di-expressing cholinergic populations (mCherry+) across the basal forebrain (*n* = 8 mice, 56–80 sections) (Bregma +0.6 mm to −1.5 mm). (**B**) BLA-projecting cholinergic neurons were silenced by injecting mice with clozapine (CLZ) 10 min prior to training. Percent time freezing during the recall session including the pre-tone (baseline) period and in response to the conditioned tone. CLZ was only administered during the training session (repeated measures (RM) two-way analysis of variance (ANOVA), Time × Group p = 0.0047; Group p = 0.0007). Sham vs. DREADD (Tone response, ****p < 0.0001, Bonferroni corrected). (**b'**) DREADD-induced silencing of BLA-projecting cholinergic neurons during training reduced BLA Fos immunoreactivity following recall. Representative BLA images from sham injected and CAV$_2$-DIO-hM4Di mice fixed and stained with anti-Fos antibodies (white) at 45–60 min following recall. Dotted line outlines the BLA (scale bar = 100 μm). (**b''**) The density of recall-activated BLA neurons under sham injected conditions vs. following selective inhibition of the BLA-projecting cholinergic neurons using CAV$_2$-hM4Di (Fos+). Fos+ cell density in BLA sham injected (black) vs. CAV$_2$-DIO-hM4Di.mcherry (red) (*n* = 7 mice/group, averaged from 22 sections sham vs. 28 sections hM4Di). Mann–Whitney test: *p = 0.0286. Lines represent median for each group. (**C**) BLA-projecting cholinergic neurons were silenced during recall (clozapine given ONLY 10 min prior to the recall). Freezing differed significantly between pre-tone vs. recall tone response for sham and DREADD groups (RM two-way analysis of variance (ANOVA), pre-tone vs. recall tone response, sham ****p < 0.0001; DREADD ***p = 0.0003). There was a significant effect of group, p = 0.0312. sham and DREADD groups were significantly different in their response to the recall tone, p = 0.0279. All multiple comparisons were Bonferroni corrected. (**c'**) hM4Di-induced silencing of BLA-projecting cholinergic neurons during recall reduced BLA Fos immunoreactivity following recall. BLA images following Fos immunostaining (scale bar = 100 μm). (**c''**) Fos+ cell density in BLA between sham injected (black) vs. CAV$_2$-DIO-hM4Di.mcherry (red) injected mice (*n* = 4–5 mice/group, averaged from 30 sections sham vs. 32 sections hM4Di). Mann–Whitney test: *p = 0.0317.

The online version of this article includes the following figure supplement(s) for figure 5:

**Figure supplement 1.** DREADD-induced silencing of basolateral amygdala (BLA)-projecting cholinergic neurons reduces threat induced activation of anterior BLA and CeC neurons.

**Figure supplement 2.** Role of nucleus basalis and posterior substantia innominata (NBM/SI$_p$) in associative threat memory recall.

resulted in ~50% of the mice being 'Low Responders' (*Figure 6—figure supplement 1B*, sham vs. cav.hM4Di[BLA] inhibition during recall). Thus, silencing BLA-projecting cholinergic neurons only during recall resulted in an all-or-none behavioral phenotype (50:50 chance of becoming a High or Low Responder).

## Cholinergic neurons activated during threat memory recall have altered intrinsic excitability

Changes in excitability of neurons have been consistently associated with the threat memory engram (*Zhang and Linden, 2003*; *Zhou et al., 2009*; *Cai et al., 2016*; *Rashid et al., 2016*; *Pignatelli et al., 2019*). We asked whether cholinergic neurons activated during memory recall differed in their intrinsic excitability compared to non-activated cholinergic neurons. To do this, we prepared acute brain slices from Fos-tTA/shGFP mice for electrophysiological recording of activated (Fos−GFP+) and non-activated (Fos−GFP−) NBM/SI$_p$ neurons two and a half hours after the recall session or from mice that remained in their home cage. Cholinergic identity was verified post-recording by single-cell reverse transcriptase polymerase chain reaction (scRT-PCR) of each recorded cell (*Figure 7A*).

Cholinergic neurons that were Fos+ following the recall session differed significantly from Fos− cholinergic neurons (*Figure 7B, C*) and from cholinergic neurons from home cage mice. Properties that showed significant differences included: action potential (AP) half-width, rheobase, and maximum firing rate (*Figure 7D*; half-width: HC vs. Fos−shGFP+ p = 0.0006, Fos−shGFP− vs. Fos−shGFP+ p = 0.021; *Figure 7E*; rheobase: Fos−shGFP− vs. Fos−shGFP+ p = 0.023; *Figure 7F*; max firing rate: HC vs. Fos−shGFP+ p = 0.003, Fos−shGFP− vs. Fos−shGFP+ p = 0.0034) as well as latency to fire (*Figure 7—figure supplement 1E*; latency: HC vs. Fos−shGFP+ p = 0.0062) and afterhyperpolarization (AHP) amplitude (*Figure 7—figure supplement 1F*, HC vs. Fos−shGFP+ p = 0.0041). Resting membrane potential, AP amplitude, AP threshold, and AHP half-width did not differ (*Figure 7—figure supplement 1A–D*).

We also compared the firing rate of cholinergic neurons in home cage mice with those expressing Fos two and a half hours after training or at longer intervals following recall (measured 2.5 hr (Fos− shGFP) and at 3 and 5 days (ADCD labeling during recall) after the recall session, *Figure 7—figure supplement 1G*). We found no differences in firing rate between home cage cholinergic neurons and cholinergic neurons that expressed Fos after training: that is, the change in firing rate was only seen in cholinergic neurons activated during recall. This increase in maximal firing rate seen after recall returned to baseline within 3–5 days (compared to recall D0, p < 0.05 for all).

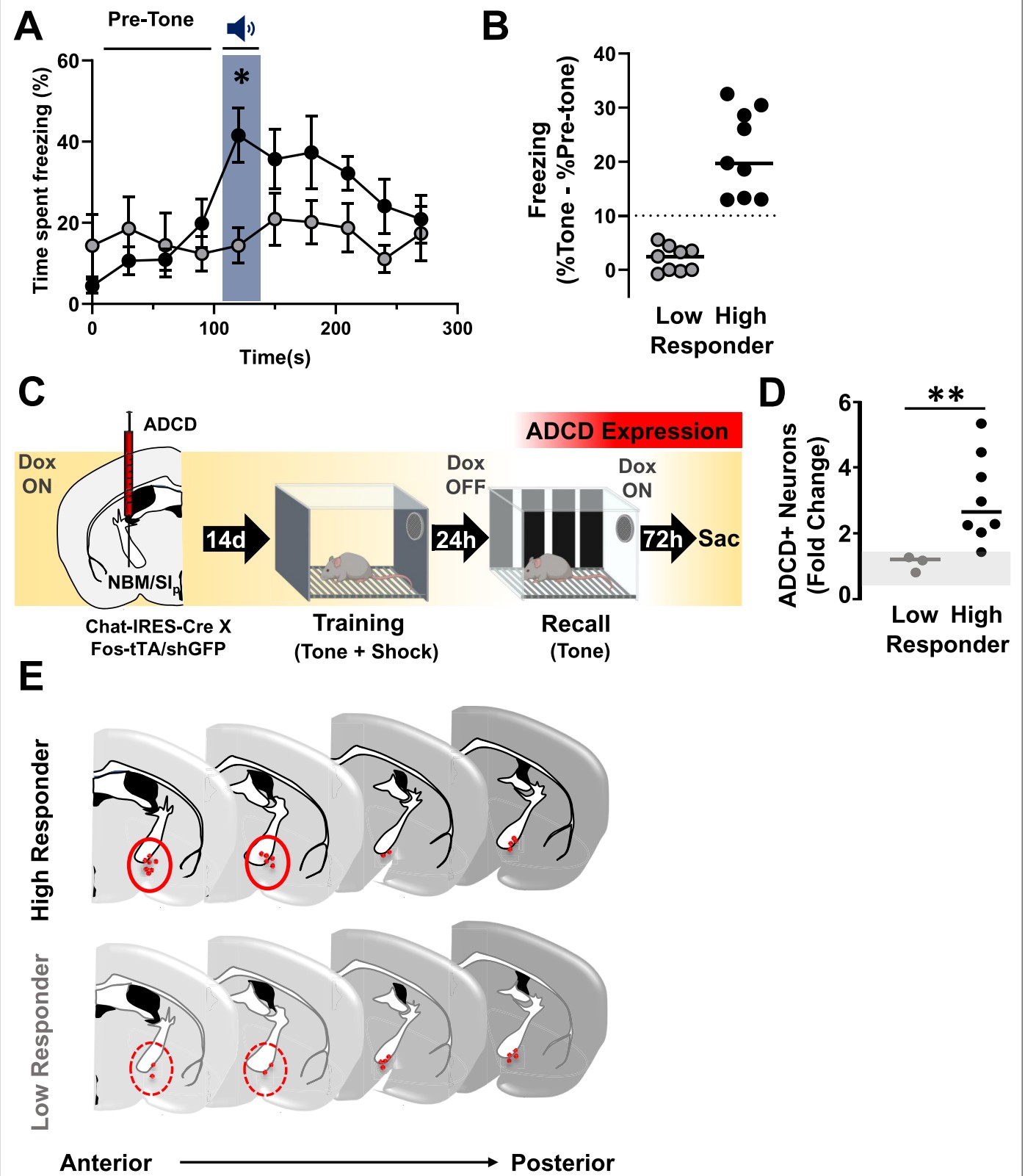

**Figure 6.** The extent of cholinergic neuronal activation in the anterior nucleus basalis and posterior substantia innominata (NBM/SI$_p$) co-varies with the behavioral performance during threat memory recall (see *Figure 6—figure supplements 1 and 2*). (**A**) Behavioral performance (freezing) from recall session showing High (black, *n* = 9) and Low (gray, *n* = 9) responding mice. High Responders show significantly higher freezing to recall tone whereas low responders do not (two-way repeated measures (RM) analysis of variance (ANOVA)). Interaction effect (Time × Group classification, p =

*Figure 6 continued on next page*

*Figure 6 continued*

0.0001; Time, p = 0.0042. High vs. Low Responder *p = 0.0454; Pre-tone vs. Tone: High Responder, p = 0.0016; Low Responder, p > 0.9999). All multiple comparisons were Bonferroni corrected. (**B**) Quantification of change in freezing responses during recall session in Low and High Responders (pre-tone to tone). Dotted line delineates 10% points change in freezing, which was set as criteria for separating the two populations (see Methods for rationale on stratification criteria, n = 9 Low Responder, n = 9 High Responder). (**C**) Mice injected in the NBM/SI$_p$ with ADCD-mCherry underwent training on Dox and recall off Dox to label recall-activated NBM/SI$_p$ cholinergic neurons (n = 11). (**D**) Quantification of change in number of cholinergic neurons activated (ADCD+) in low or High Responders relative to the home cage. The number of ADCD+ neurons differed significantly between Low and High Responders (Mann–Whitney test, **p = 0.01) (n = 3 Low Responder, n = 8 High Responder). Gray shading represents the range of fold-change in ADCD+ cells in individual home cage mice relative to the average of all home cage mice (n = 5) (Mann–Whitney test, home cage vs. Low Responder, p > 0.9999; home cage vs. High Responder, p = 0.0121). (**E**) Schematic showing anatomical distribution of ADCD-labeled NBM/SI$_p$ basal forebrain cholinergic neurons (BFCNs) activated during recall across the anterior (bregma ~−0.8 mm) to posterior (bregma ~−1.3 mm) extent of the NBM/SI$_p$ in High Responders (Top) vs. Low Responders (Bottom). Red circles highlight region of notable difference between High and Low responding mice.

The online version of this article includes the following figure supplement(s) for figure 6:

**Figure supplement 1.** Reactivation of training-activated cholinergic neurons scales with associative threat learning and with behavioral performance during memory recall.

**Figure supplement 2.** Recall-activated basal forebrain cholinergic neurons (BFCNs) in nucleus basalis and posterior substantia innominata (NBM/SI$_p$) of 'High Responder' mice are located at a basolateral amygdala (BLA)-projecting locus in the cholinergic basal forebrain.

## Distinct subsets of BLA-projecting cholinergic neurons differentially contribute to learned vs. innate threat processing

Given the importance of BFCNs in a learned threat paradigm, we next asked whether these cells participate in innate threat responses as well. We stimulated an innate threat response by exposing Fos-tTA/shGFP mice to predator odor (mountain lion urine; *Figure 8A*; *Blanchard and Blanchard, 1990*). Exposed mice increased active and passive defensive behaviors compared to mice exposed to a saline wetted pad, including freezing (*Figure 8A*, p = 0.028), avoidance (*Figure 8—figure supplement 1B*, left, p = 0.0012), and defensive digging (*Figure 8—figure supplement 1B*, right, p = 0.023).

We quantified the number of cholinergic neurons expressing Fos (Fos−shGFP+) after saline or predator odor exposure (*Figure 8B/Figure 8—figure supplement 1A*; Fos−shGFP+/ChAT+). The number of Fos−shGFP expressing cholinergic neurons was significantly elevated in the predator odor exposed group in the VP/SI$_a$ (*Figure 8B*, *Figure 8—figure supplement 1A* – middle row, p = 0.0023), but not NBM/SI$_p$ (*Figure 8—figure supplement 1A* – bottom row, p = 0.4441), or the hDB (*Figure 8—figure supplement 1A* – top row, p = 0.2465).

VP/SI$_a$ cholinergic neurons formed the second largest source of cholinergic input to the BLA in our retrograde mapping experiments (*Figure 5A*). Since VP/SI$_a$ cholinergic neurons were found to be activated during predator odor exposure, rather than NBM/SI$_p$ or hDB cholinergic neurons, we asked if the BLA-projecting pool of VP/SI$_a$ cholinergic neurons was activated by predator odor exposure. We injected the retrograde tracer FB into the BLA of Fos-tTA/shGFP mice and then exposed them to either saline (control) or predator odor (*Figure 8C*, left). FB labeled approximately 30% of ChAT-IR neurons located in the VP/SI$_a$ (data not shown). Nearly, the entire subset of BLA-projecting VP/SI$_a$ cholinergic neurons (median 94% ± Std.dev 12.5) were also GFP+ (*Figure 8C*, right).

To determine whether activity of these BLA-projecting cholinergic neurons was necessary for mice to freeze in response to predator odor, we used CAV$_2$-DIO-hM4Di to silence BLA-projecting cholinergic neurons. Silencing during predator odor exposure resulted in significantly less freezing compared to sham mice (*Figure 8D*, sham vs. cav.hM4Di$^{BLA}$ p = 0.019). Other measures of active avoidance of the predator odor were not significantly altered by silencing BLA-projecting cholinergic neurons (*Figure 8—figure supplement 1C*; avoidance p = 0.8485; defensive digging p = 0.0714). These data support the conclusion that activity of BLA-projecting cholinergic neurons is critical for normal freezing behavior in response to innate threat. Taken together, we find that distinct populations of BLA-projecting BFCNs are involved in associative threat learning and the response to innately threatening stimuli.

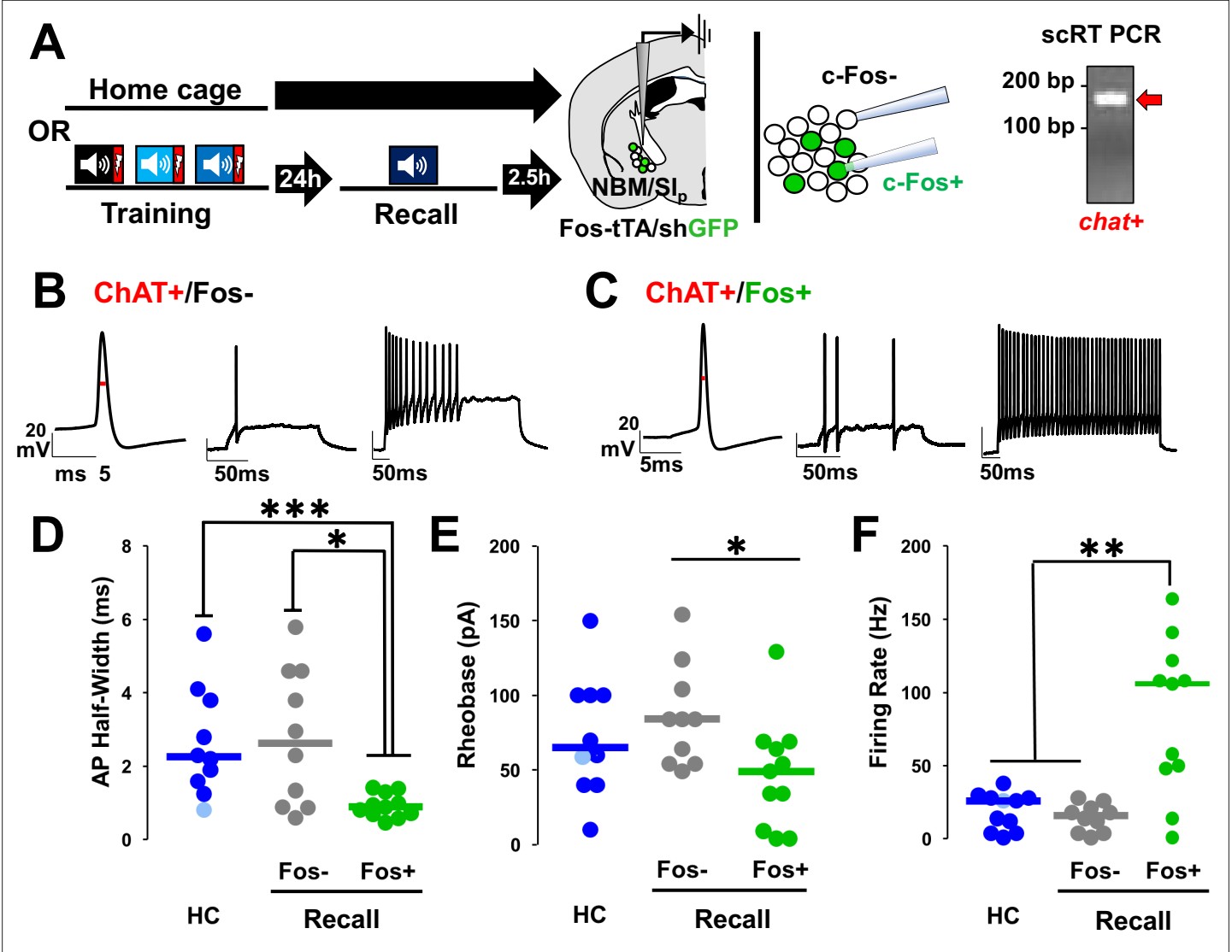

**Figure 7.** Nucleus basalis and posterior substantia innominata (NBM/SI_p) cholinergic neurons show increased intrinsic excitability following threat memory recall (see *Figure 7—figure supplement 1*). (**A**) Schematic of electrophysiological profiling of activated (Fos–shGFP+) vs. non-activated (Fos–shGFP−) neurons from mice following recall or in home cage mice, with post hoc identification of cholinergic identity by single-cell RT-PCR and evaluation of *chat* expression. (**B**) Representative traces following injection of current into a Fos–shGFP− NBM/SI_p cholinergic neuron (ChAT+/Fos−). Red line denotes action potential (AP) half-width measurement. (**C**) Representative traces following step current injection in Fos-shGFP+NBM/SI_p cholinergic neuron (ChAT+/Fos+). Red line denotes AP half-width measurement. (**D–F**) Population data (dot plot + line at median) for the electrophysiological properties of post hoc identified cholinergic neurons. Analyses assess passive and active membrane properties including AP (**D**) half-width, (**E**) rheobase, and (**F**) maximal firing rate in response to 200–500 ms depolarization from rest potential (−60 mV), from home cage (HC; n = 10–11 ChAT+ neurons from 10 to 11 mice) and following recall to tone alone (n = 10 ChAT+Fos–shGFP− neurons from 5 mice vs. n = 11 ChAT+Fos–shGFP+ neurons from 6 mice). (**D**) Kruskal–Wallis tests; AP half-width: p = 0.0054 (Dunn's Corrected p-values: HC vs. Fos–shGFP−: p = 0.8971, HC vs. Fos–shGFP+: ***p = 0.0006, Fos–shGFP− vs. Fos–shGFP+: *p = 0.0206). (**E**) Rheobase: KW = p = 0.05 (Dunn's Corrected p-values: HC vs. Fos–shGFP−: p = 0.6153, HC vs. Fos–shGFP+: p = 0.0938, Fos–shGFP− vs. Fos–shGFP+: *p = 0.0228). (**F**) Max firing rate: p = 0.0032 (Dunn's Corrected p-values: HC vs. Fos–shGFP−: p = 0.3206, HC vs. Fos–shGFP+: **p = 0.003, Fos–shGFP− vs. Fos–shGFP+: **p = 0.0034).

The online version of this article includes the following figure supplement(s) for figure 7:

**Figure supplement 1.** Latency and afterhyperpolarization (AHP) amplitudes significantly differed in recall-activated nucleus basalis and posterior substantia innominata (NBM/SI_p) cholinergic neurons.

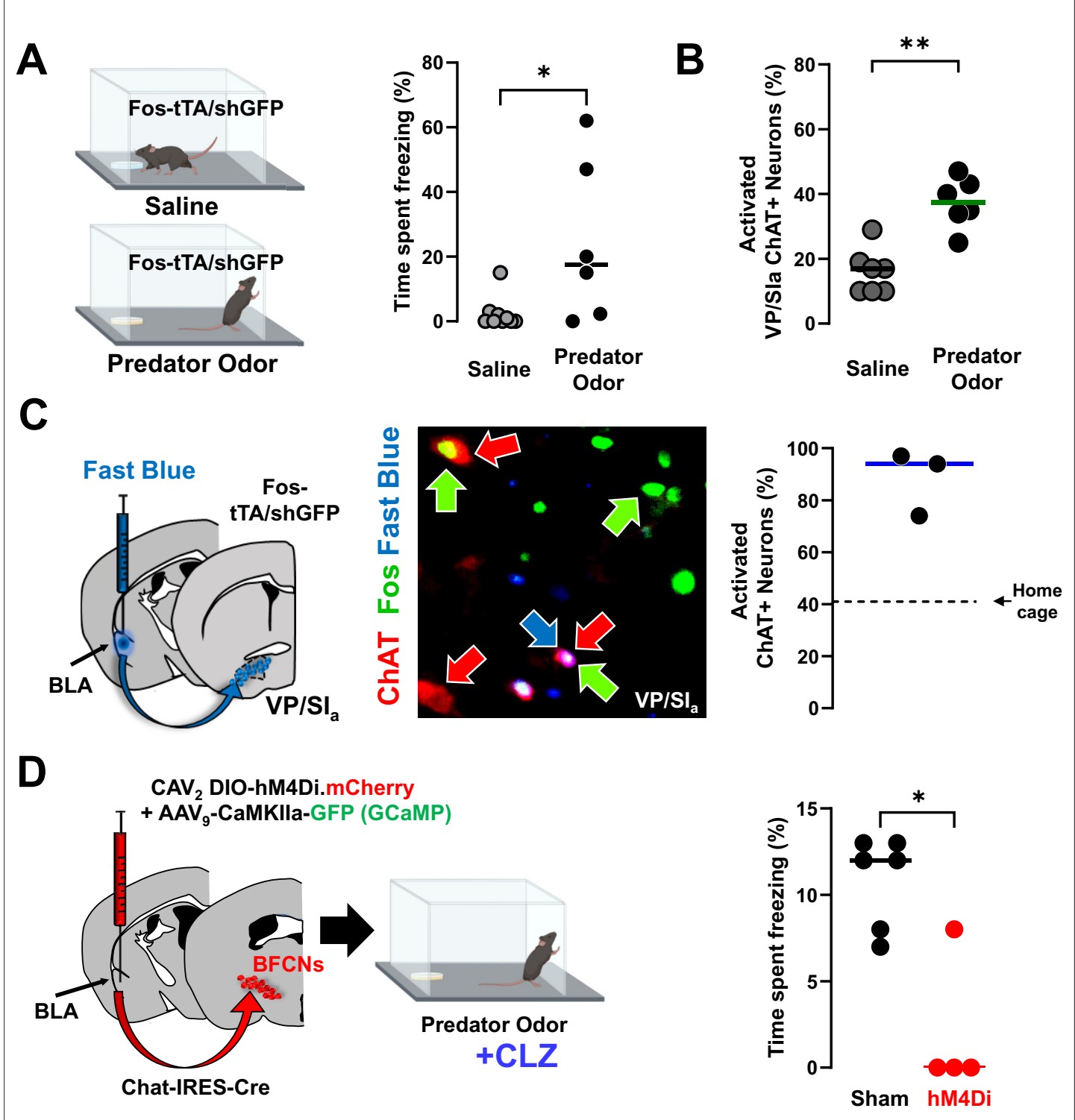

**Figure 8.** Distinct population of basolateral amygdala (BLA)-projecting cholinergic neurons contribute to innate threat processing (see *Figure 8—figure supplement 1*). (**A**) Fos-tTA/Fos–shGFP mice were placed in chambers containing a gauze pad spotted with either saline or with mountain lion urine (predator odor). Defensive behaviors were monitored for 5 min. Mice froze significantly more in the presence of predator odor than saline (Mann–Whitney, *p = 0.028). (**B**) Basal forebrain sections from the ventral pallidum (VP/SI$_a$) of Fos-tTA/shGFP mice were immunostained for ChAT and GFP 45 min following odor exposure. Predator odor-activated cholinergic neurons (GFP+/ChAT+) were quantified. Predator odor exposure increased the number of activated cholinergic neurons in the VP/SI$_a$ (Mann–Whitney: **p = 0.0023), *n* = 7 control and *n* = 6 odor exposed mice. (**C**) Fast Blue was injected into the BLA to retrogradely label BLA-projecting neurons 6 days prior to odor exposure. After exposure to predator odor sections from

*Figure 8 continued on next page*

Figure 8 continued

the basal forebrain were immunostained with antibodies recognizing ChAT and Fos and the numbers of activated cholinergic neurons were counted (ChAT+Fos+/total ChAT+). In the VP/SI$_a$ over 90% of BLA-projecting cholinergic neurons were activated (ChAT+ in red, Fos+ in green, Fast Blue in blue, $n$ = 3 mice). Dotted line indicates % of Fos+ cholinergic neurons in the home cage group in this experiment. (**D**) Chat-IRES-Cre mice injected in the BLA with a control virus (AAV$_9$-camk2a-GFP) alone (sham) or in combination with CAV$_2$-DIO-hM4Di were exposed to predator odor following injection with clozapine (CLZ). Freezing behavior was measured during a 5-min exposure (scatter plot, bar indicates mean; sham – black, hM4Di – red). Silencing BLA-projecting cholinergic neurons significantly blunted the freezing response (Mann–Whitney: *p = 0.019; sham: $n$ = 6; hM4Di: $n$ = 4 mice).

The online version of this article includes the following figure supplement(s) for figure 8:

**Figure supplement 1.** Predator odor exposure activates VP/SI$_a$ cholinergic neurons.

## Discussion

A small number of sparsely distributed cholinergic neurons in the basal forebrain provide extensive innervation to most of the brain. These cholinergic neurons and their network of axonal terminal fields play a critical role in modulating cognitive processes (*Ballinger et al., 2016*; *Záborszky et al., 2018*; *Ananth et al., 2023*).

To begin addressing whether the cholinergic system encodes stimulus-specific information, or whether it is generally recruited with salient experiences we monitored ACh release in the BLA during threat learning and retrieval. We anatomically mapped and electrophysiologically characterized behaviorally relevant BFCNs, and then investigated the contribution of different subsets of BFCNs to threat response behaviors. Taken together, our results demonstrate distinct populations of cholinergic neurons that are an integral part of encoding a learned threat memory or contribute to innate threat responses.

### Cholinergic modulation of associative threat learning

In the BLA, several molecular changes occur in response to learning conditioned stimulus-unconditioned stimulus (CS-US) associations, including new gene expression and protein synthesis (*Sears et al., 2014*). We used chemogenetics for projection-specific, cell-type-specific silencing of cholinergic neurons. We used CLZ activation of hM4Di, acting at cholinergic cell bodies and/or cholinergic terminals (*Krashes et al., 2011*; *Ray et al., 2011*; *Ferguson et al., 2013*; *Stachniak et al., 2014*; *Zhang et al., 2017*; *Jin et al., 2019*; *Nishioka et al., 2020*; *O'Neal et al., 2020*) to silence BLA-projecting BFCNs during training or during recall. This resulted in loss of freezing behavior as well as significantly reduced density of Fos expressing neurons in the BLA following recall (*Figure 5*). This reduction of Fos expression in the BLA indicates that cholinergic signaling in the BLA contributes to appropriate BLA engagement during the acquisition and recall of threat memory. While our experiments did not directly measure the BLA engram per se (i.e. activation–reactivation of the same neurons within the BLA), our data support the hypothesis that BLA-projecting cholinergic neurons play a critical role in the formation and/or activation of the BLA engram.

We have previously demonstrated that activation of presynaptic ACh receptors can induce sustained potentiation of glutamate release (*McGehee et al., 1995*; *Zhong et al., 2008*; *Jiang et al., 2013*; *Zhong et al., 2013*; *Zhong et al., 2015*; *Jiang et al., 2016*; *Zhong et al., 2017*). BLA neurons recruited during memory recall exhibit increased presynaptic glutamatergic activity (*Nonaka et al., 2014*). We further demonstrated that the increased glutamatergic transmission in BLA was dependent on presynaptic nicotinic acetylcholine receptors (nAChRs) located on glutamatergic terminals in the BLA, and that nAChR activation in the BLA was necessary for acquisition of conditioned threat memories (*Jiang et al., 2016*). Based on these findings, we propose that chemogenetic silencing of BLA-projecting cholinergic neurons during threat learning or during recall results in loss of Fos expression due to alterations in presynaptic glutamatergic transmission resulting in disruption to the formation and/or recruitment of the BLA engram.

### BFCNs 'learn' to respond to the conditioned stimulus

In this study, we used a genetically encoded ACh sensor (GRAB$_{ACH3.0}$) to monitor endogenous ACh release in the BLA during threat learning and recall. First, we found that foot shock rapidly and reliably

evoked ACh release, in line with previous observations (*Hangya et al., 2015*; *Jing et al., 2020*). When we examined responses to the tone (CS, *Figure 1—figure supplement 1*), we did not detect a significant increase in ACh in the BLA in response to a naive, unexpected tone. However, following conditioning, when mice were exposed to the conditioned tone in a novel environment 24 hr later, we observed robust ACh release in the BLA compared with the naive tone (*Figure 1D*). This enhancement of ACh release supports the notion that BLA-projecting BFCNs undergo physiological changes which allow robust responsiveness to previously naive sensory stimuli. When mice were exposed to tones in the absence of foot shocks and then exposed to the same tone 24 hr later, we did not detect increased ACh release in the BLA (*Figure 1—figure supplement 2*). Thus, plasticity of ACh release in the BLA in response to the tone requires pairing of the tone with a salient stimulus such as a foot shock.

## Changes in excitability of Fos+ cholinergic neurons

It has been proposed that alterations to synaptic weights and changes in ionic conductance resulting from learning-induced transcriptional programs allow for increased response fidelity during memory retrieval (*Yap and Greenberg, 2018*). To assess whether such changes occurred in recruited cholinergic neurons following memory retrieval, we recorded properties of neuronal excitability from activated NBM/SI$_p$ BFCNs (Fos+) and compared them with Fos− BFCNs recorded in the same brain slices (*Figure 7*). Recall-activated NBM/SI$_p$ cholinergic neurons showed increased excitability which lasted for at least several hours following threat memory retrieval, returning to baseline within days. This finding is in line with previous reports of learning-associated changes in electrical properties, which are found shortly after recall, but disappear at later time points despite the persistence of the learned behavior (*Moyer et al., 1996*; *Pignatelli et al., 2019*). Observed changes in the electrophysiological properties were not see in Fos-shGFP+ cholinergic neurons immediately following training. Thus, many of the changes in electrical properties we observed were specific to recall-activated cholinergic neurons. Within recall-activated cholinergic neurons we find several changes consistent with an increased excitability such as decreased AP half-width, decreased rheobase, and an increase in maximum firing rate. Common features of activated neurons previously reported include similar increases in firing rate, with reductions in adaptation, decreased duration of post-burst AHP, decreased AHP amplitude, and synaptic alterations (*Whitaker and Hope, 2018*).

## Differential contribution of distinct BLA-projecting BFCNs in learned vs. innate threat processing

Amygdala microcircuits play an important role in the regulation of active vs. passive avoidance behaviors (*Rickenbacher et al., 2017*; *Terburg et al., 2018*). Our finding that silencing cholinergic input to the BLA resulted in a selective loss of threat-motivated freezing behavior supports potential specificity of cholinergic modulation within BLA microcircuits for freezing, but not active, defensive behaviors. We found that BLA-projecting cholinergic neurons were necessary for freezing in response to a learned threat-associated cue (*Figure 5*), and for freezing in response to the innately threatening predator odor (*Figure 8*). Direct silencing of NBM/SI$_p$ cholinergic neurons attenuated learned threat induced freezing. Instead, predator odor activated BLA-projecting VP/SI$_p$ cholinergic neurons and resulted in a freezing response. Based on these data, we propose that distinct populations of BLA-projecting BFCNs control freezing in response to fundamentally distinct threatening situations (learned vs. innate). Additionally, we note that while silencing BLA-projecting BFCNs did reduce freezing in response to predator odor exposure, it did not alter avoidance of the odor pad indicating that threat detection was still intact in these mice.

## Memory encoding in neuromodulatory systems

Our study joins a growing literature demonstrating stimulus-encoding and rapid stimulus-contingent responses in various neuromodulatory neurons indicating that plasticity within subcortical modulatory circuits might represent a critical component of normal learning and memory recall. The BLA receives various modulatory inputs including dopamine (DA) from the ventral tegmental area (VTA) (*Tang et al., 2020*), noradrenaline (NA) from the locus coeruleus (LC) (*Uematsu et al., 2017*), and ACh from the basal forebrain. Including our present study, all three of these modulatory systems have been shown to be engaged during associative threat learning and retrieval. Each modulatory

system seems to respond rapidly and robustly to aversive stimuli like mild electrical shocks, and activity within these systems during conditioning (i.e. during CS–US pairing) is critical for generation of freezing behavior during memory recall (*Uematsu et al., 2017*; *Tang et al., 2020*). VTA dopaminergic neurons have also been shown to display plasticity in tone responsiveness such that a naive tone does not result in significant firing of DA neurons (*Tang et al., 2020*). However, following three pairings of the tone with shocks, VTA DA neurons begin responding to tone presentations with millisecond latencies, a response that is sustained the following day during memory retrieval. A majority of the shock-responsive DA neurons were also found to acquire tone responsiveness following pairing, a finding replicated within the cholinergic system in our study. While shock rapidly activates LC NA neurons, conditioned tone-related responses in these neurons seems to be slow, occurring on average several seconds following tone presentation (*Uematsu et al., 2017*). How signaling by these different modulators interacts in the BLA and informs plasticity of BLA neurons is an intriguing question.

In addition to these modulators, peptides such as oxytocin have also been shown to participate in threat memory formation. A recent study demonstrated presence of a threat memory engram within the hypothalamic oxytocinergic projection to the amygdala (*Hasan et al., 2019*). Interestingly, upon conditioning these neurons demonstrate a transmitter preference switch, releasing glutamate in the amygdala. Thus, subcortical neuromodulatory and peptidergic systems might display unique mechanisms of engram-related biophysical changes that have not been found in traditionally studied systems.

We demonstrate at least two populations of BLA-projecting cholinergic neurons that are engaged in learned vs. innate threat responses. Differences in function of other BLA-projecting BFCNs (NBM vs. HDB) in threat memory formation vs. extinction were recently demonstrated (*Hasan et al., 2019*; *Crimmins et al., 2023*), further highlighting that effects of ACh release in the BLA are highly specific to which axons release the ACh, despite the dense overlapping terminal fields from different BFCN populations within the BLA. Similar heterogeneity of responses has also been found in the dopaminergic and noradrenergic systems (*Azcorra et al., 2023*). It is possible that single-cell transcriptomic analyses of the cholinergic basal forebrain may provide insight into the functional heterogeneity observed in our study.

## Is there a cholinergic component in the associative threat memory engram?

Studies examining mechanisms of learning and memory in recent years have revived Semon's theory on memory engrams: learning must result in lasting biophysical changes that form the substrate for retrieval of the learned experience (*Semon, 1921*; *Tonegawa et al., 2015*). Josselyn and Tonegawa have recently updated the definition of engram cells, requiring that these be activated by learning, modified by learning, and reactivated by subsequent presentation of the recall-inducing stimuli, resulting in memory retrieval (*Josselyn and Tonegawa, 2020*). NBM/SI$_p$ BFCNs investigated in this study indeed fulfil these criteria as they are activated by learning, show induction of Fos and altered physiological properties with recall, are reactivated by recall, and the reactivation of previously, training-activated BFCNs was necessary for recall behavior.

Multiple studies have used threat and reward learning paradigms in rodents to examine allocation of neurons to memory engrams. These studies have looked for these engram cells in regions such as cortex, amygdala, and hippocampus focusing on glutamatergic pyramidal neurons (*Josselyn et al., 2015*). However, recent work has demonstrated that memory engrams are distributed across brainwide networks, and that reactivation of a multi-region engram more closely recapitulates natural recall behavior (*Roy et al., 2022*).

In addition to the BLA, cholinergic neurons in the NBM/SI$_p$ region project to various limbic and sensory regions such as the lateral orbital cortex, cingulate cortex, somatosensory cortex, and mediodorsal thalamus (*Ananth et al., 2023*). This raises the interesting possibility that the cholinergic signaling modulates various nodes of the threat memory engram circuit in conjunction with the amygdala, allowing for coordinated retrieval of engrams across distributed networks. Such coordinated activation of distributed engrams has been recently demonstrated to more closely recapitulate natural memory retrieval (*Roy et al., 2022*). Furthermore, functionally related regions have been shown to receive their cholinergic input from the same cholinergic nucleus (*Zaborszky et al., 2015*). We propose

that engram-enrolled cholinergic neurons bind distributed engrams to encode stimulus-convergent, efficient memory retrieval.

# Materials and methods

## Resource availibility

### Lead contact

Further information and requests for resources and reagents should be directed to and will be fulfilled by Lead Contact, Dr. David Talmage (david.talmage@nih.gov).

### Materials availability

Plasmids generated in this study have been deposited to Addgene and will be available upon publication under Talmage Lab.

## Experimental model and subject details

Adult (3–6 month) male and female Chat-IRES-Cre (B6;129S6-Chattm2(cre)Lowl/J), Jax stock number: 006410 (*Rossi et al., 2011*), Fos-tTA, Fos−shGFP (TetTag, Jax stock number: 018306, referred to as Fos-tTA/shGFP or Fos-shGFP), and Chat-IRES-Cre × Fos-tTA/shGFP mice were used. Mice within each cage were randomly assigned to experimental and control conditions. In all electrophysiology experiments, hemizygous Fos-tTA/shGFP mice on a C57BL/6 background were used. Mice were housed in a 12-hr light/dark cycle environment that was both temperature and humidity controlled. Mice had free access to food and water. All animal care and experimental procedures were approved by the Animal Care and Use Committees (ACUC) of the National Institute of Neurological Disorders & Stroke (NINDS) (Protocol #1531), SUNY Research Foundation at Stony Brook University (Protocol #1618), and Yale University (Protocol #2019-07895).

## Method details

### Viral construct

#### Construction of the ADCD probe

All cloning unless otherwise specified was performed using In-Fusion HD (Clontech). 'mCherry-P2A' was amplified using Phusion High-Fidelity DNA Polymerase (NEB) from pV2SGE (obtained as a gift from Dr. Shaoyu Ge Stony Brook University). 'oChIEF-LoxP-Lox2272' was amplified from pV2.2 (synthesized gene block from IDT). The two fragments were cloned into pAAV-WPRE linearized by BamHI. The resulting plasmid was linearized by Pml I. '7xTetO-LoxP-Lox2272-tTAH100Y.SV40' was amplified from pV2.1 (synthesized gene block from IDT) and cloned into the Pml I site. The final plasmid was packaged into AAV$_9$ viral particles. Viral packaging was performed by the University of Pennsylvania Vector Core.

#### Note re: ADCD expression in BLA neurons in the presence of doxycycline

As shown in *Figure 2—figure supplement 1C*, we noted 'leaky' expression of ADCD-mCherry in the presence of doxycycline, in the BLA of Fos-tTA mice when co-injected with a Cre expression vector expressed from a camk2a promoter. Co-injection of camk2a-Cre and ADCD-mCherry into cortex and hippocampus of wild-type (C57) mice was also found to result in 'leaky' expression despite the absence of genetically encoded tTA. Injection of ADCD-mCherry in hippocampus of PV-Cre mice did not result in expression similar to injection in Chat-IRES-Cre mice (*Figure 2—figure supplement 1A*, bottom). These findings underscore the importance of performing the appropriate controls when using these vectors in vivo.

#### Construction of the ADCD-DREADD probe

'BglII-hM4Di.mCherry-Ascl' was amplified using CloneAmpTM HiFi PCR Premix (Takara) from pAAV-hSyn-DIO-hM4D(Gi)-mCherry (*Krashes et al., 2011*) (gift from Dr.Bryan Roth; Addgene plasmid # 44362; http://n2t.net/addgene:44362; RRID:Addgene_44362). A backbone with TRE and Lox sites was ligated with 'BglII-hM4Di.mCherry-Ascl' using T4 DNA Ligase (NEB). The final plasmid was

packaged into AAV$_9$ viral particles. Viral packaging was performed by the University of North Carolina Vector Core.

## Stereotaxic surgery and viral delivery

Three- to four-month-old ChAT-IRES-Cre mice were anesthetized and stereotaxically injected bilaterally. Coordinates were calculated based on the Paxinos Mouse Brain Atlas (Franklin, K & *Franklin and Paxinos, 1997*): BLA (−1.4 mm A/P, ±3.5 mm M/L, −4.8 mm D/V) and NBM (−0.7 mm A/P, ±1.7 mm M/L, −4 mm D/V).

## Tracers

3% wt/vol solution of FB (17740-1, Polysciences Inc) was prepared in sterile milliQ water. ~0.2 μl of 3% FB was injected into the BLA bilaterally of Fos-GFP or Chat-IRES-Cre × Fos-tTA/shGFP mice. Mice were euthanized 7 days following injection.

## Behavioral testing and analysis

### Threat conditioning

All training and assessments were completed with experimenter blind to condition. Both training and recall sessions were analyzed using FreezeFrame v.3 (see below).

### Habituation

All mice were handled for a minimum of 5 min daily for three consecutive days before behavioral training began. For DREADD experiments, all mice were additionally habituated to restraint and injection with 100 μl saline administered i.p. daily.

### Training

On training day, all chambers were cleaned with 70% ethanol. Mice were placed into the behavioral chamber for a 10-min session which consisted of 3 min of habituation, followed by three tone–shock pairings (30 s 80 dB, 5 kHz tone, co-terminated with a 2-s 0.7-mA foot shock with a 1.5-min interval between each pairing), and finally 2 min of exploration. For DREADD experiments, mice were given 0.1 mg/kg CLZ (administered i.p.) (Sigma-Aldrich) 10 min prior to being placed in the chamber.

### Recall

Recall session took place 24–72 hr after completion of the training. To specifically test the response to tone-cued recall, the contextual features of the chambers were altered including texture of the floor, color of the walls, and scent of cleaner (mild lemongrass citrus-based solution). Mice were placed in the behavioral chamber for another 5 min session during which a single tone was delivered (30 s 80 dB 5 kHz tone) 2 min after being placed in the chamber. No shock was administered.

### Analysis

Percent time spent freezing was quantified using FreezeFrame v.3 (Actimetrics). Bout duration (defined as minimum required duration when animal is frozen) was set to 1 s, and threshold was manually defined as highest motion index with no movement other than breathing. Percent time spent freezing (defined as periods of no movement) was quantified across the 10 min session in bins of 30 s. The following periods were defined for statistical analysis: Baseline (average of all bins prior to tone onset) and Tone response (average of all bins following tone onset).

High Responders were defined as those mice that exhibited at least a 10 percentage point increase in % time spent freezing in the 30 s bin during the tone from the average of the pre-tone period (e.g. Pre-tone freezing 10% to tone-induced freezing of ≥20%). All other mice were considered Low responders. Prior to any behavioral manipulation, mice showed up to 10% (of total time in given time bin) freezing indicating this level of freezing to be non-associative (potentially related to novelty or generalized fear). This criterion was found to give statistically significant difference between pre-tone vs. tone only for high responders and not for low responders providing further validity to the delineation of the Low and High Responder groups.

Analysis of population composition of High and Low responders (*Figure 6—figure supplement 1*) was performed within experiment. Cross-experiment comparisons for population composition of

High and Low responders were not possible due to differences in conditions and variability within and between cohorts.

### Engram labeling

Mice were placed on doxycycline hyclate-containing chow (Cat# TD.08541 Envigo) at least 2 days prior to injection of activity-dependent viral markers. Threat conditioning was performed as mentioned above. During doxycycline withdrawal, mice were transferred to a clean cage to prevent mice from eating dox food that was dragged into the cage or buried in the bedding. To minimize stress, some bedding containing fecal pellets and urine, and nest from the old cage were transferred to the new cage.

### Predator odor exposure

#### Habituation

All mice were habituated to restraint and injection with 100 µl saline administered i.p. daily for 3 days prior to behavioral testing for DREADD experiments. On exposure day, mice were transported to the lab several hours prior to exposure and habituated to the room and ambient sounds.

#### Exposure

For exposure to predator odors, a vented mouse cage (L 13in × W 7.5in × H 5.5in) with corncob bedding (EnviroDri) was placed in a designated location in a laminar flow hood with overhead fluorescent lighting. Mt. Lion Pee (Maine outdoor solutions LLC) was obtained from https://predator-peestore.com/ and stored at 4°C. 200 µl of urine was pipetted onto a 3in × 3in 12 ply gauze pad (Cat#6312, Dukal corp.) placed in a polystyrene Petri dish (VWR) at the vented end of the cage. Mice were placed into the cage in the end away from the odor and the cage was covered using a clear plexiglass barrier. Mice were exposed for 5 min and the session was filmed using an overhead digital camcorder (Sony). Following exposure, mice were returned to their home cage or a holding cage in the case of multiple housed mice to prevent any odor transfer. Control mice were exposed to 0.9% saline. For DREADD experiments, mice were given 0.1 mg/kg CLZ (administered i.p.; Sigma-Aldrich) 15 min prior to being placed in the chamber.

#### Analysis

behavior was manually scored using Jwatcher (v0.9). Defensive digging was defined as vigorous digging performed by the mice using their snout, flinging bedding up and away from the animal. Freezing was defined as immobility without any obvious motion besides breathing. Cloth contacts were defined as front paw touches to the odor pad.

### Fiber photometry

#### Acquisition

Fiber photometry recordings were made using a Doric Lenses 1-site Fiber Photometry System. Signal was recorded using Doric Neuroscience Studio (V 5.3.3.4) via the Lock-In demodulation mode with sampling rate of 12.0 kS/s. Data were downsampled by a factor of 10 and saved as a comma-separated file. For details on connection of the setup refer to *Crouse et al., 2020*.

#### Analysis

Preprocessing of the raw data was performed using a MATLAB script provided by Doric. The baseline fluorescence ($F_0$) was calculated using a least mean squares regression over the duration of the recording session. The change in fluorescence for a given timepoint ($\Delta F$) was calculated as the difference between it and $F_0$, divided by $F_0$, and multiplied by 100 to yield % $\Delta F/F_0$. The % $\Delta F/F_0$ was calculated independently for both the signal (465 nm) and reference (405 nm) channels and a final 'corrected % $\Delta F/F_0$' was obtained by subtracting the reference % $\Delta F/F_0$ from the signal % $\Delta F/F_0$ at each timepoint. The corrected % $\Delta F/F_0$ was z-scored to give the final '$Z$ % $\Delta F/F_0$' reported. Area under the curve was calculated for 1-s duration before (baseline) and 1 s after tone onset. The average of all the baseline periods within each analysis was used as the baseline reading for the AUC analysis.

## Electrophysiology

### Brain slice preparation

For slice physiology, mice were anesthetized and transcardially perfused with cutting solution (sucrose 248 mM, KCl 2 mM, MgSO$_4$ 3 mM, KH$_2$PO$_4$ 1.25 mM, NaHCO$_3$ 26 mM, glucose 10 mM, sodium ascorbate 0.4 mM and sodium pyruvate 1 mM, bubbled with 95% O$_2$ and 5% CO$_2$) at 40°C. The brain was then rapidly removed and sliced, coronally, at 300 µM in oxygenated cutting solution at 40°C. Prior to recording, slices were incubated in oxygenated incubation solution (sucrose 110 mM, NaCl 60 mM, KCl 2.5 mM, MgCl$_2$ 7 mM, NaH$_2$PO$_4$ 1.25 mM, NaHCO$_3$ 25 mM, CaCl$_2$ 0.5 mM, MgCl$_2$ 2 mM, glucose 25 mM, sodium ascorbate 1.3 mM, and sodium pyruvate 0.6 mM) at room temperature.

### Electrophysiological recording

During recording, slices were superfused with oxygenated artificial cerebral spinal fluid (*Jiang et al., 2016*). Fos+ neurons were identified by GFP expression. Signals were recording using patch electrodes between 4 and 6 MΩ, a MultiClamp 700B amplifier, and pClamp10 software. Pipette internal solution was as follows: 125 mM K-gluconate, 3 mM KCl, 1 mM MgCl$_2$, 10 mM 4-(2-hydroxyethyl)-1-p iperazineethanesulfonic acid (HEPES), 0.2 mM CaCl$_2$, 0.1 mM ethylene glycol tetraacetic acid (EGTA), 2 mM MgATP, and 0.2 mM NaGTP (pH = 7.3). Following recording, cytoplasm was harvested via aspiration for cell-type identification using single-cell RT-PCR. Ten to twelve basic electrical properties were determined and defined as previously described (*López-Hernández et al., 2017*). Recordings were excluded if they did not meet the following criteria: (1) membrane potential less than or equal to −50 mV, (2) input resistance between 100 and 300 MΩ, (3) series resistance <10 MΩ that was unchanged throughout the recording, and (4) firing a 45 mV AP at rheobase.

## Single-cell reverse transcription-PCR

Single-cell samples were pressure ejected into a fresh RT buffer prep (Applied biosystems). Samples were sonicated in a total volume of 20 µl at 40°C for 10 min before addition of RT enzyme mix (Applied Biosystem). Tubes were incubated at 37°C for 60 min and then 95°C for 5 min. Two rounds of amplification (30 cycles each) were done for the detection of Chat transcripts. For the first round of amplification (reaction volume 25 µl) included 2× mastermix, sterile water, 0.2 mM of each primer, 1 ml of cDNA sample. For the second amplification, the reaction included 1 µl of the previous (first-round) PCR product, 2× mastermix, sterile water, and 0.2 mM of each primer. Whole brain cDNA was run in parallel with the single-cell samples. After amplification, the PCR products (159 bp) were analyzed on 3% gels.

## Immunohistochemistry

Following perfusion, brains were fixed overnight at 4°C in 4% Paraformaldehyde (PFA) (in 1× phosphate-buffered saline (PBS)) and were then transferred to a 30% sucrose solution (in 1× PBS). Brains were flash frozen in OCT Compound (Tissue Tek) and stored at −80°C until cryosectioning. 50 µm cryosections were mounted onto Superfrost slides (Fisher Scientific) in sets of 3 and allowed to dry overnight at room temperature. Sections were blocked overnight at 4°C in a PBS solution containing 0.3% Triton X-100 and 3% normal donkey serum and then incubated with primary antibody in a PBS-T solution (0.1% Triton X-100 and 1% normal donkey serum), overnight (24 hr at 4°C). The next day, sections were rinsed in PBS-T and incubated in secondary antibody for 2 hr at room temperature in PBS-T along with NeuroTrace-435 (Invitrogen). Sections were treated with an autofluorescence eliminator reagent (EMD Millipore) according to the manufacturer's guidelines and mounted in Fluoromount-G (Southern Biotech). Details regarding antibodies can be found in the Key Resources Table (KRT).

## Quantification and statistical analysis

### Imaging and analysis

All imaging was conducted on an Olympus wide-field slide-scanner microscope at 20× magnification (VS-120 and VS-200 systems, Z-step = 3 µm). Images were processed using the cell counter plugin on ImageJ. For Fos+ cell counts in the amygdala, only neurons (Nissl/ Neurotrace positive) with nuclear Fos stain were counted. The amygdala was identified, and a region of interest (ROI) defined using

ROI manager in ImageJ. Total area of the ROI was measured and noted. Fluorescence threshold was set to eliminate background fluorescence in ImageJ (defined as hazy background signal detected in space between neurons and white matter). This eliminated non-specific fluorescence and out of focus signals. Fos+ nuclei were then counted using the cell counter plugin.

For ADCD cell counts, mCherry+ neurons at the NBM/SI injection site were counted. NBM was consistently identified as the cluster of cholinergic cell bodies at the base of the internal capsule in the Globus Pallidus and the SI as the area located directly ventral to the GP as denoted by the Paxinos Mouse Brain Atlas (3rd Edition). 100% of the analyzed area of every third brain section was counted (~150 μm apart). Since the NBM/SI regions lack defined boundaries, we present the data as cell counts as opposed to cell density.

For Fos analysis in the BLA, Fos+ cells were counted in the area enclosed within the external and amygdalar capsules. Since the shape of the BLA changes along the anterior–posterior axis, Fos+ cell counts were normalized to the area enclosed within the external and amygdalar capsules and presented as density of Fos+ cells.

## Statistical analysis

Statistical analyses were done using GraphPad Prism (GraphPad Software Inc, San Diego, CA, USA), Sigmaplot 12.5 (Systat Software, Inc, San Jose, CA, USA), and OriginPro 9.1 (Origin Lab Corporation, Northampton, MA, USA). Normality of the data was assessed using Shapiro–Wilk and Smirnov–Kolmogorov tests. Data that were not normally distributed according to both normality tests, were analyzed using appropriate non-parametric tests. Detailed information on statistical tests used, p-values, and sample sizes, and other descriptive statistics can be found in the text (figure legends) and/or in the statistical reporting table (*Supplementary file 1*). Sample sizes for behavior experiments were determined using a power calculation based on effect sizes in pilot experiments with power set to 0.8.

Parametric tests used: Repeated measures (RM) one-way analysis of variance (ANOVA), RM two-way ANOVA, Welch's ANOVA, paired *t*-test (two-tailed), Welch's *t*-test.

Non-parametric tests used: Mann–Whitney test, Wilcoxon matched-pairs signed rank test, Kruskal–Wallis test, Friedman test.

p-value criteria: *$p \leq 0.05$, **$p \leq 0.01$, ***$p \leq 0.001$, ****$p \leq 0.0001$.

## Acknowledgements

This work was supported by the Intramural Research Program of NINDS. This work was also supported by DA14241, DA037566, and MH077681 to MRP, and early phases by NS022061 and MH109104 to LWR and DAT. RBC was supported by a NINDS Training Grant (T32) NS007224. We thank Dr. Shaoyu Ge (Stony Brook University, NY) for providing reagents and insightful discussions aiding in the conceptualization of the project. We also thank Drs Josh Dubnau and Qiaojie Xiong (Stony Brook University, NY) for providing feedback and discussions on experiments presented in this manuscript. We thank Wendy Akmentin, Dr. Li Bai, and Taylor Muir for expert technical assistance in data curation. Figure schematics were created with BioRender.com.

## Additional information

### Funding

| Funder | Grant reference number | Author |
|---|---|---|
| National Institute of Neurological Disorders and Stroke | 1ZIANS009424 | David A Talmage |
| National Institute of Neurological Disorders and Stroke | 1ZIANS009416 | Lorna W Role |
| National Institute of Neurological Disorders and Stroke | NS22061 | David A Talmage |

| Funder | Grant reference number | Author |
| --- | --- | --- |
| National Institute of Mental Health | U01-MH109104 | David A Talmage |
| National Institute of Mental Health | MH077681 | Marina R Picciotto |
| National Institute on Drug Abuse | DA14241 | Marina R Picciotto |
| National Institute of Neurological Disorders and Stroke | NS007224 | Richard Crouse |
| National Institute of Neurological Disorders and Stroke | 1ZIANS009422 | Lorna W Role |
| National Institute on Drug Abuse | DA037566 | Marina R Picciotto |

The funders had no role in study design, data collection, and interpretation, or the decision to submit the work for publication.

## Author contributions

Prithviraj Rajebhosale, Data curation, Formal analysis, Validation, Investigation, Visualization, Methodology, Writing - original draft, Writing – review and editing; Mala R Ananth, Data curation, Software, Formal analysis, Validation, Investigation, Visualization, Methodology, Writing - original draft, Writing – review and editing; Ronald Kim, Data curation, Investigation, Methodology; Richard Crouse, Data curation, Software, Formal analysis, Validation, Investigation, Methodology, Writing – review and editing; Li Jiang, Chongbo Zhong, Christian Arty, Shaohua Wang, Alice Jone, Niraj S Desai, Investigation; Gretchen López-Hernández, Investigation, Methodology; Yulong Li, Resources, Methodology; Marina R Picciotto, Supervision, Funding acquisition, Methodology, Writing – review and editing; Lorna W Role, David A Talmage, Conceptualization, Supervision, Funding acquisition, Methodology, Project administration, Writing – review and editing

## Author ORCIDs

Prithviraj Rajebhosale (ID) http://orcid.org/0000-0001-9893-3025
Richard Crouse (ID) https://orcid.org/0000-0002-9509-9263
Marina R Picciotto (ID) http://orcid.org/0000-0002-4404-1280
Lorna W Role (ID) https://orcid.org/0000-0001-5851-212X
David A Talmage (ID) http://orcid.org/0000-0003-4627-3007

## Ethics

All animal care and experimental procedures were approved by the Animal Care and Use Committees (ACUC) of the National Institute of Neurological Disorders & Stroke (NINDS) (Protocol #1531), SUNY Research Foundation at Stony Brook University (Protocol #1618), and Yale University (Protocol #2019-07895).

## Decision letter and Author response

Decision letter https://doi.org/10.7554/eLife.86581.sa1
Author response https://doi.org/10.7554/eLife.86581.sa2

# Additional files

## Supplementary files

• Supplementary file 1. Statistical reporting table. This file contains exact sample sizes for each group, group median, 95% confidence interval of the median, actual confidence interval, statistical test used, p-values, and test statistic for each of the reported plots in the manuscript.

• MDAR checklist

## Data availability

Source data for the fiber photometry experiments presented in Figure 1 and supplements are provided as individual source data files. Code for fiber photometry data was previously published in *Crouse et al., 2020*.

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

# Appendix 1

## Appendix 1—key resources table

| Reagent type (species) or resource | Designation | Source or reference | Identifiers | Additional information |
|---|---|---|---|---|
| Strain, strain background (*Mus musculus*) | Chat-IRES-Cre | The Jackson Laboratory | B6;129S6-Chattm2(cre) Lowl/J | Stock number: 006410 |
| Strain, strain background (*Mus musculus*) | Fos-tTA, Fos−shGFP | The Jackson Laboratory | TetTag | Stock number: 018306 |
| Strain, strain background (*Escherichia coli*) | Stellar Competent Cells, HST08 | Takara | Cat#636766 | |
| Genetic reagent (*AAV9*) | AAV9-camk2a-GCaMP6f-WPRE-SV40 | Penn Vector Core | | |
| Genetic reagent (*AAV9*) | AAV9-DIO-eCFP | This paper, Vector Biolabs | | Custom made |
| Genetic reagent (*AAV9*) | AAV9-hSyn-GACh4.3 | Vigene Biosciences Inc | | |
| Genetic reagent (*AAV8*) | AAV8-DIO-hM4Di-mCherry | Addgene | Cat#44362 | |
| Genetic reagent (*AAV9*) | AAV9-TRE-DIO-oChIEF-mCherry-P2A-tTAH100Y.SV40 | This paper | plasmid DNA | See Methods and *Figure 2—figure supplement 1*, can be obtained from Talmage lab. |
| Genetic reagent (*AAV9*) | AAV9-TRE-DIO-hM4Di-mCherry | This paper | Cat#169415 | Deposited to Addgene, see methods and *Figure 2—figure supplement 1*, can be obtained from Talmage lab. |
| Genetic reagent (*CAV2*) | CAV2-DIO-hM4Di-mCherry | Dr. EJ Kremer, Institut de Génétique Moléculaire de Montpellier, France | | |
| Antibody | anti-ChAT (Goat polyclonal) | Millipore | Cat# AB144P; RRID:AB_2079751 | IHC (1:500) |
| Antibody | anti-GFP (Rabbit polyclonal) | Thermo Fisher Scientific | Cat#: A-11122; RRID:AB_221569 | IHC (1:1000) |
| Antibody | anti-GFP (Rabbit polyclonal) | Abcam | Cat#: ab13970; RRID:AB_300798 | IHC (1:500) |
| Antibody | anti-mCherry (Mouse monoclonal) | Takara | Cat#: 632543; RRID:AB_2307319 | IHC (1:500) |
| Antibody | anti-DsRed (Rabbit polyclonal) | Takara | Cat#: 632496; RRID:AB_10013483 | IHC (1:500) |
| Antibody | anti-c-Fos (Rabbit polyclonal) | Synaptic Systems | Cat#: 226003; RRID:AB_2231974 | IHC (1:500) |
| Antibody | anti-Rabbit IgG (H+L)-AlexaFluor 488 (Donkey polyclonal) | Thermo Fisher | Cat#: A32790; RRID:AB_2762833 | IHC (1:1000) |
| Antibody | anti-Rabbit IgG (H+L)-Rhodamine Red-X (Donkey polyclonal) | Jackson Immunoresearch | Cat#: 711-295-152; RRID:AB_2340613 | IHC (1:1000) |
| Antibody | anti-Goat IgG (H+L)-AlexaFluor 594 (Donkey polyclonal) | Thermo Fisher | Cat#: A-11058; RRID:AB_142540 | IHC (1:1000) |

*Appendix 1 Continued on next page*

*Appendix 1 Continued*

| Reagent type (species) or resource | Designation | Source or reference | Identifiers | Additional information |
|---|---|---|---|---|
| Antibody | anti-Chicken IgY-Cy2 (Donkey polyclonal) | Gift from Dr.Shaoyu Ge, Stony Brook University NY | | IHC (1:1000) |
| Chemical compound | NeuroTrace 435/455 Blue Fluorescent Nissl Stain | Thermo Fisher | Cat#: N21479 | IHC (1:500) |
| Recombinant DNA reagent | pAAV-hSyn-DIO-hM4D(Gi)-mCherry (plasmid) | Addgene | Cat#44362 | |
| Recombinant DNA reagent | pV2SGE (plasmid) | This paper | Gift from Dr.Shaoyu Ge, Stony Brook University NY | Used in the construction of reagent #25 |
| Recombinant DNA reagent | pAAV-TRE-DIO-oChIEF-mCherry-P2A-tTAH100Y.SV40 (plasmid) | This paper | Deposited to Addgene | Addgene Cat# 169414 |
| Recombinant DNA reagent | pAAV-TRE-DIO-hM4Di-mCherry (plasmid) | This paper | Deposited to Addgene | Addgene Cat# 169415 |
| Sequence-based reagent | chat_F | IDT | PCR primers | TCTGGCAACTTCGTCGGA |
| Sequence-based reagent | chat_R | IDT | PCR primers | CTCCTGGGCTGTTACGCAC |
| Sequence-based reagent | pV2.1 – Gene Block 7xTetO-LoxP-Lox2272-tTAH100Y.SV40 | IDT | Gene block, custom | |
| Sequence-based reagent | pV2.2 – Gene Block oChIEF-LoxP-Lox2272 | IDT | Gene block, custom | |
| Commercial assay or kit | In-Fusion HD Cloning Plus | Takara/Clontech | Cat#: 638920 | |
| Commercial assay or kit | High-Capacity cDNA Reverse Transcription Kit | Applied Biosystems | Cat#: 4368814 | |
| Peptide, recombinant protein | T4 DNA Ligase | NEB | Cat#M0202S | |
| Peptide, recombinant protein | BglII | NEB | Cat#R0144S | |
| Peptide, recombinant protein | AscI | NEB | Cat#R0558S | |
| Peptide, recombinant protein | BamHI-HF | NEB | Cat#R3136S | |
| Peptide, recombinant protein | PmlI | NEB | Cat#R0532S | |
| Peptide, recombinant protein | Phusion High-Fidelity DNA Polymerase | NEB | Cat#M0530S | |
| Chemical compound, drug | Clozapine | Sigma-Aldrich | Cat#C6305-25MG | |
| Chemical compound, drug | Fast Blue | Polysciences Inc | Cat#17740-1 | |
| Other | Mt.Lion Urine | Maine outdoor solutions LLC | | Obtained from https://predatorpeestore.com/. |
| Software, algorithm | Prism | GraphPad Software Inc | RRID:SCR_002798 | |
| Software, algorithm | Sigmaplot 12.5 | Systat Software Inc | RRID:SCR_003210 | |
| Software, algorithm | OriginPro 9.1 | Origin Lab Corporation | RRID:SCR_014212 | |

*Appendix 1 Continued on next page*

*Appendix 1 Continued*

| Reagent type (species) or resource | Designation | Source or reference | Identifiers | Additional information |
|---|---|---|---|---|
| Software, algorithm | Fiji is just imagej | Fiji | RRID:SCR_002285 | |
| Software, algorithm | FreezeFrame v3 | Actimetrics | RRID:SCR_014429 | |
| Software, algorithm | MATLAB | Mathworks | RRID:SCR_001622 | |
| Software, algorithm | Pre-processing analysis MATLAB Script for FiberPhotometry | Doric | | |
| Software, algorithm | ACh sensor analysis MATLAB script | Crouse, Richard B., et al. Elife 9 (2020): e57335 | | |

