## [Editor Report]

This important study examines the existence of a specific population of memory encoding acetylcholine neurons of the basal forebrain which regulate the amygdala for fear expression. Using a combination of techniques including genetic access to c-Fos expressing neurons, in-vivo chemogenetics, and optical detection of acetylcholine (ACh), the authors present convincing evidence that posteriorly located amygdala projecting basal forebrain cholinergic neurons participate in cue-specific threat learning and memory. This paper will be of interest to those studying circuit-level mechanisms of learning and emotion regulation.

---

## [Decision Letter]

**Decision letter after peer review:**

Thank you for submitting your article "Basal forebrain cholinergic neurons are part of the threat memory engram" for consideration by *eLife*. Your article has been reviewed by 3 peer reviewers, including Joshua Johansen as the Reviewing Editor and Reviewer #3, and the evaluation has been overseen by Laura Colgin as the Senior Editor. The following individual involved in the review of your submission has agreed to reveal their identity: Ekaterina Likhtik (Reviewer #2).

Essential revisions:

1) The authors use hM4Di to "silence" Fos-tagged neurons in the basal forebrain, but they have not validated the efficiency or the possible various effects of this reagent.

It is possible that hM4Di actually has a relatively small effect on suppressing the AP activity of neurons. Nevertheless, hM4Di might still be an effective manipulation, because it was shown to additionally reduce transmitter release at the nerve terminal (see e.g. Stachniak et al. (Sternson) 2014, Neuron). Thus, the authors should evaluate in control experiments whether hM4Di expression plus CNO actually electrically silences the AP-firing of ChAT neurons in the BF as they seem to suggest, and/or if it reduces ACh release at the terminals. For example, one experiment to test the latter would be to perfuse CNO locally in the BLA; after expressing hM4Di in the cholinergic neurons of the BF. At the very least, the assumed action of hM4Di, and the possible caveats in the interpretation of these results should be discussed in the paper. See Reviewer 1, Point 1.

2) Throughout the paper, the authors use comparisons of cell activity between groups to address questions about projection-specific and cue-specific cell activation and reactivation. However, statistical comparisons are sometimes done between biological replicates (animal groups e.g. Figure 5A), whereas a lot of them are done between technical replicates (data points/slices, e.g. Figure 2B, 5B, 7B). Adding statistics that compare biological replicates would help increase confidence in the results.

3) To demonstrate engram-like specificity, in figure 4C the authors show fold change in cholinergic reactivation in low and high responders (animals that show low and high defensive freezing upon cue presentation) as normalized by cell activity while sitting in the home cage. However, the authors also collected a better control for this comparison, which is shown in figure S4, where the animals were exposed to an unconditioned tone cue. Comparing fold change to this tone-alone condition would provide stronger evidence for the authors' point, as this would directly compare the specificity of cholinergic reactivation to a conditioned vs an unconditioned cue. A discussion of the same comparison is relevant for figure 2 (and is shown in figure S4) but is not mentioned in the text.

4) The significant correlation between cue-evoked percent change in defensive freezing from pretone and fold change in cholinergic cell activity relative to the home cage that is shown in figure 4D is somewhat confusing. Is the correlation considering all the points shown (high and low responders as depicted by black and grey points)? It's first reported as one correlation but then is discussed as two populations that have different results. Further, is the average amount of reactivation for the home-cage controls used here the same denominator for each reported animal? Similarly to the point above, a correlation looking at fold change from tone-alone would also be helpful to determine the degree to which cholinergic reactivation is specific to threat-association learning versus the more general attentional component that this system is known for.

5) Some important controls are missing from the 'engram' causal manipulation study (Figure 3) which makes it difficult to definitively support the authors' contention that the captured cells are in fact 'engram' cells. The established definition of engram cells refers to those which store memory and are activated during learning and memory expression. While the authors do show conditioning-induced changes in CS activation in NBM-ACh cells, they don't have the necessary controls for their causal manipulations to rule out other non-engram factors which could account for their results. For example, it is possible that the captured cells are simply activated by auditory cues (even prior to learning, the immuno approach in Figure 2 may not be sensitive enough to detect this) and that they are transmitting CS-activated ACh signals to BLA which are necessary for memory expression. It is also possible that inhibiting a small number of NBM-ACh neurons (not restricted to 'engram' cells) reduces memory expression (early engram studies by the Josselyn lab controlled for this). The authors could address this with further experiments such as including 1) a control group in which cells are captured (off-dox) during a habituation session with CS presentation prior to on-dox learning and 2) using a dilute virus to manipulate a similar number of cells as in the engram group. Alternatively, they could present a more careful discussion of this issue and potentially avoid the use of the term 'engram'.

6) The authors suggest that the ACh 'engram' cells participate in the expression of learned defensive responses through projections to the BLA. However, they have not demonstrated this definitively. To do this would require testing the effects of the inactivation of ACh-engram neurons which project to BLA on learned freezing responses. This may be technically challenging, but one approach could be to use the ADCD technique they've developed and inactivate terminals of captured cells in the BLA (chemogenetically or optogenetically). At a minimum, the authors should examine whether retrogradely labeled ACh cells projecting to BLA are Fos activated during a recall session (compared with appropriate controls).

7) For most experiments, significantly more raw data should be shown (e.g. raw example traces for GRAB-ACh3.0), and also brain section images for almost all experiments. Significantly more raw data should also be shown in the Main Figures. Please see Reviewer # 1, "Specific points relating to data presentation and statistical analysis", for details and specific examples."

*Reviewer #1 (Recommendations for the authors):*

Specific points relating to data presentation and statistical analysis:

1) The reporting of statistical data (N's, p- values, and the test used) is a bit cryptic. In general, the authors seem to report these values in the Figure legends. This, however, has the advantage that the reader has to constantly jump between the results text and legends text if he/she wishes to check the values for a given experiment. I would therefore suggest that the authors report all values of statistical tests in the results text (N's, p- values, and the test used).

Also, in some instances, N-values seem to be missing from the legend texts. Please check all legends / Results text accordingly.

2) Figure 1:

– Traces of GRAB-ACh3.0 should be shown more extensively. In Figure 1B, only the onset to the CS is shown – the response along the entire tone (30 s) should also be shown.

– Furthermore, the trace in Figure 1B seems to be an average over all mice. In addition, traces from individual mice should be shown, to allow an estimation of the S/N ratio of these measurements.

– In the quantification (Figure 1B, right), it would be easily possible to show the individual data points (there are 11 mice). I did not understand what the leftmost quantification tells us ("Pre-tone"). Please explain how this was calculated, and inasmuch it is relevant.

– What is the unit "Z%deltaF/F0"? In my view, this should either be Z-scored or else, deltaF/F0 – please clarify.

3) Figure 2:

– the authors should extensively show images of brain sections. It is curious to see that the entire paper doesn't show a single image at low magnification, which shows the investigated basal forebrain areas, and stained neurons in there, in their anatomical context. This would allow the reader to put the images into context, and to roughly validate whether the analyses by the authors (e.g. counted cells) make sense. For example, for Figure 2A, please show an additional low mag image of the corresponding brain area, that shows this brain area (NBM/SIp) in the wider context. The actual image can then be shown as an insert.

– In the plot of Figure 2B, a tick at the value of 0 is missing – please add this (this also applies to Figure 2C, 4B, 4C, 5A, right, 5B, right, 5C, right, 7A, 7D, S6B, S6C, S7A-C – please add all ticks at zero). Adding the tick at zero will help the reader eyeball the values of the data points.

– In some plots, the y-axis doesn't start at 0, but at an artificially chosen higher value. This increases "graphically" the effect size; however, it doesn't allow the reader to look at the data in the most unbiased way (indeed, some Journals do not allow this practice) (see Figure 7C, Figure S6A, right). Please newly plot the data from zero.

– The authors did not report in the Statistics part, which criteria they used to add "stars" onto the plots. For example, Figure 2B has *** (3 stars), but the reported p-value is p = 0.0017. According to the criteria used by many authors, p < 0.05 (1 star), p < 0.01 (two stars), and p < 0.001 (three stars). Thus, Figure 2B should have 2 stars, but not 3. Please explain the criteria you used, and please validate all star symbols according to the criteria you used.

– What is the unit of the cell numbers plotted in Figure 2B, C? Were these indeed absolute numbers as the axis labelling seems to suggest, or was there a normalization for the analyzed area? Please comment.

4) Figure 3:

– While the freezing data is shown at a quite high time resolution for the training day (Figure 3B), for recall (Figure 3C), only excessively binned data are shown. It would be advantageous for the reader to see the data at the same time resolution, in a quasi-continuous trace, to evaluate the effects also of silencing.

5) Figure 4:

– There are several issues with the definition of high and low freezers, and with the analysis in this Figure.

– Hig+low freezers: The Methods state "High responders were defined as those mice that exhibited at least a 10 percentage point increase in % time spent freezing in the bin during the tone from the bin immediately pre-tone (e.g. Pre-tone freezing 10% to tone-induced freezing of {greater than or equal to}20%)." However, the Results text speaks about a "3-fold increase" (l. 235) – these are different criteria, please explain.

– Related to the above, the y-axis labeling in Figure 4B is unclear – is this the difference in freezing (as the Methods would suggest) or a fold-change (like the Results would suggest)? Also, the 3-star symbol for p = 0.005 seems not justified, should be 2 stars according to often-used criteria (see also above).

– l. 238 states: "There was no difference in mCherry expression in 'Low Responders' compared to the home cage group (fold change ~ 1),"

– this should be tested statistically.

– I don't see that the freezing data in Figure 4D (y-axis) shows a "positive correlation" with the number of Fos-labelled cells (x-axis). Indeed, looking at the high responders only, the data falls very close to a flat line. Please explain how the correlation was analyzed – this was probably done by fitting all data, i.e. also including the "low" responders (?). The obvious absence of a correlation for the "high responders" alone (black data points) is worrisome.

– In Figure 4D, the location of the data points (y-axis, freezing; about -10 – 0%) does not seem to correspond to what is plotted in Figure 4B (~ 2%; please explain).

– Figure 4F, and Figure S5: The result about the location of labelled cells along the a-p axis of the brain should be documented with a series of original images, both at low and high mag (see general comments, above).

6) Figure 5:

– The location of back-labelled cells should again be amply documented with several low and high mag images from all analyzed areas (hDB, VP/SIa, NBM/SIp).

– 3 stars in Figure 5B might be 2 stars.

7) Figure S4:

– was the data in Figure S4C a further analysis of data in Figure 2B? If yes, why are the animal numbers seemingly different between the two datasets?

8) Figure S5:

– The legend doesn't state the N-numbers.

– The box plot should be explained (meaning of the boxes). Even better, individual data points should be shown instead, if this is feasible.

9) Figure S6:

– For the data in A, again example images at low and high magnification should be shown …

– For the images in B, please indicate lateral-medial, and dorsal-ventral directions in these images (assuming that these are coronal sections?).

– The data in C again suggest that there might also be an effect on contextual fear learning (see also above).

10) Figure S7

– The text describing the data in A states "did not freeze to the tone" (l. 305). This is not a scientifically warranted statement, since there seems to be residual freezing. Rather, the data should be tested statistically.

– The N-numbers for the data in A, and B should be stated in the legends.

– For these and possibly similar experiments, the authors need to show in detailed histological images, for each individual mouse, that the expression of hM4Di has occurred in the expected set of cholinergic BF neurons.

11) Figure S8:

– I wonder why the firing rate data of the "Training" group in panel H is extremely small (around 5 Hz or less), and smaller than in all other groups. Is there another type of modulation of intrinsic excitability, a down-modulation, after training? Please explain.

12) Figure S9:

– In panel C, could you show the freezing data at a higher time resolution, e.g. the 30s bins also used in Figure 3B?

*Reviewer #2 (Recommendations for the authors):*

– For the flow of the story, I recommend switching the order of presentation where the lack of effect on behavior after silencing the CeA-projecting NBM/SIp neurons (lines 286-301) comes after the data showing the effect of silencing the BLA-projecting NBM/SIp.

– I would also recommend moving figures S7A and S7B into the main figures as they provide important evidence for the author's point.

– Given that the authors don't see an effect of clozapine on memory when inactivating during the consolidation period (Figure S7C), it might be worthwhile to discuss why this might be the case, and the effects of cholinergic input starting cascades of neural activation in the BLA during training.

– Overall, there are missing Bregmas and mapping of placements in several places (e.g. Figure 1A – Bregma and mapping of placements/spread of the virus across animals, it appears that the recordings are relatively anterior in the BLA; Bregma in Figures 5, S3C, 6SB, S6C – not clear how much of a CeA there is at this location, 7D; a zoomed out section of where the example data in figure 7C is coming from).

– It would also be helpful to indicate the area where the signal fluorescence is being calculated in traces shown in figures 1 and S1.

*Reviewer #3 (Recommendations for the authors):*

1) For the 'engram' data in Figure 2, the logic of the double labeling is a bit confusing as described and it's not clear, from a simple reading of the paper, how the authors achieve double labeling of encoding (training activated) and recall activated cells. After digging around in the Methods, Figs/legends I was able to figure it out, but it was difficult initially. The authors should state this in a simple way in the main text along with a description of how this works at the level of transgene expression at different time points.

2) Some important controls are missing from the 'engram' causal manipulation study (Figure 3) which makes it difficult to definitively support the authors' contention that the captured cells are in fact 'engram' cells. The established definition of engram cells refers to those which store memory and are activated during learning and memory expression. While the authors do show conditioning-induced changes in CS activation in NBM-ACh cells, they don't have the necessary controls for their causal manipulations to rule out other non-engram factors which could account for their results. For example, it is possible that the captured cells are simply activated by auditory cues (even prior to learning, the immuno approach in Figure 2 may not be sensitive enough to detect this) and that they are transmitting CS-activated ACh signals to BLA which are necessary for memory expression. It is also possible that inhibiting a small number of NBM-ACh neurons (not restricted to 'engram' cells) reduces memory expression (early engram studies by the Josselyn lab controlled for this). The authors could address this with further experiments such as including 1) a control group in which cells are captured (off-dox) during a habituation session with CS presentation prior to on-dox learning and 2) using a dilute virus to manipulate a similar number of cells as in the engram group. Alternatively, they could present a more careful discussion of this issue and potentially avoid the use of the term 'engram'.

3) The authors suggest that the ACh 'engram' cells participate in the expression of learned defensive responses through projections to the BLA. However, they have not shown this definitively. To do this would require testing the effects of the inactivation of ACh-engram neurons which project to BLA on learned freezing responses. This may be technically challenging, but one approach could be to use the ADCD technique they've developed and inactivate terminals of captured cells in the BLA (chemogenetically or optogenetically). At a minimum, the authors should examine whether retrogradely labeled ACh cells projecting to BLA are Fos activated during a recall session (compared with appropriate controls).

4) In Figure 5, the authors report a reduction in BLA Fos-activated neurons with chemogenetic inhibition of BLA-projecting ACh neurons. However, this experiment requires a control group without training (or, ideally, with unpaired CS/US training) to demonstrate that the inhibition is affect learning-induced activation of BLA neurons. While others have reported this using other paradigms previously, it is still important to show here using their own preparation.

[Editors’ note: further revisions were suggested prior to acceptance, as described below.]

Thank you for resubmitting your work entitled "Basal forebrain cholinergic neurons are part of the threat memory engram" for further consideration by *eLife*. Your revised article has been evaluated by Laura Colgin (Senior Editor) and Josh Johansen (Reviewing Editor).

The manuscript has been improved but there are some remaining issues that need to be addressed, as outlined below:

We thank the authors for their response to the reviewers' previous suggestions. Many of the findings have been strengthened through the addition of new experiments and analyses. However, there are remaining concerns about the statistical approaches used and low population size in some experiments which should be addressed. Related to sample sizes, power analyses or clear demonstrations of existing standards in the field should be used to justify animal numbers. If necessary, further animals should be added to experimental groups (e.g. the differential role of NBM/SIp and VP/SIa in learned vs. innate threats) or, as suggested by reviewer 1, data with low n could be removed if it does not seriously undermine the conceptual advance of the paper. In addition, the strong focus on the activated basal forebrain cholinergic cells as being part of the threat memory engram is not well supported. The authors should moderate the claims related to this aspect of the paper.

Furthermore, the title and/or abstract should provide a clear indication of the biological system under investigation (i.e., species name or broader taxonomic group, if appropriate). Please revise your title and/or abstract with this advice in mind.

*Reviewer #1 (Recommendations for the authors):*

For this first revision, improvements in the data display have been implemented, at least concerning the traces for GRAB-ACh measurements. In addition, new data has been added; thus, the ms has been reorganized quite substantially. Nevertheless, there are still a lot of questions regarding the statistical analyses, and the anatomical analysis. In one instance, it seems that a control data set was used twice (Figure 5C and Figure 5SD; see Specific point 16 below).

Since my list of points is long, the question arises what might be the best strategy to go forward with the paper? Most of my points are about data display, statistical analyses, and similar, and it should be feasible to address these points. On the other hand, I also feel that especially the anatomical analysis relating to Figures 5 and 6 is still not clear and that some of the points the authors want to make related to these Figures, especially about the a-p distribution of (re-)activated Ach neurons (differential role of NMB/SIp versus VPSIa neurons in learned fear), differential role of these pools in "high" and "low" responders, and differential role of the same in learned versus innate fear, are unclear, and might go beyond the scope of the present paper. In my view, the authors should therefore consider the possibility to remove some of the controversial data, and related claims (see my detailed comments – these data are also often plagued by low n-numbers and incomplete anatomical analyses), and concentrate on the remaining main findings of the paper.

Specific points (in order of appearance)

1) l. 156/157 "when it was not explicitly paired with a shock" (data in Figure 1S3D).

– Please explain in the legend the exact temporal sequence between shock(s) and tone(s) used for this "non-explicitly paired" protocol (or does this refer to the "shock-only" protocol?).

2) Figure 1 S3C, D: The legend to this Figure states that here, the results from a "shock-only" training are displayed for the subsequent "recall" session.

First, the authors should also show the freezing response for the standard group undergoing standard fear learning (3x tone-shock pairings); this is important for a comparison to the "shock-only" training group.

Second, the Results text states that Figure 1S3D shows the ACh response when "it was not explicitly paired with a shock".

– Is this the same as the "shock-only" protocol described in the legend? Please clarify, and make changes in the manuscript as appropriate.

3) Figure 1 S4 was not called out from the Results text; this should be changed i.e. what was done in this Figure, should be (briefly) mentioned in the Results.

4) The use of star symbols in Figure 2C, in conjunction with what is stated in the corresponding Results text, is misleading and tends to over-interpret the statistical significance of this data set.

Specifically, l. 202 states: "Significantly more double-positive cholinergic neurons were seen following the complete (tone + shock followed by tone recall) associative threat learning paradigm compared to mice that underwent session 1 without shocks (Figure 2C, p=0.0249)."

– This is misleading. The Figure legend and statistics table state that the value "p = 0 0.0249" refers to the "overall" significance of the Kruskal-Wallis (KW) test used here, but NOT to the specific comparison cited in the above sentence. For that specific comparison, the p-value was higher ("p=0.0464, Dunn's corrected", see Figure legend).

– In response, the authors should spell out the "overall" significance of this dataset (effect of group; p = 0.0249; KW) in the Results section, followed by the pertinent p-value for important specific comparisons (in this case, p = 0.046 for the comparison between the "tone-tone" and tone+shock – tone" groups). The n-values for these pertinent comparisons should also be given in the Results text.

Furthermore, in this dataset (Figure 2C), the post-hoc test for the comparisons between the home cage (HC) and standard fear conditioning group is NOT significant (p = 0.72). This value should be indicated by a "bracket" symbol in Figure 2C, and the meaning of this non-significance should be discussed in the manuscript.

5) The p-values, n-values, and the statistical tests applied, should be reported together either in the figure legends or in the Results text (please choose one section to use for reporting these statistics, in addition to the statistical table) throughout the entire manuscript (see point 4, and see also my previous point "Data Recommendation 1").

6) Figure 2D:

l. 210 states:

"All three conditions showed few reactivated neurons and no differences between groups (p=0.9471)"

– The finding that all these 3 control groups are not significantly different from each other, is not surprising and does not go to the point of the control experiment. Rather, it should be tested statistically whether the number of re-activated neurons in these three conditions is significantly different from the one after standard fear conditioning (the data in Figure 2C, right column).

7) I think the title of the Sub-header:

"Reactivation of cholinergic neurons activated by training is required for learned behavioral responses" (l. 214) is wrong, it should read "Activation" instead (and then replace "activated by" with "during").

To my understanding, the chemogenetic silencing experiments in Figure 3 test the role of activated cholinergic neurons but do not suppress the "re-activation" of neurons (during recall…), since ADCD expression of hM4Di is driven during the training session.

8) In Figure 3 S1B, the average trace of time-resolved freezing is shown, for the same data set as the one in Figure 3D. However, the number of mice included in the time-resolved display was lower (n = 5 and 6) than in the display of Figure 3D (n = 7 and 8). This must be homogenized, i.e. all data used for the statistical analysis in Figure 3D (n = 8 and 7 mice) should also be shown in a time-resolved fashion for Figure 3S1B.

9) l. 273 states:

" We injected cav.hM4DiBLA or sham control mice with clozapine (CLZ) 10 min prior to initiating cue-conditioned threat learning"

– Please add here that these were IP (?) injections, for a quick reference for the reader. Indeed, this information should possibly also be added to the Methods (I did not find it upon a rapid check).

10) The authors should control for the possibility of leaky expression of hM4Di, following presumed Cre-dependent retrograde expression from a CAV2 virus after injections in the BLA (Figure 5). Because, as I assume, CLZ was not injected into the basal forebrain (BF) but rather IP (see point 9), a leaky expression of hM4Di beyond the cholinergic BF has the potential to strongly change the interpretation of this experiment. Thus, the authors have to show, first, whether hM4Di-mCherry is expressed at the injection site in the BLA itself (this would lead to a direct inhibition of BLA neurons). This should be done by showing example images of hM4di-mCherry fluorescence on the level of the BLA, and by analyzing the number of hM4Di-mCherry+ neurons in the BLA over the entire dataset. Second, the authors should show extensive raw example images of the BF at different a-p locations (and of further brain areas in a zoomed-out fashion), in an attempt to exclude the possibility that hM4Di might also be expressed in non-cholinergic neurons (ChAT could be visualized by IHC). Only a non-leaky expression of hM4Di-mCherry, essentially limited to ChAT-positive neurons in the BF, would support the conclusions of the authors.

11) On l. 298, the results of a control experiment is reported, which is not documented ("data not shown"). Some journals do not allow the statement "data not shown". The authors should document the control data set, or else remove this statement from the manuscript. See also l. 441, another "data not shown", which should be similarly documented or else removed.

12) Data in Figure 5C.

When checking the p-values from the legends/statistics table, I was puzzled by the fact the comparison of the two "grey" data sets has a reported p-value of < 0.0001, although many of the data points between the two groups overlap. On the other hand, the comparison between the "sham" group and hM4Di group at the "pre-tone" time had a p-value of 0.096, although the overlap of the data points is only partial. Please comment, and validate the underlying tests.

13) Related to point 12, there is a strong tendency that freezing before the tone is lower in the hM4Di group as compared to the "sham" group. Although the authors reported a p-value of 0.096 (and therefore, "not significant") for this comparison (Figure 5C), this p-value should nevertheless be verified. Because the trend is seen in this data set, as well as in several other ones (Figure 5B and Figure 3D), the authors should discuss the possibility that chemogenetic suppression of ACh release might, in addition to the effects studied here, also reduce contextual fear learning (see also my previous "Specific comment 2").

14) Still related to the data in Figure 5C:

The time-resolved display of tone-evoked freezing during the recall day should be shown both for this data set (Figure 5C), and for the data set in Figure 5B, similarly as was done above for the data in Figure 3D (see panel Figure 3 S1B). This would also be helpful to gain insight into the p-values resulting from ANOVA / KW testing (see point 12), which were probably performed on similar time-resolved data sets (because the effect of "time" was always tested).

15) l. 299 states:

"Differences in recall-induced Fos expression between sham and cav.hM4DiBLA mice were maximal in rostral portions of the BLA (between bregma -0.8mm to -1.4mm) (Figure 5-Supplement 1B)"

– First, the analysis in Figure 5 S1B seems to have been done in only N = 1 mouse, which is problematic (please confirm the n-number, and state it in the Results text and in the legend to that Figure).

Second, I actually fail to see an a-p gradient as claimed by the authors (higher in anterior than posterior). If one looks at the number of Fos+ cells in the hM4Di group (red data points), then it seems that hM4Di suppresses the number of Fos+ cells, in relative terms, stronger at more posterior levels (e.g. reduction by > 80% at -1.2 mm) than at anterior levels (e.g. reduction by < 40% at -0.8 mm). Wouldn't this suggest that opposite from the author's interpretation, MORE cholinergic neurons are specifically (in a hM4Di-dependent manner) activated posteriorly than anteriorly? Please comment.

16) Visual inspection of the data in Figure 5 S1D strongly suggests that the data in the control ("sham") group (n = 9, grey data) are actually the same as the data for the sham group in Figure 5C. In Figure 5C, retrograde expression from the BLA was studied, whereas in Figure 5 S1D, the effects of retrograde expression of hM4Di from the CeA were studied. If the experiment was originally set up for 3 groups of mice (thus, sham control – CAV2-hM4Di in BLA – CAV2-hM4Di in CeA), then this should be reported as such, the data should be shown in one panel, and the statistical comparisons should be done accordingly (i.e. the "multiple comparison" correction would need to be done for 3 groups). If, on the other hand, the CeA experiments were done without proper controls, and were simply compared to the control group from another experiment, then the CeA data should be removed. Please comment, and make appropriate changes in the manuscript.

17) Figure 5 S2 claims that neurons in the "NBM/SIp" versus neurons in the "VP/SIa" have been targeted selectively with AAV9:DIO:hM4Di-mCherry. This should be documented extensively, by showing expression of hM4Di-mCherry throughout several brain slices in a de-zoomed fashion for an example mouse, and by showing average results from the quantification of the localizations in a suitable manner (for example, by plotting cell density/number along the a-p axis), for all investigated mice. Otherwise, it remains unclear whether the two areas have indeed been targeted differentially.

18) In Figure 5 S2A, the "triple star" symbol comparing the two control data points might be removed. It is not surprising that the tone recall causes increased freezing in the "sham" group.

19) The data set reported in Figure 5 S2B looks underpowered (only n = 3 and 4 mice). I highly doubt that with such a small sample size, significant conclusions can be drawn in a behavioral experiment. For some reason, the data in this experiment seem to scatter much less than in comparable experiments even with larger n's (compare to the neighboring panel A, bottom) – please comment. Also, the use of star symbols (3 stars?), and the use of the bracket symbols is misleading (the brackets suggest that the data was pooled amongst all "pre-tone" and "recall tone" groups; the rationale of this analysis approach remained cryptic). Importantly, there was no group effect in the ANOVA (p = 0.1574 was reported). Thus, no further post-hoc group comparisons should be done. Overall, I think the authors need to first, show that they can indeed differentially target NBM/SIp versus VP/SIa (see point 17). If this can be shown, then they would have to obtain significantly more data in both the control and effect groups for the VP/SIa targeting, to drive the point home that targeting these two areas has differential behavioral effects.

On the other hand, the authors might simply remove the data in Figure 5 S2A and B from the manuscript, and remove the claim about the differential effects of the NBM/SIp and VP/SIa. These claims are currently little supported by experimental evidence, and I feel they are not important for the overall conclusions of the paper.

20) Figure 6A shows time-resolved freezing responses on the recall day for n = 9 control mice, and n = 9 effect mice. However, it remained unclear from which of the previous experimental groups these data were taken; whether a sub-selection was made, and whether animals were pooled from various manipulations. This should be explained in detail in the Results text and legend.

21) The "fold-change" analysis of expression analyzed in Figure 6D remained unclear to me (sorry if I didn't get it). Please explain this briefly in the Results text, and make sure it is explained in more detail in the Methods. Also, please give details about the viral construct used in Figure 6C (currently it only states "ADCD").

22) The picture shown in Figure 6E is simply a scheme of the idea that neurons activated in "high responders", in your view, should be localized more anteriorly, but it is in no way a documentation of an anatomical observation. This scheme should be replaced by experimental evidence.

23) The data that might relate to the above issue (localization of ChAT+, activated neurons along the a-p axis in "high" and "low" responders), might be contained in Figure 6 S2. However, the data seems to be only partially analyzed, and it is confusing to fully understand how the plot in panel C was derived. First, it was unclear from which group of mice the images in Figure 6 S2A were taken. Please explain in the legend from which experiment these images were taken, which mouse genotype was used (Fos-tTA/shGFP with or without ChAT-Cre?), and which viral constructs were injected in these mice.

For panel C, I assume that the red data point represents an analysis of the images in panel B (correct?), but I am unsure from which data set the black data points were taken (# of recall-activated BFCNs (ADCD)") – please mention this in detail in the legend of the Figure. Were the black data points related to the images in panel A? were they taken from "high" or "low" responder mice?

24) The images shown in Figure 6 S2C, should be shown in a de-zoomed fashion for several representative brain images on the level of the BF, and beyond. Such a documentation of the data would represent the control for "leaky" expression mentioned above (best after co-staining against ChAT; see point 10).

25) The 2-star symbol in Figure 7D above the comparison "HC" – Fos+" should probably be 3-stars, not 2-stars (p = 0.0006).

26) The authors should use the less "pushy" and more conventional criteria for using star symbols for statistical significance (i.e. 1 star, p < 0.05; 2 stars, < 0.01; 3 stars < 0.001), and omit the "4 star" symbol and definition throughout. The 1-3 star criteria should be stated in the Methods (I didn't find it). The paper in its current version has only a single "4 star" symbol (Figure 3D), which should be changed to "3 star" according to the above definition, or removed (see also point 18) (see also my previous point "Data Recommendations 4").

27) The bracket symbol in Figure 7F is probably wrong. On the left side, it should only point to the "HC" group, but not to both the "HC" and "Fos-" group as it stands now.

28) The authors claim that "latency to fire an AP" and AHP might significantly change in re-activated ChAT neurons (Figure 7 S1E, F). However, these effects are somewhat more subtle than the other effects shown in the main Figure 7. For example, the effect of "latency" seems to be driven merely by some outlier neurons in the home cage group with very long latencies (the authors would need to show example recordings), whereas the data from the Fos- and Fos+ groups virtually overlay and the latter comparison is the more pertinent one. Thus, I think it is more cautious to drop this claim.

29) l. 427 states:

"Next, we quantified the number of cholinergic neurons expressing Fos (Fos shGFP+) after saline or predator odor exposure"

– Please explain how long the mice were exposed to the odors.

30) In Figure 8, the authors seem to analyze the results of a triple-labeling anatomical experiment; i.e. there are neurons that are Fos+; neurons that are back-labelled by fast Blue; neurons that are ChAT+, and all combinations thereof. Thus, to give the reader a better view of the various neuron pools, the authors should report the numbers of all pools of neurons (i.e. the "green", "red", "blue" neurons and all combinations thereof). This can be either done numerically or in the form of a Venn diagram. As it stands, only part of the information is given (Figure 8C, right; which seems to correspond to the percentage of back-labelled, ChAT+ neurons which were also Fos+). However, this is only part of a bigger picture. For example, the images in Figure 8C, and in Figure 8 S1A (middle, right) show also many non-ChAT+ neurons, that were re-activated in the VP/SIa area.

31) What does the dashed line label "Home cage" in Figure 8C, right panel, mean, and how was it derived?

32) l. 442:

" Nearly the entire subset of BLA-projecting VP/SIa cholinergic neurons were also Fos-shGFP+ (Figure 8C right)."

– Another non-scientific statement. Rather, the average plus/minus S.D. of the quantification should be reported. In addition, information about all labelled neuronal populations should be given (see above, point 30).

33) The retrograde expression experiment of hM4Di from the BLA in Figure 8D is expected to drive hM4Di expression in all cholinergic BLA-projectors, not selectively in the VP/SIa. This caveat in the interpretation should be mentioned more carefully when the results of this section are summarized (lines 451 – 454).

*Reviewer #2 (Recommendations for the authors):*

I thank the authors for addressing my concerns, and I'm happy to recommend the manuscript for publication in its current form. There are a few figures that would benefit from additional labeling (outlined below).

Please add anatomical references for basal forebrain in figures F1S4, F2S1a, F4c, F5S1a, F5S1b , F6S2a.

*Reviewer #3 (Recommendations for the authors):*

The authors have addressed many of my previous concerns and now nicely dissociate the role of NBM/SIp and VP/SIp ACh neurons in learned threat memory processing and show that NBM/SIp ACh cells are activated by memory recall and necessary for threat memory expression. However, I still have a lingering concern with their strong emphasis on the role of NBM/SIp ACh neuron (and their projections to BLA) as being memory 'engram' cells. While some of their data are consistent with this idea, they do not test key criteria for defining engram neurons, and the new data supporting a role of NBM/SIp ACh cells which project to the BLA in the threat memory 'engram' is only correlative.

Specifically, while there is evidence in support of the NBM/SIp ACh cell population (independent of where they project) in memory engrams (activated by memory cues, necessary for memory expression). There are several criteria for defining engram neurons (see Josselyn and Tonegawa Science 2020) which haven't been met including sufficiency (stimulation should produce memory in the absence of recall cues), mimicry (creation of an artificial engram) and long-lasting cellular or synaptic changes in engram neurons (here they only see excitability differences at 2.5 hr, not at long term memory timepoints). It is possible (even likely) that activity in these neurons modulates/facilitates threat memory formation and expression and are not storing the memory trace themselves.

There is even less evidence that NBM/SIp ACh cells which project to BLA being engram cells. The authors report an increase in ACh release in BLA (which isn't necessarily coming only from NBM/SIp) and added new data showing that NBM/SIp ACh cells projecting to BLA are fos activated.

While there is data supporting a potential role for these cells in engram processing, it is inappropriate to so strongly insist (in the title, abstract, and throughout) that NBM/SIp ACh neurons, and particularly those which project to BLA, are engram cells. The authors should moderate this aspect of the paper and perhaps broaden the discussion of the potential role of these cells in threat memory processing as they've already nicely done in the 'Cholinergic Modulation of Associative Threat Learning' section of the Discussion.

---

## [Author Response]

Essential revisions:1) The authors use hM4Di to "silence" Fos-tagged neurons in the basal forebrain, but they have not validated the efficiency or the possible various effects of this reagent.It is possible that hM4Di actually has a relatively small effect on suppressing the AP activity of neurons. Nevertheless, hM4Di might still be an effective manipulation, because it was shown to additionally reduce transmitter release at the nerve terminal (see e.g. Stachniak et al. (Sternson) 2014, Neuron). Thus, the authors should evaluate in control experiments whether hM4Di expression plus CNO actually electrically silences the AP-firing of ChAT neurons in the BF as they seem to suggest, and/or if it reduces ACh release at the terminals. For example, one experiment to test the latter would be to perfuse CNO locally in the BLA; after expressing hM4Di in the cholinergic neurons of the BF. At the very least, the assumed action of hM4Di, and the possible caveats in the interpretation of these results should be discussed in the paper. See Reviewer 1, Point 1.

We clearly demonstrate that activation of hM4Di expressed in basal forebrain cholinergic neurons with clozapine significantly alters behavior and/or activation of neurons in projection targets of BFCNs (see Figure 3, Figure 3-supplement 1, Figure 5, Figure 5 Supplement 1, Figure 5-Supplement 2, Figure 6-Supplement 1 and Figure 8). As discussed on Page 17 Lines 488-493 of the revised manuscript, whether the effects we see on behavior result from inhibition of action potential firing per se, or from decreasing ACh release from terminals (or both) is not clear. However, the effect on behavior is striking and consistent with prior demonstrations that activating hM4Di (or hM3Dq) in cholinergic neurons changes membrane potential, blocks action potential firing and reduces firing rate (Zhang et al. 2017 and Jin et al. 2019) demonstrating DREADD-mediated excitation and inhibition of basal forebrain cholinergic neurons can alter electrical activity and behavior.

2) Throughout the paper, the authors use comparisons of cell activity between groups to address questions about projection-specific and cue-specific cell activation and reactivation. However, statistical comparisons are sometimes done between biological replicates (animal groups e.g. Figure 5A), whereas a lot of them are done between technical replicates (data points/slices, e.g. Figure 2B, 5B, 7B). Adding statistics that compare biological replicates would help increase confidence in the results.

We have replotted our data as scatter plots showing values from individual animals

(biological replicates) in New Figures 1 – 8 and in Figure 1-Supplements1, 2, and 3,

Figure 5-Supplements 1 and 2, Figure 6-Supplements 1 and 2, Figure 7-Supplement 1,

and Figure 8-Supplement 1. All statistical analyses have been calculated using biological replicates. To further clarify these analyses, we have included a statistical reporting table that details sample size, group medians, confidence intervals, statistical tests used (and when appropriate, post-hoc corrections). This can be found as a separate file uploaded with the manuscript associated documents.

3) To demonstrate engram-like specificity, in figure 4C the authors show fold change in cholinergic reactivation in low and high responders (animals that show low and high defensive freezing upon cue presentation) as normalized by cell activity while sitting in the home cage. However, the authors also collected a better control for this comparison, which is shown in figure S4, where the animals were exposed to an unconditioned tone cue. Comparing fold change to this tone-alone condition would provide stronger evidence for the authors' point, as this would directly compare the specificity of cholinergic reactivation to a conditioned vs an unconditioned cue. A discussion of the same comparison is relevant for figure 2 (and is shown in figure S4) but is not mentioned in the text.

We compared cholinergic neuron response to tone under three conditions: (1) in animals that were exposed to three tone-shock pairings, and then to a single tone; (2) in animals exposed to three tones (no shock) and subsequently to a single tone; and (3) in animals exposed to three shocks (no tone) and subsequently to a single tone. These responses were compared to responses from animals that remained in the home cage. There was no significant increase in ACh release in the BLA in response to tone presentation unless the tone had previously been paired with foot shock (condition 1) (new Figure 1 and Figure 1Supplements 1-4), and no significant increase in tone associated re- activation of cholinergic neurons (using IEG expression as a read-out) relative to home cage unless the tone had been previously paired with foot shock (new Figures 2, 4), and no significant difference between responses to conditions 2, 3, or 4 (Figure 2C- home cage-home cage vs. tone-tone comparison p=0.5012). Based on these analyses, for the subsequent experiments, we have used the home cage conditions as the control group for comparison.

4) The significant correlation between cue-evoked percent change in defensive freezing from pretone and fold change in cholinergic cell activity relative to the home cage that is shown in figure 4D is somewhat confusing. Is the correlation considering all the points shown (high and low responders as depicted by black and grey points)? It's first reported as one correlation but then is discussed as two populations that have different results. Further, is the average amount of reactivation for the home-cage controls used here the same denominator for each reported animal? Similarly to the point above, a correlation looking at fold change from tone-alone would also be helpful to determine the degree to which cholinergic reactivation is specific to threat-association learning versus the more general attentional component that this system is known for.

We agree that our prior presentation was confusing, and we have removed the correlation plots (see new Figure 6 and new Figure 6-Supplement 1). We now use data from a distinct cohort of mice (distinct from the ADCD labeling experiments) that underwent behavioral testing to demonstrate high vs. low responder behavioral performance (new Figure 6A/B). To the reviewer’s point regarding the denominator, counts of ADCD+ neurons from animals that underwent the recall session were normalized to the average of number of ADCD+ neurons in home cage animals in their respective cohorts.

5) Some important controls are missing from the 'engram' causal manipulation study (Figure 3) which makes it difficult to definitively support the authors' contention that the captured cells are in fact 'engram' cells. The established definition of engram cells refers to those which store memory and are activated during learning and memory expression. While the authors do show conditioning-induced changes in CS activation in NBM-ACh cells, they don't have the necessary controls for their causal manipulations to rule out other non-engram factors which could account for their results. For example, it is possible that the captured cells are simply activated by auditory cues (even prior to learning, the immuno approach in Figure 2 may not be sensitive enough to detect this) and that they are transmitting CS-activated ACh signals to BLA which are necessary for memory expression. It is also possible that inhibiting a small number of NBM-ACh neurons (not restricted to 'engram' cells) reduces memory expression (early engram studies by the Josselyn lab controlled for this). The authors could address this with further experiments such as including 1) a control group in which cells are captured (off-dox) during a habituation session with CS presentation prior to on-dox learning and 2) using a dilute virus to manipulate a similar number of cells as in the engram group. Alternatively, they could present a more careful discussion of this issue and potentially avoid the use of the term 'engram'.

Point 5.1: We have measured NBM/SI_p_ cholinergic response to the CS tone using two distinct approaches. First, we have used GRAB_ACh3.0_ to quantify changes in ACh levels in the BLA in response to the CS tone (New Figures 1, and Figure 1-Supplements 1-4). Second, we use an immediate early gene transcriptional response to identify cholinergic neurons that were activated in response to various conditions, (home cage, CS alone, USCS followed by CS). These latter experiments have used a viral approach (ADCD), immunostaining of Fos-shGFP transgene expression and immunostaining of endogenous Fos protein expression. Within the limits of sensitivity of each approach, we have not seen a significant increase in NBM/SI_p_ activation in response to the tone (CS) unless the tone had been paired with foot shock. For example, we did not see increases in ACh release in the BLA in response to naïve tone (first tone presentation during training, Figure 1C and Figure 1-Supplement 1), to three repeat tones followed 24 hr later by tone alone (Figure 1Supplement 2) or in response to tone 24 hr after three foot shocks (Figure 1-Supplement 3). Likewise, we did not see activation or re-activation of NBM/SIp cholinergic neurons in response to tone (above levels seen in home cage animals), unless the tone had been paired with foot shock (Figures 2, 3 and Figure 6-Supplement 1).

Point 5.2: The BLA receives cholinergic input from both the NBM/SI_p_ and the VP/SI_a_ (roughly in a 2 or 3:1 ratio based on the number of cholinergic neurons retrogradely labeled, see for example Figure 5). We have compared silencing of each of these populations on freezing in response to the CS (using AAV-DIO-hM4Di injected specifically into either the NBM or the VP of Chat-IRES-Cre animals, new Figure 5-Supplement 2). Silencing the NBM cholinergic neurons eliminated a significant freezing response to the CS whereas inhibiting the VP cholinergic neurons did not affect freezing in response to the CS. The lack of effect on freezing in response to the CS following VP cholinergic silencing is not consistent with the possibility that generally reducing ACh in the BLA reduces memory expression.

6) The authors suggest that the ACh 'engram' cells participate in the expression of learned defensive responses through projections to the BLA. However, they have not demonstrated this definitively. To do this would require testing the effects of the inactivation of ACh-engram neurons which project to BLA on learned freezing responses. This may be technically challenging, but one approach could be to use the ADCD technique they've developed and inactivate terminals of captured cells in the BLA (chemogenetically or optogenetically). At a minimum, the authors should examine whether retrogradely labeled ACh cells projecting to BLA are Fos activated during a recall session (compared with appropriate controls).

We retrogradely labeled cholinergic neurons projecting to BLA using the retrograde tracer FastBlue in [chat-cre X fos-tTA/shGFP] mice. We simultaneously injected ADCD into the NBM/SI_p_. Animals were trained off-dox. Mice were sacrificed following recall and we quantified BLA-projecting (Fast Blue+) cholinergic neurons that were activated during training (ADCD+) and reactivated during recall (GFP+). ~20% of BLA-projecting cholinergic neurons were reactivated (engram) in the Training + Recall group, while only ~4% of BLAprojecting cholinergic neurons are reactivated in the home cage group (see new Figure 4). These data, combined with the results of silencing BLA-projecting cholinergic neurons, show that BLA-projecting cholinergic neurons are indeed part of the cholinergic threat memory engram.

7) For most experiments, significantly more raw data should be shown (e.g. raw example traces for GRAB-ACh3.0), and also brain section images for almost all experiments. Significantly more raw data should also be shown in the Main Figures. Please see Reviewer # 1, "Specific points relating to data presentation and statistical analysis", for details and specific examples."

We have revised all figures and added additional data to both main figures and supplemental figures to include more raw data to accompany the quantified data. In addition, we have provided additional detailed information on our statistical analyses (both in Figures, legends and in the new Statistical reporting table). Almost all plots now show individual data points. We have extended the GRABACh3.0 traces to show the full recording intervals (shown as average plots with SEMs and as individual traces). Specific changes include:

Figure1/Figure 1-Supplements1-3- we have changed the box plots to scatter plots.

Figure 1-Supplement1C/ Figure 1-Supplement 2D/ Figure 1-Supplement 3B- we have added the averaged fiber photometry trace over the full duration of the session.

Figure 1-Supplement 4C- we provide traces from individual mice showing responses to naïve tone vs. the conditioned tone at tone onset. And in panel B we show the averaged response for the full 28s (last 2s of naïve tone have the shock response and have been removed from the trace to preserve the scale of the y-axis to see tone responses).

Figure 1-Supplement 4A- we provide both raw images and atlas images showing mapping of fiber tip placements and GRAB_ACh3.0_ expression in the BLA.

Figure2B/3B- we show a schematic that delineates the region imaged and show both low and higher magnification images.

Figure 3-Supplement 1- we show a time-resolved freezing plot for ADCD-DREADD recall experiment and clearly delineate the time bins used for analysis presented in the main figures.

Figure 5B/C- we show a schematic delineating anatomical orientations.

Figure 5-Supplement 1A- we show images of GFP expression across the A-P extent of the BLA showing spread of the viral injection.

Figure 5-Supplement 1B- we show images of Fos IHC following recall from home cage, sham and cav-injected mice with clozapine delivered during training across the A-P extent of the BLA.

Figure 5-Supplement 1C- we show an atlas overlay with a low mag image of the amygdala with Fos IHC from control and cav-injected mice following recall with silencing during recall.

Figure 6-Supplement 2A- we provide low and high magnification images of ChAT+ neurons expressing Fos across the A-P axis of the NBM/SI_p_ between high and low responders.

Figure 6-Supplement 2B- we show hM4Di-mCherry expression across the basal forebrain following injection of CAV-DIO-hM4Di-mCherry in the BLA of Chat-IRES-Cre mice.

Figure 8-Supplement 1A- we show low and high magnification images of ChAT and FosGFP labeled cells in the hDB, VP/SIa, and NBM/SIp following predator odor exposure.

Reviewer #1 (Recommendations for the authors):Specific points relating to data presentation and statistical analysis:1) The reporting of statistical data (N's, p- values, and the test used) is a bit cryptic. In general, the authors seem to report these values in the Figure legends. This, however, has the advantage that the reader has to constantly jump between the results text and legends text if he/she wishes to check the values for a given experiment. I would therefore suggest that the authors report all values of statistical tests in the results text (N's, p- values, and the test used).Also, in some instances, N-values seem to be missing from the legend texts. Please check all legends / Results text accordingly.

We have now included a separate statistical reporting table which details sample size, statistical test, and p-value for each comparison reported in the manuscript. In addition, these details have been included in the figure legends for reference along with the corresponding figures. We have also included p-values of the statistical comparisons directly in the Results section.

2) Figure 1:– Traces of GRAB-ACh3.0 should be shown more extensively. In Figure 1B, only the onset to the CS is shown – the response along the entire tone (30 s) should also be shown.– Furthermore, the trace in Figure 1B seems to be an average over all mice. In addition, traces from individual mice should be shown, to allow an estimation of the S/N ratio of these measurements.– In the quantification (Figure 1B, right), it would be easily possible to show the individual data points (there are 11 mice). I did not understand what the leftmost quantification tells us ("Pre-tone"). Please explain how this was calculated, and inasmuch it is relevant.– What is the unit "Z%deltaF/F0"? In my view, this should either be Z-scored or else, deltaF/F0 – please clarify.

Please find our response to each individual point below:

1. We have now added the full GRABACh3.0 traces showing the average +/- SEM throughout the tone (Figure 1-Supplement 2D and Figure 1-Supplement 4B) and tone-shock pairings (Figure 1-Supplement 1C).

2. To aid in estimation of the S/N we have now included individual traces of data shown in Figure 1D in Figure 1-Supplement 4C. In addition we have included individual responses for the tone only controls (3X tone presentation on Day 1 without shock followed by tone presentation on Day 2) during naïve tone (Tone 1) presentation and 24h tone presentation in Figure 1-Supplement 2C.

3. We have now included individual data points (biological replicates) for all figures. We define the period prior to stimulus onset as the “pre-tone period.” This can better be referred to as baseline and has been amended accordingly in all figures and in the text. In Figure 1C you will note the baseline and tone-response period are indicated by a black bar above. We use 1s prior to tone onset for baseline measurements and the 1s following tone onset for our tone measurements. To combine our ‘slope’ and ‘max’ metrics and better represent the data, we have now reanalyzed our ACh release data to compute the area under the curve (AUC) at baseline or in response to the tone. Please note the AUC metric has now been incorporated throughout Figure 1 and Figure 1-Supplements 1-3.

4. Our GRABACh3.0 data is displayed as a z-score of the change in fluorescence over baseline fluorescence (deltaF/F0) over time. We have labeled the axis of each figure accordingly as “z-score X time”.

3) Figure 2:– the authors should extensively show images of brain sections. It is curious to see that the entire paper doesn't show a single image at low magnification, which shows the investigated basal forebrain areas, and stained neurons in there, in their anatomical context. This would allow the reader to put the images into context, and to roughly validate whether the analyses by the authors (e.g. counted cells) make sense. For example, for Figure 2A, please show an additional low mag image of the corresponding brain area, that shows this brain area (NBM/SIp) in the wider context. The actual image can then be shown as an insert.– In the plot of Figure 2B, a tick at the value of 0 is missing – please add this (this also applies to Figure 2C, 4B, 4C, 5A, right, 5B, right, 5C, right, 7A, 7D, S6B, S6C, S7A-C – please add all ticks at zero). Adding the tick at zero will help the reader eyeball the values of the data points.– In some plots, the y-axis doesn't start at 0, but at an artificially chosen higher value. This increases "graphically" the effect size; however, it doesn't allow the reader to look at the data in the most unbiased way (indeed, some Journals do not allow this practice) (see Figure 7C, Figure S6A, right). Please newly plot the data from zero.– The authors did not report in the Statistics part, which criteria they used to add "stars" onto the plots. For example, Figure 2B has *** (3 stars), but the reported p-value is p = 0.0017. According to the criteria used by many authors, p < 0.05 (1 star), p < 0.01 (two stars), and p < 0.001 (three stars). Thus, Figure 2B should have 2 stars, but not 3. Please explain the criteria you used, and please validate all star symbols according to the criteria you used.– What is the unit of the cell numbers plotted in Figure 2B, C? Were these indeed absolute numbers as the axis labelling seems to suggest, or was there a normalization for the analyzed area? Please comment.

1. We have included additional supplemental figures to accompany Figures 1, 5, 6, and 8 that include raw images of IHC data for added transparency. We have also added schematics showing the exact location where analyses were conducted for Figures 2 and 3.

2. We have added tick marks at 0 for all plots.

3. We have updated all our plots to start at 0.

4. The criteria used for significance is as follows: * ≤0.05, **≤0.01, ***≤0.001, **** ≤ 0.0001. All plots have been checked to ensure consistency.

5. In Figure 2, we report the average number of double labeled (mCherry+ and GFP+) cells within each condition from slices at bregma -0.7/-0.8mm. This value is not normalized to area since the NBM/SI_p_ lacks easily defined boundaries.

4) Figure 3:– While the freezing data is shown at a quite high time resolution for the training day (Figure 3B), for recall (Figure 3C), only excessively binned data are shown. It would be advantageous for the reader to see the data at the same time resolution, in a quasi-continuous trace, to evaluate the effects also of silencing.

We have provided ‘quasi-continuous’ traces for both the training and recall sessions from Figure 3 in new Figure 3-Supplement 1A/B. Above the tracers we have indicated tone/shock onset, and the time-windows that were used for analysis of the recall data (pretone vs. recall tone response period). The binned comparisons of these data during the training and recall sessions are shown in Figure 3C/D.

5) Figure 4:– There are several issues with the definition of high and low freezers, and with the analysis in this Figure.– Hig+low freezers: The Methods state "High responders were defined as those mice that exhibited at least a 10 percentage point increase in % time spent freezing in the bin during the tone from the bin immediately pre-tone (e.g. Pre-tone freezing 10% to tone-induced freezing of {greater than or equal to}20%)." However, the Results text speaks about a "3-fold increase" (l. 235) – these are different criteria, please explain.– Related to the above, the y-axis labeling in Figure 4B is unclear – is this the difference in freezing (as the Methods would suggest) or a fold-change (like the Results would suggest)? Also, the 3-star symbol for p = 0.005 seems not justified, should be 2 stars according to often-used criteria (see also above).– l. 238 states: "There was no difference in mCherry expression in 'Low Responders' compared to the home cage group (fold change ~ 1),"– this should be tested statistically.– I don't see that the freezing data in Figure 4D (y-axis) shows a "positive correlation" with the number of Fos-labelled cells (x-axis). Indeed, looking at the high responders only, the data falls very close to a flat line. Please explain how the correlation was analyzed – this was probably done by fitting all data, i.e. also including the "low" responders (?). The obvious absence of a correlation for the "high responders" alone (black data points) is worrisome.– In Figure 4D, the location of the data points (y-axis, freezing; about -10 – 0%) does not seem to correspond to what is plotted in Figure 4B (~ 2%; please explain).– Figure 4F, and Figure S5: The result about the location of labelled cells along the a-p axis of the brain should be documented with a series of original images, both at low and high mag (see general comments, above).

We have amended this figure significantly to state our point more clearly. Please note: this is now new Figure 6.

1. We have added two panels of data into Figure 6 to better explain our responder criteria. In Figure 6A/B, we stratified the freezing response of animals into two groups: High vs. Low responders. High Responders were defined as mice who showed a >10 percentage points increase in time spent freezing in response to the tone compared to the pre-tone period. In Figure 6A, the full time-course of the recall trace is shown to highlight the difference in response to the tone between High and Low responders. In Figure 6B, the dotted line at 10% marks the criteria used to stratify the two groups. The methods section has now been edited to clearly state the criteria to match the figure (Page 43 Line 1231-1239). This is also described along with the associated figures in the Results section on (Page 11-12 Lines 329-343).

2. We have now presented this as the difference between pre-tone and tone period according to the criteria in Figure 6A. The statistical testing is now done and shown on the quasi-continuous trace of recall. The details of post-hoc tests comparing each time bin to the other can be found in the statistical reporting table.

3. We have added a grey shaded area to Figure 6D to indicate the range of ADCD+ neurons captured in home cage animals (ADCD+ neuron counts in each home cage animal normalized to average of all home cage animals). We have included a formal statistical comparison between home cage and low (home cage vs. low, fold change ~ 1, p>0.99) or high responders (home cage vs. high, fold change ~3, p=0.0121) in the associated figure legend and in the Results section on Page 12 Lines 350-353.

4. We have removed the correlation plots and analyses from this figure. Instead, we believe our data are consistent with a threshold effect, whereby mice with ~2 fold increase in activation of cholinergic neurons results in the freezing behavior distinguishable from baseline freezing. Additionally, we find that this “critical mass” of activated cholinergic neurons is localized to a specific region within the anterior NBM/SI_p_ of the basal forebrain.

5. We have added raw images of ChAT and Fos IHC following recall in Figure 5Supplement 2A from a High and Low responder mouse to supplement the schematic shown in Figure 6E. Quantification of counts of ADCD+ cells captured during recall across the A-P extent can be found in Figure 5-Supplement 2C.

6) Figure 5:– The location of back-labelled cells should again be amply documented with several low and high mag images from all analyzed areas (hDB, VP/SIa, NBM/SIp).– 3 stars in Figure 5B might be 2 stars.

1. We have provided injection site mapping for data presented in Figure 5A in Figure 5-Supplement 1A. Images of CAV back-labeled cells are provided in Figure 6Supplement 2B across the hDB, SI_a_, and NBM/SI_p_.

2. We have replotted Figure 5b’’ (previously 5B) data as biological replicates rather than technical replicate. The significance value has updated accordingly in the figure as well in the text (p=0.0286, *).

7) Figure S4:– was the data in Figure S4C a further analysis of data in Figure 2B? If yes, why are the animal numbers seemingly different between the two datasets?

We have now replotted and analyzed all our data by biological replicates and have rectified this discrepancy. Figure S4C can now be found as new Figure 6-Supplement 1A.

8) Figure S5:– The legend doesn't state the N-numbers.– The box plot should be explained (meaning of the boxes). Even better, individual data points should be shown instead, if this is feasible.

1. We have now included sample sizes in the figure legends and in the statistical reporting table.

2. This is now Figure 6-Supplement 2C. We have converted this plot to individual data points as suggested. The data is described in the Results section on Page 12 Lines 363-366.

9) Figure S6:– For the data in A, again example images at low and high magnification should be shown …– For the images in B, please indicate lateral-medial, and dorsal-ventral directions in these images (assuming that these are coronal sections?).– The data in C again suggest that there might also be an effect on contextual fear learning (see also above).

1. Newly added Figure 5-Supplement 1A/B shows a detailed A/P mapping of the injection sites and Fos labeling for experiments presented in Figure 5.

2. Figure 5-Supplement 1 images have been additionally labeled with medial/lateral (M/L) and dorsal/ventral (D/V) designations for ease.

3. We find no statistically significant differences in pre-tone freezing between groups (Figure 5B, Sham vs. hM4Di, Pre-tone p=0.0679; Figure 5C, Sham vs. hM4Di, Pretone p=0.0966).

10) Figure S7– The text describing the data in A states "did not freeze to the tone" (l. 305). This is not a scientifically warranted statement, since there seems to be residual freezing. Rather, the data should be tested statistically.– The N-numbers for the data in A, and B should be stated in the legends.– For these and possibly similar experiments, the authors need to show in detailed histological images, for each individual mouse, that the expression of hM4Di has occurred in the expected set of cholinergic BF neurons.

1. We have amended our text in the Results section to note that silencing BLA projecting cholinergic neurons during training (Figure 5B) *blunted the recall induced freezing behavior* (p<0.0001) and subsequent activation of BLA neurons (p<0.0286). Similarly, silencing BLA-projecting cholinergic neurons during recall (Figure 5C) *reduced recall-induced freezing behavior* (p=0.0279) and the activation of BLA neurons (p=0.0317). Amended text in the Results section can be found on Page 9-10 Lines 271-282. Further details of the statistical comparisons can be found in the Statistical Reporting Table.

2. Sample size has been included in the figure legends and in the statistical reporting table.

3. We have included additional raw images to support the data. An example injection site is now included as part of Figure 5A, center panel, along with an example of ChAT+ cells that also have mCherry expression. The distribution of these hM4diexpressing cholinergic neurons is quantified in Figure 5A, left panel. Additionally, images across the A-P axis of the basal forebrain of hM4Di-mCherry expressing cells have been provided in Figure 6-Supplement 2B. We visually inspected each animal in these experiments for bilateral GFP expression in the BLA and presence of mCherry+ cells in the basal forebrain prior to including (or excluding) mice in the analysis. We have provided representative images showing the A/P extent of the GFP expression (injection area) from one animal in Figure 5-Supplement 1A.

11) Figure S8:– I wonder why the firing rate data of the "Training" group in panel H is extremely small (around 5 Hz or less), and smaller than in all other groups. Is there another type of modulation of intrinsic excitability, a down-modulation, after training? Please explain.

We have also been intrigued by this observation. However, we reserve any conclusions regarding this decrease as it is not statistically significant and overlaps with the firing rate from neurons recorded from the home-cage condition. Further experiments will be necessary to investigate whether there is a bonafide decrease in excitability in Fos+ BFCNs following training.

12) Figure S9:– In panel C, could you show the freezing data at a higher time resolution, e.g. the 30s bins also used in Figure 3B?

Please find the requested plot in Author response image 1. The associated summary plot for the recall session is now included in new Figure 5-Supplement 2A.

**Author response image 1. sa2fig1:** 

Reviewer #2 (Recommendations for the authors):– For the flow of the story, I recommend switching the order of presentation where the lack of effect on behavior after silencing the CeA-projecting NBM/SIp neurons (lines 286-301) comes after the data showing the effect of silencing the BLA-projecting NBM/SIp.

We thank the reviewer for this recommendation and have rearranged the results and figures accordingly to address this. The amended results can be found on Page 10-11 Lines 297-314.

– I would also recommend moving figures S7A and S7B into the main figures as they provide important evidence for the author's point.

We agree that this set of experiments is critical to the main point of the manuscript. We have taken this suggestion and have now included these experiments into a significantly modified main figure, Figure 5 (Panels B-C).

– Given that the authors don't see an effect of clozapine on memory when inactivating during the consolidation period (Figure S7C), it might be worthwhile to discuss why this might be the case, and the effects of cholinergic input starting cascades of neural activation in the BLA during training.

We agree the possible function of ACh during consolidation is of interest, but feel that this requires a more complete experimental investigation that is beyond the scope of the present study, and as such we have removed prior Figure S7c as too preliminary.

– Overall, there are missing Bregmas and mapping of placements in several places (e.g. Figure 1A – Bregma and mapping of placements/spread of the virus across animals, it appears that the recordings are relatively anterior in the BLA; Bregma in Figures 5, S3C, 6SB, S6C – not clear how much of a CeA there is at this location, 7D; a zoomed out section of where the example data in figure 7C is coming from).

We have now included raw images and schematics to provide transparency to the data. These have been added to Figures 2, 3, 4, and Figure 1-Supplement 4, Figure 5-Supplement 1, Figure 6-Supplement 2, and Figure 8-Supplement 1. In new Figure 1-Supplement 4A we have included a schematic of the relocalized fiber placements from fiber photometry experiments presented in Figure 1. In Figures 2 and 3, we have now included low magnification and high magnification images of ADCD labeled neurons within the NBM/SI_p_. An A/P extent of the Fast Blue injection locations for the newly added Figure 4 is included as panel within Figure 4A. We have now included raw images detailing the AP extent of the evaluated BLA as related to Figure 5 data, shown in Figure 5-Supplement 1A/B. In Figure 5-Supplement 1C, we have included low and high magnification images to localize the CeC within the image. In Figure 6-Supplement 2 we provide images across the A/P extent of the basal forebrain to show where activated cholinergic neurons are clustered in High vs. Low responders. Finally, in Figure 8-Supplement 1, we show raw images of across the basal forebrain of activated cholinergic neurons in response to predator odor. We believe these additions have substantially strengthened the manuscript.

– It would also be helpful to indicate the area where the signal fluorescence is being calculated in traces shown in figures 1 and S1.

We now indicate with a black bar above our traces in Figure 1 the period that is being quantified for baseline (1s before stimulus onset) and tone response (1s following tone onset). We have also included additional text in the methods section to clarify this on Page 45 Lines 1288-1291.

Reviewer #3 (Recommendations for the authors):1) For the 'engram' data in Figure 2, the logic of the double labeling is a bit confusing as described and it's not clear, from a simple reading of the paper, how the authors achieve double labeling of encoding (training activated) and recall activated cells. After digging around in the Methods, Figs/legends I was able to figure it out, but it was difficult initially. The authors should state this in a simple way in the main text along with a description of how this works at the level of transgene expression at different time points.

We have updated the text to clarify our double-labeling strategy (Page 6 and 7 Lines 170188). We have also added a schematic in Figure 2-Supplement 1B accompanied by detailed text in the figure legend (Page 28 Lines 807-817) detailing how the ADCD-engram labeling works. In brief, we deliver the ADCD construct in the basal forebrain of Chat-Cre X FostTA-shGFP mice. In this manner, the virally delivered ADCD construct can only be expressed in cholinergic neurons, in the absence of doxycycline (DOX OFF). Once expression initiates, it is maintained even in the presence of Dox by a dox-insensitive tTa, thereby resulting in permanent labeling. Intercepting the peak of a shGFP signal (for example 2.5h following a second behavioral assay) allows for identification of cells that express mCherry (activated during DOX OFF) and cells that express GFP.

2) Some important controls are missing from the 'engram' causal manipulation study (Figure 3) which makes it difficult to definitively support the authors' contention that the captured cells are in fact 'engram' cells. The established definition of engram cells refers to those which store memory and are activated during learning and memory expression. While the authors do show conditioning-induced changes in CS activation in NBM-ACh cells, they don't have the necessary controls for their causal manipulations to rule out other non-engram factors which could account for their results. For example, it is possible that the captured cells are simply activated by auditory cues (even prior to learning, the immuno approach in Figure 2 may not be sensitive enough to detect this) and that they are transmitting CS-activated ACh signals to BLA which are necessary for memory expression. It is also possible that inhibiting a small number of NBM-ACh neurons (not restricted to 'engram' cells) reduces memory expression (early engram studies by the Josselyn lab controlled for this). The authors could address this with further experiments such as including 1) a control group in which cells are captured (off-dox) during a habituation session with CS presentation prior to on-dox learning and 2) using a dilute virus to manipulate a similar number of cells as in the engram group. Alternatively, they could present a more careful discussion of this issue and potentially avoid the use of the term 'engram'.

Throughout the manuscript, we have evaluated the cholinergic response to the tone using 1) GRABACh 3.0 to quantify changes in ACh levels in the BLA and 2) using an IEG response to tag and identify cholinergic neurons that were activated under specific behavioral conditions. Using either approach, we have not seen a significant increase in activation or ACh release in response to the unconditioned tone (Figure 1C-left, 1D-left, 2C/D, Figure 1-Supplement 2, Figure 1-Supplement 3, Figure 6-Supplement 1A) unless it has been paired with a footshock (see Figure 1C-right, 2C, 3D).

The BLA receives a majority of cholinergic input from the NBM/SI_p_ with additional cholinergic input from the VP/SI_a_. We evaluated the contribution of each of these populations to the freezing response to the conditioned tone. We find that silencing the NBM/SI_p_ results in a decrease in freezing to the conditioned tone (Figure 5-Supplement 2A), while silencing the VP/SI_a_ population does not affect the freezing response (Figure 5Supplement 2B). The lack of effect with VP inhibition suggests that silencing a random, small number of BFCNs, or decreasing ACh release in the BLA alone is not sufficient to affect memory recall.

3) The authors suggest that the ACh 'engram' cells participate in the expression of learned defensive responses through projections to the BLA. However, they have not shown this definitively. To do this would require testing the effects of the inactivation of ACh-engram neurons which project to BLA on learned freezing responses. This may be technically challenging, but one approach could be to use the ADCD technique they've developed and inactivate terminals of captured cells in the BLA (chemogenetically or optogenetically). At a minimum, the authors should examine whether retrogradely labeled ACh cells projecting to BLA are Fos activated during a recall session (compared with appropriate controls).

In new Figure 4, we retrogradely label cholinergic neurons that project BLA using the retrograde tracer FastBlue in [Chat-IRES-Cre X Fos-tTA/shGFP] mice. Simultaneously, we injected our ADCD virus into the NBM/SI. Cells were captured off-dox during training. Mice were sacrificed following recall to capture GFP expression in recall-activated cells. We quantified BLA-projecting cholinergic neurons (Fast Blue+) that were activated during training (ADCD+) and became reactivated during recall (GFP+). We find that ~20% of BLAprojecting cholinergic neurons are reactivated (engram) in the Training + Recall group, while only ~4% of BLA-projecting cholinergic neurons are reactivated in the home cage group. These data show that BLA-projecting cholinergic neurons are indeed part of the cholinergic threat memory engram.

4) In Figure 5, the authors report a reduction in BLA Fos-activated neurons with chemogenetic inhibition of BLA-projecting ACh neurons. However, this experiment requires a control group without training (or, ideally, with unpaired CS/US training) to demonstrate that the inhibition is affect learning-induced activation of BLA neurons. While others have reported this using other paradigms previously, it is still important to show here using their own preparation.

We have now included images across the A/P axis in the BLA for Fos expression in home cage animals alongside Fos expression following recall in animals that had a sham surgery or those that were injected with CAV2-DIO-hM4Di with cholinergic inhibition during training (Figure 5-Supplement 1B).

[Editors’ note: what follows is the authors’ response to the second round of review.]

The manuscript has been improved but there are some remaining issues that need to be addressed, as outlined below:We thank the authors for their response to the reviewers' previous suggestions. Many of the findings have been strengthened through the addition of new experiments and analyses. However, there are remaining concerns about the statistical approaches used and low population size in some experiments which should be addressed. We suggest using the 'eLife Transparent Reporting Form' as a guide. Related to sample sizes, power analyses or clear demonstrations of existing standards in the field should be used to justify animal numbers. If necessary, further animals should be added to experimental groups (e.g. the differential role of NBM/SIp and VP/SIa in learned vs. innate threats) or, as suggested by reviewer 1, data with low n could be removed if it does not seriously undermine the conceptual advance of the paper. In addition, the strong focus on the activated basal forebrain cholinergic cells as being part of the threat memory engram is not well supported. The authors should moderate the claims related to this aspect of the paper.Furthermore, the title and/or abstract should provide a clear indication of the biological system under investigation (i.e., species name or broader taxonomic group, if appropriate). Please revise your title and/or abstract with this advice in mind.

We have addressed these concerns in the following manner:

1. We excluded the data related to the lack of engagement of the VP in cue associated-learned threat from the manuscript, as suggested, as this does not compromise the main claims of the revised manuscript.

2. We removed the use of term “engram” from the title, abstract and results text. We do discuss the definition of “engram” in the Discussion section.

3. We tempered claims regarding our demonstration of a cholinergic engram throughout the text and now consider arguments pro and con in the context of the literature more carefully in the Discussion section.

Re: clarifying species used in title and/or abstract:

– Line 42 (Abstract) now reads: “Using a genetically-encoded acetylcholine (ACh) sensor in mice…”

– Lines 54-55 (Abstract) is also modified to read: “…specific types of memory within the cholinergic basal forebrain of mice.”

Reviewer #1 (Recommendations for the authors):Specific points (in order of appearance)1) l. 156/157 "when it was not explicitly paired with a shock" (data in Figure 1S3D).– Please explain in the legend the exact temporal sequence between shock(s) and tone(s) used for this "non-explicitly paired" protocol (or does this refer to the "shock-only" protocol?).2) Figure 1 S3C, D: The legend to this Figure states that here, the results from a "shock-only" training are displayed for the subsequent "recall" session.First, the authors should also show the freezing response for the standard group undergoing standard fear learning (3x tone-shock pairings); this is important for a comparison to the "shock-only" training group.Second, the Results text states that Figure 1S3D shows the ACh response when "it was not explicitly paired with a shock".– Is this the same as the "shock-only" protocol described in the legend? Please clarify, and make changes in the manuscript as appropriate.

We provided schematics as well as text to explicitly delineate the behavioral protocol and to describe each result obtained. Please note that the full context of the phrase that was excerpted by the reviewer "it was not explicitly paired with a shock” reads as follows:

“To verify that the increases in ACh release were indeed specific to the tone-shock association and not due to generalization from prior shock exposure, we also subjected mice to 3 shocks (day 1) followed by a tone presentation 24 hr later (day 2) (Figure 1- Supplement 3A). While mice demonstrated freezing behavior during the session on day 2, there was no significant increase in freezing behavior to the 24h tone presentation (Figure 1-Supplement 3C, p=0.2418). There was no increase in ACh in response to the tone when it was not explicitly paired with a shock, confirming that the changes in ACh release were indeed associative (Figure 1-Supplement 3D; baseline (pre-tone, day 2) to 24h tone (tone presentation, day2): p=0.7272).”

When read within the context of the entire paragraph it seems clear that we are referring to the shock alone training, followed by a tone alone recall. The grouping of these sentences into a single paragraph, focused on discussing a single figure (that starts with a schematic of the experimental design) is as explicit a presentation of the protocol and the data obtained as we could put together. However, we welcome further clarification from the reviewer.

3) Figure 1 S4 was not called out from the Results text; this should be changed i.e. what was done in this Figure, should be (briefly) mentioned in the Results.

We had included these additional plots of the data and schematics in response to reviewer request during the last review. As stated, these plots refer to the data that has been described in the text (Figure 1). We believe that the figure legend accompanying Figure 1 S4 clarifies the contents of this figure.

See lines 125-126 where we reference Figure 1 S4:

“Given this, the first question we asked was whether acetylcholine was released in the BLA during associative threat learning (Figure 1 and Figure 1-Supplements, S1-S4).”

4) The use of star symbols in Figure 2C, in conjunction with what is stated in the corresponding Results text, is misleading and tends to over-interpret the statistical significance of this data set.

We use a single star to denote p≤0.05 (as defined in the manuscript; line 1284, as quoted below and as standard usage in some journals and statistical analysis algorithims). The identified major issue is the use of a single star in Figure 2C to identify a p value = 0.0464, (i.e. ≤0.05.) The use of the star and the accompanying text is just a statement of the p value and is not intended to be misleading or an interpretation.

Also please see Line 1293:

“p-value criteria: * p≤0.05, ** p≤0.01, *** p≤0.001, **** p≤0.0001**”**

Specifically, l. 202 states: "Significantly more double-positive cholinergic neurons were seen following the complete (tone + shock followed by tone recall) associative threat learning paradigm compared to mice that underwent session 1 without shocks (Figure 2C, p=0.0249)."– This is misleading. The Figure legend and statistics table state that the value "p = 0 0.0249" refers to the "overall" significance of the Kruskal-Wallis (KW) test used here, but NOT to the specific comparison cited in the above sentence. For that specific comparison, the p-value was higher ("p=0.0464, Dunn's corrected", see Figure legend).– In response, the authors should spell out the "overall" significance of this dataset (effect of group; p = 0.0249; KW) in the Results section, followed by the pertinent p-value for important specific comparisons (in this case, p = 0.046 for the comparison between the "tone-tone" and tone+shock – tone" groups). The n-values for these pertinent comparisons should also be given in the Results text.Furthermore, in this dataset (Figure 2C), the post-hoc test for the comparisons between the home cage (HC) and standard fear conditioning group is NOT significant (p = 0.72). This value should be indicated by a "bracket" symbol in Figure 2C, and the meaning of this non-significance should be discussed in the manuscript.5) The p-values, n-values, and the statistical tests applied, should be reported together either in the figure legends or in the Results text (please choose one section to use for reporting these statistics, in addition to the statistical table) throughout the entire manuscript (see above point 4, and see also my previous point "Data Recommendation 1").

As suggested, we had included this information in the figure legends for which quantitative analyses was assessed.

As the reviewer directs, these statistical data should be reported “either in the figure legends or in the Results text”. We had complied with this request, presenting the data in the figure legends and provided a detailed table for reporting statistics for full transparency.

6) Figure 2D:l. 210 states:"All three conditions showed few reactivated neurons and no differences between groups(p=0.9471)"– The finding that all these 3 control groups are not significantly different from each other, is not surprising and does not go to the point of the control experiment. Rather, it should be tested statistically whether the number of re-activated neurons in these three conditions is significantly different from the one after standard fear conditioning (the data in Figure 2C, right column).

We have performed the requested analysis and the updated statistical metrics have been provided in the statistical reporting table.

7) I think the title of the Sub-header:"Reactivation of cholinergic neurons activated by training is required for learned behavioral responses" (l. 214) is wrong, it should read "Activation" instead (and then replace "activated by" with "during").To my understanding, the chemogenetic silencing experiments in Figure 3 test the role of activated cholinergic neurons but do not suppress the "re-activation" of neurons (during recall…), since ADCD expression of hM4Di is driven during the training session.

The design of this experiment, as illustrated in the schematic, was to explicitly assay the effect of selective inhibition on neurons that were activated in training and then subsequently re-activated by recall. We have used a classic experimental paradigm to do so (Josselyn and Tonegawa, 2020. Science). That is, we expressed the inhibitory hM4Di construct in cholinergic neurons that had been activated during training (Activity dependent Cre dependent virus, dox off during training), We then use the DREADD ligand, clozapine, right before and during recall to selectively prevent these previously activated neurons (activated during training) from being activated again (hence, re-activated) during the recall session. The schematic in this figure and the figure legend attempt to clarify this further as follows:

“A. ADCD-hM4Di was injected into the NBM/SIp of Chat-IRES-Cre x Fos-tTA/shGFP mice. Two weeks later animals underwent training on regular chow (Dox chow removed 24 hr prior to training session) to allow hM4Di.mCherry to be selectively expressed in training activated cholinergic neurons. Three days later, recall was tested in Dox on conditions. Clozapine (CLZ) was injected 10min before the recall session to activate the inhibitory DREADD, hM4Di specifically in previously activated cholinergic neurons.”

8) In Figure 3 S1B, the average trace of time-resolved freezing is shown, for the same data set as the one in Figure 3D. However, the number of mice included in the time-resolved display was lower (n = 5 and 6) than in the display of Figure 3D (n = 7 and 8). This must be homogenized, i.e. all data used for the statistical analysis in Figure 3D (n = 8 and 7 mice) should also be shown in a time-resolved fashion for Figure 3S1B.

The reviewer is correct in noting that two mice are missing from each group in the population data, time resolved plot. The reason for this omission is that those mice were part of a cohort, where the video aspects of the data were temporarily stored on a drive that was damaged in a major flood that affected multiple labs within building 35 (NIH; Bethesda) last December. As such, we could not go back to generate the specific time-resolved plots for those 4 mice. We provided the data for the remainder of the mice to aid in clarity, as requested. We also reanalyzed our data by removing the missing mice from the summary data plots and this did not alter the statistical validity of our findings as shown in Author response image 2.

9) l. 273 states:" We injected cav.hM4DiBLA or sham control mice with clozapine (CLZ) 10 min prior to initiating cue-conditioned threat learning"– Please add here that these were IP (?) injections, for a quick reference for the reader. Indeed, this information should possibly also be added to the Methods (I did not find it upon a rapid check).

A search of the text reveals at least 4 instances in which we have detailed that the mice were injected i.p. Lines: 1126, 1132, 1169, 1183. In addition, we have now added another note regarding this procedural detail in the Results section at:

Line 215 “(0.1 mg/kg; injected intraperitoneally (i.p.)).”

10) The authors should control for the possibility of leaky expression of hM4Di, following presumed Cre-dependent retrograde expression from a CAV2 virus after injections in the BLA (Figure 5). Because, as I assume, CLZ was not injected into the basal forebrain (BF) but rather IP (see point 9 above), a leaky expression of hM4Di beyond the cholinergic BF has the potential to strongly change the interpretation of this experiment. Thus, the authors have to show, first, whether hM4Di-mCherry is expressed at the injection site in the BLA itself (this would lead to a direct inhibition of BLA neurons). This should be done by showing example images of hM4di-mCherry fluorescence on the level of the BLA, and by analyzing the number of hM4Di-mCherry+ neurons in the BLA over the entire dataset. Second, the authors should show extensive raw example images of the BF at different a-p locations (and of further brain areas in a zoomed-out fashion), in an attempt to exclude the possibility that hM4Di might also be expressed in non-cholinergic neurons (ChAT could be visualized by IHC). Only a non-leaky expression of hM4Di-mCherry, essentially limited to ChAT-positive neurons in the BF, would support the conclusions of the authors.

Figure 5A shows that the expression of hM4Di is wholly restricted to ChAT+ neurons. We have no reason to suspect that the Chat-IRES-Cre mice used in this study display leaky cre expression. We refer the reviewer to the Jax Labs page for this mouse line, with particular reference to the details posted there relevant to the ways in which the line we have used was created to address a previous cre leak issue - https://www.jax.org/strain/031661.

Figure 6 Supplement 2 shows several zoomed out images of the BFCNs in response to this same request made previously by Reviewers.

11) On l. 298, the results of a control experiment is reported, which is not documented ("data not shown"). Some journals do not allow the statement "data not shown". The authors should document the control data set, or else remove this statement from the manuscript. See also l. 441, another "data not shown", which should be similarly documented or else removed.

We have now removed this phrase from the manuscript. We apologize for the prior inclusion of “data not shown” as we weren’t aware that *eLife* prohibits the inclusion of such references.

12) Data in Figure 5C.When checking the p-values from the legends/statistics table, I was puzzled by the fact the comparison of the two "grey" data sets has a reported p-value of < 0.0001, although many of the data points between the two groups overlap. On the other hand, the comparison between the "sham" group and hM4Di group at the "pre-tone" time had a p-value of 0.096, although the overlap of the data points is only partial. Please comment, and validate the underlying tests.

These data were analyzed, as is appropriate, using a repeated measures two-way ANOVA (as noted in the figure legends and in the statistical reporting table). Given the design of this experiment and the exigencies of the statistical tests used, we think that this analysis is more quantitatively rigorous than the readers’ visual inspection of overlap between populations in a scatter plot. Please see the plot Author response image 3.

**Author response image 3. sa2fig3:** 

13) Related to point 12 above, there is a strong tendency that freezing before the tone is lower in the hM4Di group as compared to the "sham" group. Although the authors reported a p-value of 0.096 (and therefore, "not significant") for this comparison (Figure 5C), this p-value should nevertheless be verified. Because the trend is seen in this data set, as well as in several other ones (Figure 5B and Figure 3D), the authors should discuss the possibility that chemogenetic suppression of ACh release might, in addition to the effects studied here, also reduce contextual fear learning (see also my previous "Specific comment 2").

We have used standard, and in our opinions correct, statistical tests to evaluate the quantitative data in this study and therefore feel that it would be unscientific to discuss mechanisms underlying results that do not differ significantly as detected by these tests.

14) Still related to the data in Figure 5C:The time-resolved display of tone-evoked freezing during the recall day should be shown both for this data set (Figure 5C), and for the data set in Figure 5B, similarly as was done above for the data in Figure 3D (see panel Figure 3 S1B). This would also be helpful to gain insight into the p-values resulting from ANOVA / KW testing (see above, point 12), which were probably performed on similar time-resolved data sets (because the effect of "time" was always tested).

We can provide these plots additionally as a supplemental figure. However, our summary plots in the main figures are the commonplace reporting standard for these types of data. We feel that adding still more supplemental figures and analyses that do not change the conclusions of the study would unnecessarily extend this manuscript length. The manuscript has already seen a significant “ballooning” of supplemental figures to show each individual data set obtained, in response to prior requests of the reviewer.

In case it is the consensus of the editors and other reviewers that presentation of each individual time course points is required, we provide them in Author response image 4 for inclusion at your discretion.

**Author response image 4. sa2fig4:** 

15) l. 299 states:"Differences in recall-induced Fos expression between sham and cav.hM4DiBLA mice were maximal in rostral portions of the BLA (between bregma -0.8mm to -1.4mm) (Figure 5-Supplement 1B)"– First, the analysis in Figure 5 S1B seems to have been done in only N = 1 mouse, which is problematic (please confirm the n-number, and state it in the Results text and in the legend to that Figure).Second, I actually fail to see an a-p gradient as claimed by the authors (higher in anterior than posterior). If one looks at the number of Fos+ cells in the hM4Di group (red data points), then it seems that hM4Di suppresses the number of Fos+ cells, in relative terms, stronger at more posterior levels (e.g. reduction by > 80% at -1.2 mm) than at anterior levels (e.g. reduction by < 40% at -0.8 mm). Wouldn't this suggest that opposite from the author's interpretation, MORE cholinergic neurons are specifically (in a hM4Di-dependent manner) activated posteriorly than anteriorly? Please comment.

We have removed the plot shown in Figure 5 S1B as it is not critical to the point of the analysis. To address the reviewer’s point, anterior BLA is located from bregma -0.8 to ~-2.0 after which posterior BLA begins to predominate (these are referred to in the Paxinos mouse brain atlas as BLA and BLP respectively). In our studies, DREADD mediated suppression of Fos activity was seen in the regions analyzed and presented but not beyond those bounds. This observation was consistent with a previously published report showing the segregation of negative valence processing BLA neurons, and as such we had included that plot in this manuscript.

We would like to note that we make no comments or claims regarding the relationship between BLA anterior to posterior anatomy and the basal forebrain anterior to posterior anatomy. We are unsure what led the reviewer to this conclusion but would be willing to address it explicitly to aid in clarity if indeed this is deemed a major issue as noted by the reviewer.

16) Visual inspection of the data in Figure 5 S1D strongly suggests that the data in the control ("sham") group (n = 9, grey data) are actually the same as the data for the sham group in Figure 5C. In Figure 5C, retrograde expression from the BLA was studied, whereas in Figure 5 S1D, the effects of retrograde expression of hM4Di from the CeA were studied. If the experiment was originally set up for 3 groups of mice (thus, sham control – CAV2-hM4Di in BLA – CAV2-hM4Di in CeA), then this should be reported as such, the data should be shown in one panel, and the statistical comparisons should be done accordingly (i.e. the "multiple comparison" correction would need to be done for 3 groups). If, on the other hand, the CeA experiments were done without proper controls, and were simply compared to the control group from another experiment, then the CeA data should be removed. Please comment, and make appropriate changes in the manuscript.

Yes, in fact we in fact explicitly stated this in the figure legends:

“Note, sham group is the same as the sham group shown in Figure 5C as this experiment was conducted in the same cohort.”

The reviewer is correct in noting that this was a post-hoc identification of CeA injection sites in subset of mice. We are removing this data from the manuscript as the point made is not critical.

17) Figure 5 S2 claims that neurons in the "NBM/SIp" versus neurons in the "VP/SIa" have been targeted selectively with AAV9:DIO:hM4Di-mCherry. This should be documented extensively, by showing expression of hM4Di-mCherry throughout several brain slices in a de-zoomed fashion for an example mouse, and by showing average results from the quantification of the localizations in a suitable manner (for example, by plotting cell density/number along the a-p axis), for all investigated mice. Otherwise, it remains unclear whether the two areas have indeed been targeted differentially.

In our hands, the two injection sites used to target VP vs. NBM result in expression restricted to each region.

Nevertheless, based on comments and concerns raised in “point 19”, we have decided to remove the VP/SIa DREADD data from this manuscript.

18) In Figure 5 S2A, the "triple star" symbol comparing the two control data points might be removed. It is not surprising that the tone recall causes increased freezing in the "sham" group.

We respectfully disagree with the reviewer. While it might not be surprising that tone recall causes increased freezing in the “sham” group, it is important to validate this group as an appropriate control in this experiment. In addition, please note that it was the explicit recommendation of Reviewer 1 in the previous round of reviews that all comparisons are tested for statistical significance.

21) The "fold-change" analysis of expression analyzed in Figure 6D remained unclear to me (sorry if I didn't get it). Please explain this briefly in the Results text, and make sure it is explained in more detail in the Methods. Also, please give details about the viral construct used in Figure 6C (currently it only states "ADCD").

We clarified this in the “response to reviewers letter” submitted alongside the previously revised manuscript. Here is our prior reply:

Reviewer1 Data Recommendation 5:

We agree that our prior presentation was confusing, and we have removed the correlation plots (see new Figure 6 and new Figure 6-Supplement 1). We now use data from a distinct cohort of mice (distinct from the ADCD labeling experiments) that underwent behavioral testing to demonstrate high vs. low responder behavioral performance (new Figure 6A/B). To the reviewer’s point regarding the denominator, counts of ADCD+ neurons from animals that underwent the recall session were normalized to the average of number of ADCD+ neurons in home cage animals in their respective cohorts.

And please note that the figure legend states:

“D. Quantification of change in number of cholinergic neurons activated (ADCD+) in low or High Responders relative to the home cage. The number of ADCD+ neurons differed significantly between Low and High Responders (Mann-Whitney test, p=0.01) (n=3 Low Responder, n=8 High Responder). Grey shading represents the range of fold-change in ADCD+ cells in individual home cage animals relative to the average of all home cage animals (n=5). (Mann-Whitney test, home cage v. Low Responder, p>0.9999; home cage v. High Responder, p=0.0121).”

Also see Lines 326-327: Next, we quantified the fold change in the number of mCherry+ neurons in each group relative to corresponding home cage control mice (Figure 6D).

We believe that this is a clear description, but welcome specific suggestions as to what we need to do to make our procedures and analysis optimally transparent to the reviewer.

The viral construct used has been extensively documented in the text upon its introduction, in a supplemental figure dedicated to it, and in the methods section.

Additionally, Lines 324-325: we labeled cholinergic neurons activated during the recall session with ADCD-mCherry (on dox during training, off dox during recall; Figure 6C).

22) The picture shown in Figure 6E is simply a scheme of the idea that neurons activated in "high responders", in your view, should be localized more anteriorly, but it is in no way a documentation of an anatomical observation. This scheme should be replaced by experimental evidence.

This is indeed, as we mention in the figure legends, just a schematic. We would like to clarify, however, that this is not a hypothesized or speculated distribution but as previously requested, a documentation of our data. We include images showing this A-P difference in activation and quantification of a subset of high responding mice which were provided in Figure 6 Supp2.

23) The data that might relate to the above issue (localization of ChAT+, activated neurons along the a-p axis in "high" and "low" responders), might be contained in Figure 6 S2. However, the data seems to be only partially analyzed, and it is confusing to fully understand how the plot in panel C was derived. First, it was unclear from which group of mice the images in Figure 6 S2A were taken. Please explain in the legend from which experiment these images were taken, which mouse genotype was used (Fos-tTA/shGFP with or without ChAT-Cre ?), and which viral constructs were injected in these mice.

Once again, we urge the reviewer to read the figure legends and text associated with this figure. We provide ample description of how these data were gathered. Specifically:

“A. Representative images of the NBM/SI_p_ at bregma locations -0.8, 1.0, and 1.3mm from a Fos-tTA/shGFP High Responder (top) and Low Responder (bottom) mouse sacrificed 2.5h after recall. Brain sections were stained for ChAT (magenta), and GFP to amplify Fos signal (green). ChAT+ and Fos+ co-labeled cells are marked by yellow arrowheads. Scale bar = 50µm. Insets show magnified images of ChAT+ and Fos+ neurons. Scale bar = 10µm.”

24) The images shown in Figure 6 S2C, should be shown in a de-zoomed fashion for several representative brain images on the level of the BF, and beyond. Such a documentation of the data would represent the control for "leaky" expression mentioned above (best after co-staining against ChAT; see point 10 above).

We refer the reviewer to point #10 above.

25) The 2-star symbol in Figure 7D above the comparison "HC" – Fos+" should probably be 3-stars, not 2-stars (p = 0.0006).

We have now corrected this.

26) The authors should use the less "pushy" and more conventional criteria for using star symbols for statistical significance (i.e. 1 star, p < 0.05; 2 stars, < 0.01; 3 stars < 0.001), and omit the "4 star" symbol and definition throughout. The 1-3 star criteria should be stated in the Methods (I didn't find it). The paper in its current version has only a single "4 star" symbol (Figure 3D), which should be changed to "3 star" according to the above definition, or removed (see also point 18) (see also my previous point "Data Recommendations 4").

We certainly do not intend to be pushy – In fact the statistical analysis software used (Graphpad Prism) provides “4 star” symbols. The convention followed is included in the text as delineated above.

Our provision of the detailed statistical reporting table should circumvent any confusion in this regard.

27) The bracket symbol in Figure 7F is probably wrong. On the left side, it should only point to the "HC" group, but not to both the "HC" and "Fos-" group as it stands now.

No, actually, the symbol in Figure 7 F is not wrong. The Fos+ group differs from both the HC and Fos- group. Please see the excerpt from the statistical reporting table Author response table 1.

**Author response table 1. sa2table1:** 

Max Firing Rate	0.0032
HC vs. Fos-	0.3206
HC vs. Fos+	0.003
Fos- vs. Fos+	0.0034

28) The authors claim that "latency to fire an AP" and AHP might significantly change in re-activated ChAT neurons (Figure 7 S1E, F). However, these effects are somewhat more subtle than the other effects shown in the main Figure 7. For example, the effect of "latency" seems to be driven merely by some outlier neurons in the home cage group with very long latencies (the authors would need to show example recordings), whereas the data from the Fos- and Fos+ groups virtually overlay and the latter comparison is the more pertinent one. Thus, I think it is more cautious to drop this claim.

Indeed, as in standard practice, we show all the data and all of the results of the statistical analyses for all biophysical measures made whether “subtle” or not. These analyses consider the full range and distribution of the data without bias as to normality. Our commitment to presentation of all the features for all the recorded neurons, whether statistically significantly different or not, uses both the main and supplemental figures to provide full transparency. This is considered best practice for reporting biophysical data in general and the results of these single cell recording + post hoc PCR, labor-intensive experiments.

29) l. 427 states:" Next, we quantified the number of cholinergic neurons expressing Fos (Fos shGFP+) after saline or predator odor exposure"– Please explain how long the mice were exposed to the odors.

The figure legend states this:

“A. Fos-tTA/Fos-shGFP mice were placed in chambers containing a gauze pad spotted with either saline or with mountain lion urine (predator odor). Defensive behaviors were monitored for 5 min. Animals froze significantly more in the presence of predator odor than saline (Mann-Whitney, p=0.028).”

30) In Figure 8, the authors seem to analyze the results of a triple-labeling anatomical experiment; i.e. there are neurons that are Fos+; neurons that are back-labelled by fast Blue; neurons that are ChAT+, and all combinations thereof. Thus, to give the reader a better view of the various neuron pools, the authors should report the numbers of all pools of neurons (i.e. the "green", "red", "blue" neurons and all combinations thereof). This can be either done numerically or in the form of a Venn diagram. As it stands, only part of the information is given (Figure 8C, right; which seems to correspond to the percentage of back-labelled, ChAT+ neurons which were also Fos+). However, this is only part of a bigger picture. For example, the images in Figure 8C, and in Figure 8 S1A (middle, right) show also many non-ChAT+ neurons, that were re-activated in the VP/SIa area.

The purpose of this experiment, and indeed the entire study, is to investigate the contribution of different populations of basal forebrain cholinergic neurons to threat responses. At no time do we say or even imply that other neuronal types are not involved as well, but these “non-ChAT+” cells are not our focus. The requested analyses are outside the scope of this study and would not add or take away from our conclusions.

31) What does the dashed line label "Home cage" in Figure 8C, right panel, mean, and how was it derived?

The meaning of the dashed line (which we referred to as dotted) was delineated in the figure legend:

“Dotted line indicates % of Fos+ cholinergic neurons in the home cage group in this experiment.”

32) l. 442:" Nearly the entire subset of BLA-projecting VP/SIa cholinergic neurons were also Fos-shGFP+ (Figure 8C right)."– Another non-scientific statement. Rather, the average plus/minus S.D. of the quantification should be reported. In addition, information about all labelled neuronal populations should be given (see point 30).

Line 409 now adds: “(median 94% ± Std.dev 12.5)”.

33) The retrograde expression experiment of hM4Di from the BLA in Figure 8D is expected to drive hM4Di expression in all cholinergic BLA-projectors, not selectively in the VP/SIa. This caveat in the interpretation should be mentioned more carefully when the results of this section are summarized (lines 451 – 454).

The summary statement referred to is a conclusion based on the entirety of the mapping experiments in the manuscript. It does not refer specifically to a single figure.

Reviewer #2 (Recommendations for the authors):I thank the authors for addressing my concerns, and I'm happy to recommend the manuscript for publication in its current form. There are a few figures that would benefit from additional labeling (outlined below).Please add anatomical references for basal forebrain in figures F1S4, F2S1a, F4c, F5S1a, F5S1b , F6S2a.

We thank the reviewer for their positive recommendation and have added the necessary bregma annotations.

Reviewer #3 (Recommendations for the authors):The authors have addressed many of my previous concerns and now nicely dissociate the role of NBM/SIp and VP/SIp ACh neurons in learned threat memory processing and show that NBM/SIp ACh cells are activated by memory recall and necessary for threat memory expression. However, I still have a lingering concern with their strong emphasis on the role of NBM/SIp ACh neuron (and their projections to BLA) as being memory 'engram' cells. While some of their data are consistent with this idea, they do not test key criteria for defining engram neurons, and the new data supporting a role of NBM/SIp ACh cells which project to the BLA in the threat memory 'engram' is only correlative.Specifically, while there is evidence in support of the NBM/SIp ACh cell population (independent of where they project) in memory engrams (activated by memory cues, necessary for memory expression). There are several criteria for defining engram neurons (see Josselyn and Tonegawa Science 2020) which haven't been met including sufficiency (stimulation should produce memory in the absence of recall cues), mimicry (creation of an artificial engram) and long-lasting cellular or synaptic changes in engram neurons (here they only see excitability differences at 2.5 hr, not at long term memory timepoints). It is possible (even likely) that activity in these neurons modulates/facilitates threat memory formation and expression and are not storing the memory trace themselves.There is even less evidence that NBM/SIp ACh cells which project to BLA being engram cells. The authors report an increase in ACh release in BLA (which isn't necessarily coming only from NBM/SIp) and added new data showing that NBM/SIp ACh cells projecting to BLA are fos activated.While there is data supporting a potential role for these cells in engram processing, it is inappropriate to so strongly insist (in the title, abstract, and throughout) that NBM/SIp ACh neurons, and particularly those which project to BLA, are engram cells. The authors should moderate this aspect of the paper and perhaps broaden the discussion of the potential role of these cells in threat memory processing as they've already nicely done in the 'Cholinergic Modulation of Associative Threat Learning' section of the Discussion.

We thank the reviewer for their positive comments and thoughtful recommendations. We agree with the reviewer that the criteria as established by the study of glutamatergic projection neurons in the BLA and hippocampus are not completely demonstrated in our study. However, our findings do satisfy the criteria defining engram cells as mentioned in Josselyn and Tonegawa (Science, 2020). Among the 4 methods to evaluate engram cells, we have used 2 to demonstrate observation and necessity. Additionally, a similar study of oxytocinergic cells being part of a threat memory engram was recently published (Hasan MT., et al. Neuron, 2019) wherein they satisfy the same 2 criteria as our study.

Nonetheless, we have now moderated our claims regarding the presence of a cholinergic engram throughout the manuscript.

We have changed the title of the manuscript to reflect this, and modified figure titles and legends, as well as their mention in the Results section.

Finally, we have added some text to the Discussion section to reflect our thoughts on the matter:

Lines 567-595:

“Is there a Cholinergic Component in the Associative Threat Memory Engram?

Studies examining mechanisms of learning and memory in recent years have revived Semon’s theory on memory engrams: learning must result in lasting biophysical changes that form the substrate for retrieval of the learned experience (Semon 1921, Tonegawa, Liu et al. 2015). Josselyn and Tonegawa have recently updated the definition of engram cells, requiring that these be activated by learning, modified by learning, and reactivated by subsequent presentation of the recall-inducing stimuli, resulting in memory retrieval (Josselyn and Tonegawa 2020). NBM/SI_p_ BFCNs investigated in this study indeed fulfil these criteria as they are activated by learning, show induction of Fos and altered physiological properties with recall, are reactivated by recall, and the reactivation of previously, training-activated BFCNs was necessary for recall behavior.

Multiple studies have used threat and reward learning paradigms in rodents to examine allocation of neurons to memory engrams. These studies have looked for these engram cells in regions such as cortex, amygdala and hippocampus focusing on glutamatergic pyramidal neurons (Josselyn, Köhler and Frankland 2015). However, recent work has demonstrated that memory engrams are distributed across brain-wide networks, and that reactivation of a multi-region engram more closely recapitulates natural recall behavior (Roy, Park et al. 2022).

In addition to the BLA, cholinergic neurons in the NBM/SI_p_ region project to various limbic and sensory regions such as the lateral orbital cortex, cingulate cortex, somatosensory cortex, and mediodorsal thalamus (Ananth, Rajebhosale et al. 2023). This raises the interesting possibility that the cholinergic signaling modulates various nodes of the threat memory engram circuit in conjunction with the amygdala, allowing for coordinated retrieval of engrams across distributed networks. Such coordinated activation of distributed engrams has been recently demonstrated to more closely recapitulate natural memory retrieval (Roy, Park et al. 2022). Furthermore, functionally related regions have been shown to receive their cholinergic input from the same cholinergic nucleus (Zaborszky, Csordas et al. 2015). We propose that engram-enrolled cholinergic neurons bind distributed engrams to encode stimulus-convergent, efficient memory retrieval.”

We hope these changes will prove to be satisfactory.